**[231]Pa and [230]Th in the ocean model of the Community Earth System Model (CESM1.3)**

Sifan Gu[1], Zhengyu Liu[,2]

[1]Department of Atmospheric and Oceanic Sciences and Center for Climate Research, University of Wisconsin-Madison, Madison, WI, USA

2.Atmospheric Science Program, Department of Geography, Ohio State University, Columbus, OH, USA

Correspondence to: Sifan Gu (sgu28@wisc.edu)

Abstract

Sediment [231]Pa/[230]Th activity ratio is emerging as an important proxy for deep ocean circulation in the past. In order to allow for a direct model-data comparison and to improve our understanding of sediment [231]Pa/[230]Th activity ratio, we implement [231]Pa and [230]Th in the ocean component of the Community Earth System Model (CESM). In addition to the fully coupled implementation of the scavenging behavior of [231]Pa and [230]Th with the active marine ecosystem module (p-coupled), another form of [231]Pa and [230]Th have also been implemented with prescribed particle flux fields of the present climate (p-fixed). The comparison of the two forms of [231]Pa and [230]Th helps to isolate the influence of the particle fluxes from that of ocean circulation. Under present day climate forcing, our model is able to simulate water column [231]Pa and [230]Th activity and sediment [231]Pa/[230]Th activity ratio in good agreement with available observations. In addition, in response to freshwater forcing, the p-coupled and p-fixed sediment [231]Pa/[230]Th activity ratios behave similarly over large areas of low productivity on long timescale, but can differ substantially in some regions of high productivity and on short timescale, indicating the importance of biological productivity in addition to ocean transport. Therefore, our model provides a potentially powerful tool to help the interpretation of sediment [231]Pa/[230]Th reconstructions and to improve our understanding of past ocean circulation and climate changes.

## 1. Introduction

Sediment $^{231}$Pa/$^{230}$Th activity ratio has been one major proxy for ocean circulation in the past (e.g. Yu et al. 1996; McManus et al. 2004; Gherardi et al. 2009). $^{231}$Pa (32.5 ka half-life) and $^{230}$Th (75.2 ka half-life) are produced at a constant rate approximately uniformly in the ocean by the α decay of $^{235}$U and $^{234}$U, respectively, with a production activity ratio of 0.093 (Henderson and Anderson, 2003). Water column $^{231}$Pa and $^{230}$Th are subject to particle scavenging and transport to sediments (Bacon and Anderson, 1982; Nozaki et al., 1987). Different scavenging efficiency results in different ocean residence time: $^{231}$Pa has a residence time of approximately 111 years and $^{230}$Th has a residence time of approximately 26 years (Yu et al., 1996). Longer residence time of $^{231}$Pa than $^{230}$Th makes $^{231}$Pa more subject to ocean transport and therefore in the modern ocean about 45% of $^{231}$Pa produced in the Atlantic is transported to the Southern Ocean (Yu et al., 1996), resulting a lower than 0.093 sediment $^{231}$Pa/$^{230}$Th activity ratio in the North Atlantic and higher than 0.093 sediment $^{231}$Pa/$^{230}$Th activity ratio in the Southern Ocean.

The application of the principle above to interpret sediment $^{231}$Pa/$^{230}$Th as the strength of Atlantic meridional overturning circulation (AMOC), however, can be complicated by other factors, leading to uncertainties in using $^{231}$Pa/$^{230}$Th as a proxy for past circulation (Keigwin and Boyle, 2008; Lippold et al., 2009; Scholten et al., 2008). In addition to the ocean transport, sediment $^{231}$Pa/$^{230}$Th is also influenced by particle flux and composition (Chase et al., 2002; Geibert and Usbeck, 2004; Scholten et al., 2008; Siddall et al., 2007; Walter et al., 1997). The region of a higher particle flux tends to have a higher $^{231}$Pa/$^{230}$Th (Kumar et al., 1993; Yong Lao et al., 1992), which is referred to as the "particle flux effect" (Siddall et al., 2005). Regional high particle flux in the water column will favor the removal of isotopes into the sediment, which leads to more isotopes transported into this region due to the down-gradient diffusive flux and subsequently more removal of isotopes into the sediment. Since $^{231}$Pa has a longer residence time, this effect is more prominent on $^{231}$Pa than on $^{230}$Th and therefore sediment $^{231}$Pa/$^{230}$Th will be higher in high productivity regions. Also, opal is able to scavenge $^{231}$Pa much more effectively than $^{230}$Th, leading to higher $^{231}$Pa/$^{230}$Th in high opal flux regions such as the Southern

Ocean (Chase et al., 2002). Moreover, sediment $^{231}$Pa/$^{230}$Th is suggested to record circulation change only within 1,000 m above the sediment, instead of the whole water column, complicating the interpretation of sediment $^{231}$Pa/$^{230}$Th reconstructions (Thomas et al., 2006). For example, sediment $^{231}$Pa/$^{230}$Th approaching 0.093 during Heinrich Stadial event 1(HS1) from the subtropical North Atlantic is interpreted as the collapse of AMOC (McManus et al., 2004). If sediment $^{231}$Pa/$^{230}$Th only records deepest water mass, it is possible that during HS1, AMOC shoals, as opposed to a fully collapse, yet an increase of deep water imported from the Southern Ocean featuring high $^{231}$Pa/$^{230}$Th can increase the sediment $^{231}$Pa/$^{230}$Th approaching the production ratio (0.093) (Thomas et al., 2006). Therefore, it is important to incorporate $^{231}$Pa and $^{230}$Th into climate models for a direct model-data comparison and to promote a thorough understanding of sediment $^{231}$Pa/$^{230}$Th as well as past ocean circulation.

$^{231}$Pa and $^{230}$Th have been simulated in previous modeling studies (Dutay et al., 2009; Luo et al., 2010; Marchal et al., 2000; Rempfer et al., 2017; Siddall et al., 2005). Marchal et al., (2000) simulates $^{231}$Pa and $^{230}$Th in a zonally averaged circulation model, using the reversible scavenging model of Bacon and Anderson, (1982). One step further, Siddall et al. (2005) extends Marchal et al., (2000) by including particle dissolution with prescribed particle export production in a 3-D circulation model. Rempfer et al., (2017) further couples $^{231}$Pa and $^{230}$Th with active biogeochemical model and includes boundary scavenging and sediment resuspensions to improve model performance in simulating water column $^{231}$Pa and $^{230}$Th activity. Here we follow previous studies to implement $^{231}$Pa and $^{230}$Th into the Community Earth System Model (CESM). Our standard $^{231}$Pa and $^{230}$Th are coupled with active marine ecosystem model ("p-coupled") and therefore is influenced by both ocean circulation change and particle flux change. To help to understand the influence of the particle flux, we have also implemented an auxiliary version of $^{231}$Pa and $^{230}$Th ("p-fixed") for which the particle fluxes are fixed at prescribed values. Therefore, p-fixed $^{231}$Pa/$^{230}$Th is only influenced by ocean circulation change. By comparing the p-fixed $^{231}$Pa/$^{230}$Th with the p-coupled $^{231}$Pa/$^{230}$Th, we will be able to separate the effect of circulation change from particle flux change. In

addition, the p-fixed $^{231}$Pa and $^{230}$Th can be run without the marine ecosystem
module, reducing computational cost by a factor of 3 in the ocean-alone model
simulation, making it a computationally efficient tracer for sensitivity studies.
This paper describes the details of $^{231}$Pa and $^{230}$Th in CESM and serves as a
reference for future studies using this tracer module. In section 2, we describe the
model and the implementation of $^{231}$Pa and $^{230}$Th. In sections 3, we describe the
experimental design. We will finally compare simulated $^{231}$Pa and $^{230}$Th fields with
observations, show model sensitivities on model parameter and also sediment
$^{231}$Pa/$^{230}$Th ratio response to freshwater forcing in Section 4.

**2. Model Description**
2.1 Physical Ocean Model
We implement $^{231}$Pa and $^{230}$Th in the ocean model (Parallel Ocean Program
version 2, POP2) (Danabasoglu et al., 2012) of CESM (Hurrell et al., 2013). CESM is a
state-of-the-art coupled climate model and studies describing model components
and analyzing results can be found in a special collection in Journal of Climate
(http://journals.ametsoc.org/topic/ccsm4-cesm1). We run the ocean-alone model,
which is coupled to data atmosphere, land, ice and river runoff under the normal
year forcing of CORE-II data (Large and Yeager, 2008), using the low-resolution
version of POP2 with a nominal 3° horizontal resolution and 60 vertical layers.

2.2 Biogeochemical component (BGC)
CESM has incorporated a marine ecosystem module that simulates biological
variables (Moore et al., 2013). The marine ecosystem module has been validated
against present day observations extensively (e.g. Doney et al., 2009; Long et al.,
2013; Moore et al., 2002, 2004; Moore and Braucher, 2008). The implementation of
$^{231}$Pa and $^{230}$Th requires particle fields: $CaCO_3$, opal and particulate organic carbon
(POC). These particle fields can be obtained through the ecosystem driver from the
ecosystem module (Jahn et al., 2015). The ecosystem module simulates the particle
fluxes in reasonable agreement with the present-day observations. The pattern and
magnitude of the annual mean particle fluxes ($CaCO_3$, opal, POC) leaving the
euphotic zone at 105m are similar to the satellite observations (Fig. 7.2.5 and 9.2.2
in Sarmiento and Gruber 2006) (Fig. 1 a~c): particle fluxes are higher in the high
productivity regions such as high latitudes and equatorial Pacific; opal flux is high in
the Southern Ocean. The remineralization scheme of particle is based on the ballast
model of Armstrong et al., (2002). Detailed parameterizations for particle
remineralization are documented in Moore et al., (2004) with temperature
dependent remineralization length scales for POC and opal. We do not consider dust
because it is suggested to be unimportant for $^{231}$Pa and $^{230}$Th fractionation (Chase et
al., 2002; Siddall et al., 2005).

2.3 $^{231}$Pa and $^{230}$Th implementation

$^{231}$Pa and $^{230}$Th are produced from the α decay of $^{235}$U and $^{234}$U uniformly

everywhere at constant rate $\beta^i$ ($\beta^{Pa}$ = 2.33*10$^{-3}$ dpm m$^{-3}$ yr$^{-1}$, $\beta^{Th}$ = 2.52*10$^{-2}$ dpm m$^{-3}$
yr$^{-1}$). $^{231}$Pa and $^{230}$Th are also subjective to radioactive decay with the decay
constant of $\lambda^i$ ($\lambda^{Pa}$ = 2.13*10$^{-5}$ yr$^{-1}$, $\lambda^{Th}$ = 9.22*10$^{-6}$ yr$^{-1}$).

Another important process contributes to $^{231}$Pa and $^{230}$Th activity is the

reversible scavenging by sinking particles (Bacon and Anderson, 1982), which
describes the adsorption of isotopes onto sinking particles and desorption after the
dissolution of particles. This process transports $^{231}$Pa and $^{230}$Th downward and
leads to a general increase of $^{231}$Pa and $^{230}$Th activity with depth. The reversible
scavenging considers total isotope activity ($A_t^i$) as two categories (Eq. (1)):
dissolved isotopes ($A_d^i$) and particulate isotopes ($A_p^i$) (superscript i refers to $^{231}$Pa
and $^{230}$Th) and $A_p^i$ is the sum of the isotopes associated with different particle types
($A_{j,p}^i$) (subscript j refers to different particle types: CaCO$_3$, opal and POC):

$$A_t^i = A_d^i + A_p^i = A_d^i + \sum_j A_{j,p}^i$$

(1)


Dissolved and particulate isotopes are assumed to be in equilibrium, which is a
reasonable assumption in the open ocean (Bacon and Anderson, 1982; Henderson et

al., 1999; Moore and Hunter, 1985). The ratio between the particulate isotope activity and the dissolved isotope activity is set by a partition coefficient, K (Eq. (2)):

$$K_j^i = \frac{A_{j,p}^i}{A_d^i \cdot R_j}$$

(2)

, where $R_j$ is the ratio of particle concentration ($C_j$) to the density of seawater (1024.5 kg m$^{-3}$). Subscript j refers to different particle types (CaCO$_3$, opal and POC). Values of partition coefficient K used in our control simulation follows Chase et al., 2002 and Siddall et al., 2005 (Table 2).

Particulate isotopes ($A_p^i$) will be transported by sinking particles, which is described by $w_s \frac{\partial A_p^i}{\partial z}$ (Eq. (3)), where $w_s$ is the sinking velocity. We don't differentiate between slow sinking small particles and rapid sinking large particles as in Dutay et al., (2009) and consider all particles as slowly sinking small particles with sinking velocity of $w_s$ =1000 m yr$^{-1}$ (Arsouze et al., 2009; Dutay et al., 2009; Kriest, 2002), which is similar to Rempfer et al., (2017) and Siddall et al., (2005). Any particulate isotopes ($A_p^i$) at the ocean bottom layer are removed from the ocean as sediment, which is the sink for the isotope budget. Detailed vertical differentiation scheme to calculate this term in the model is provided in the supplementary material. The reversible scavenging scheme applied here is the same as the neodymium implementation in POP2 (Gu et al., 2017).

Therefore, the conservation equation for $^{231}$Pa and $^{230}$Th activity can be written as

$$\frac{\partial A_t^i}{\partial t} = \beta^i - \lambda^i A_t^i - w_s \frac{\partial A_p^i}{\partial z} + Transport$$

(3),

where the total isotope activity is controlled by decay from U (first term), radioactive decay (second term), reversible scavenging (third term) and physical transport by the ocean model (fourth term, including advection, convection and diffusion). $A_p^i$ can be calculated by combining Eq. (1) and Eq. (2):

$$A_t^i = A_d^i + A_d^i \cdot (K_{POC}^i \cdot R_{POC} + K_{CaCO_3}^i \cdot R_{CaCO_3} + K_{opal}^i \cdot R_{opal})$$
$$= A_d^i \cdot (1 + K_{POC}^i \cdot R_{POC} + K_{CaCO_3}^i \cdot R_{CaCO_3} + K_{opal}^i \cdot R_{opal}), \qquad (4)$$
which leads to
$$A_d^i = \frac{A_t^i}{1 + K_{POC}^i \cdot R_{POC} + K_{CaCO_3}^i \cdot R_{CaCO_3} + K_{opal}^i \cdot R_{opal}}, \qquad (5)$$
put this back to Eq. (1), we get
$$A_p^i = A_t^i \cdot (1 - \frac{1}{1 + K_{POC}^i \cdot R_{POC} + K_{CaCO_3}^i \cdot R_{CaCO_3} + K_{opal}^i \cdot R_{opal}}) \qquad (6)$$

Particle fields used in the reversible scavenging can be either prescribed or
simultaneously generated from the marine ecosystem module. Therefore, two forms
of [231]Pa and [230]Th are implemented in POP2: "p-fixed" and "p-coupled". P-fixed [231]Pa
and [230]Th use particle fluxes prescribed as annual mean particle fluxes generated
from the marine ecosystem module under present day climate forcing (Fig.1). P-
coupled [231]Pa and [230]Th use particle fluxes computed simultaneously from the
marine ecosystem module. P-fixed and p-coupled [231]Pa and [230]Th can be turned on
at the case build time and the p-coupled [231]Pa and [230]Th requires the ecosystem
module to be turned on at the same time.

Comparing with previous studies of modeling [231]Pa and [230]Th, our p-fixed
version is the same as Siddall et al., (2002), except that different prescribed particle
fluxes are used. The p-coupled version allows coupling to biogeochemical module,
which is similar to Rempfer et al., (2017), but we do not include boundary
scavenging and sediment resuspensions as in Rempfer et al., (2017) because
boundary scavenging and sediment resuspensions are suggested to be unimportant
to influence the relationship between [231]Pa_p/[230]Th_p and AMOC strength (Rempfer et
al., 2017).

**3. Experiments**
We run a control experiment (CTRL) and two experiments with different
partition coefficients to show model sensitivity. We have both p-fixed and p-coupled
$^{231}$Pa and $^{230}$Th in CTRL, but only p-fixed $^{231}$Pa and $^{230}$Th in sensitivity experiments.
Equilibrium partition coefficients for $^{231}$Pa and $^{230}$Th vary among different particle
types and the magnitude of the partition coefficients for different particle types
remains uncertain (Chase et al., 2002; Chase and Robert F, 2004; Luo and Ku, 1999).
Since the control experiment in Siddall et al., (2005) is able to simulate major
features of $^{231}$Pa and $^{230}$Th distributions, we use the partition coefficients from the
control experiment in Siddall et al., (2005) in our CTRL (Table 2). Two sensitivity
experiments are performed with decreased (EXP_1) and increased (EXP_2) partition
coefficients by a factor of 5 (Table 2).
All the experiments are ocean-alone experiments with the normal year
forcing by CORE-II data (Large and Yeager, 2008). The $^{231}$Pa and $^{230}$Th activities are
initiated from 0 in CTRL and are integrated for 2,000 model years until equilibrium
is reached. EXP_1 and EXP_2 are initiated from 1,400 model year in CTRL and are
integrated for another 800 model years to reach equilibrium.
Since sediment $^{231}$Pa/$^{230}$Th in North Atlantic has been used to reflect the
strength of AMOC, to test how sediment $^{231}$Pa/$^{230}$Th in our model responds to the
change of AMOC and the change of particle fluxes, we carried out a fresh water
perturbation experiment (HOSING) with both p-fixed and p-coupled $^{231}$Pa and $^{230}$Th.
Starting from 2,000 model year of CTRL, a freshwater flux of 1 Sv is imposed over
the North Atlantic region of 50°N~70°N and the experiment is integrated for 1400
model years until both p-fixed and p-coupled sediment $^{231}$Pa/$^{230}$Th ratio have
reached quasi-equilibrium. The partition coefficients used in HOSING are the same
as in CTRL.
**4. Results**
4.1 Control Experiment
P-fixed and p-coupled version of $^{231}$Pa and $^{230}$Th in CTRL show identical
results (Fig. 2-4). P-fixed and p-coupled dissolved and particulate $^{231}$Pa and $^{230}$Th in
CTRL are highly correlated with each other with correlations greater than 0.995 and
regression coefficients are all near 1.0 ($R^2$>0.995). The correlation coefficient
between p-fixed and p-coupled sediment $^{231}$Pa/$^{230}$Th activity ratios in CTRL is 0.99
and the regression coefficient is 0.9 ($R^2$=0.98). This is expected because the particle
fields used in p-fixed version are prescribed as the climatology of the particle fields
used in the p-coupled version.  Therefore, under the same climate forcing, p-fixed
and p-coupled version of $^{231}$Pa and $^{230}$Th should be very similar. For the discussion
of results in CTRL below, we only discuss the p-fixed $^{231}$Pa and $^{230}$Th.

The residence time of both $^{231}$Pa and $^{230}$Th in CTRL are comparable with

observations. The residence time is calculated as the ratio of global average total
isotope activity and the radioactive ingrowth of the isotope. Residence time in CTRL
is 118 yr for $^{231}$Pa and 33 yr for $^{230}$Th (Table 2), which are of the same magnitude as
111 yr for $^{231}$Pa and 26 yr for $^{230}$Th in observation (Yu et al., 1996).

CTRL can simulate the general features of dissolved water column $^{231}$Pa and

$^{230}$Th activities. Dissolved $^{231}$Pa and $^{230}$Th activities increase with depth in CTRL, as
shown in two GEOTRACES transects (Deng et al., 2014; Hayes et al., 2015) in the
Atlantic (Fig. 2 and 3). The dissolved $^{231}$Pa and $^{230}$Th activities in CTRL are also at
the same order of magnitude as in observations in the most of the ocean, except that
simulated values are larger than observations in the abyssal, which is also the case
in Siddall et al., (2005) and Rempfer et al., (2017) (their Fig. 2 and 3, experiment
Re3d). Our model is unable to simulate the realistic dissolved $^{231}$Pa and $^{230}$Th
activities in the abyssal probably because boundary scavenging and sediment
resuspensions are not included in our model. In Rempfer et al., 2017, without
boundary scavenging and sediment resuspension, dissolved $^{231}$Pa and $^{230}$Th
activities are quite large in the deep ocean. However, if boundary scavenging and
sediment resuspension are included, the water column dissolved $^{231}$Pa and $^{230}$Th
activity is in the right magnitude compared with observation. Therefore, we hypothesize
that with boundary scavenging and sediment resuspensions added, dissolved $^{231}$Pa
and $^{230}$Th activities in the abyssal should be greatly reduced.

A more quantitative model-data comparison is shown in Fig. 5. The linear

regression coefficient between model results and observations (references of
observations are listed in Table 3), an indication of model ability to simulate $^{231}$Pa
and $^{230}$Th activity (Dutay et al., 2009), is near 1.0 for dissolved $^{231}$Pa and $^{230}$Th (1.02
for $[^{231}$Pa$]_d$ and 1.14 for $[^{230}$Th$]_d$), suggesting that CTRL can simulate the dissolved
$^{231}$Pa and $^{230}$Th in good agreement with observations. However, the simulation of
the particulate activity is not as good as the dissolved activity. Particulate activity is
overall larger than observation in the surface ocean and smaller than observation in
the deep ocean for both particulate $^{231}$Pa and $^{230}$Th. The regression coefficient for
particulate $^{231}$Pa and $^{230}$Th is 0.02 for $[^{231}$Pa$]_p$ and 0.05 for $[^{230}$Th$]_p$. The poor
performance in simulating water column particulate $^{231}$Pa and $^{230}$Th activities is also
in previous modeling studies (Dutay et al., 2009; Siddall et al., 2005), because of
similar modelling scheme applied. However, the simulated $^{231}$Pa$_p$/$^{230}$Th$_p$ is in
reasonable agreement with observations. The $^{231}$Pa$_p$/$^{230}$Th$_p$ along two GEOTRACES
transects (Fig. 2 and 3) show the similar pattern and magnitude as in Rempfer et al.,
(2017), consistent with observations. Decrease of $^{231}$Pa$_p$/$^{230}$Th$_p$ with depth is well
simulated, which is suggested to be caused by the lateral transport of $^{231}$Pa from
North Atlantic to Southern Ocean by AMOC (Gherardi et al., 2009; Lippold et al.,
2011, 2012a; Luo et al., 2010; Rempfer et al., 2017).

The sediment $^{231}$Pa/$^{230}$Th in CTRL is overall consistent with observations

(references of observations are listed in Table 3). The North Atlantic shows low
sediment $^{231}$Pa/$^{230}$Th activity ratio as in observations because $^{231}$Pa is more subject
to the southward transport by active ocean circulation than $^{230}$Th because of its
longer residence time. The Southern Ocean maximum in the sediment $^{231}$Pa/$^{230}$Th
activity ratio is also simulated in CTRL. High opal fluxes in the Southern Ocean,
which preferentially removes $^{231}$Pa into sediment ($K_{opal}^{231Pa} > K_{opal}^{230Th}$) (Chase et al.,
2002), leading to increased sediment $^{231}$Pa/$^{230}$Th activity ratio. In addition,
upwelling in the Southern Ocean brings up deep water enriched with $^{231}$Pa, which is
transported from the North Atlantic, to shallower depth and further contribute to
the scavenging. CTRL can also produce higher sediment $^{231}$Pa/$^{230}$Th activity ratio in
regions with high particle production (e.g. the Eastern equatorial Pacific, the North
Pacific and the Indian Ocean) due to the "particle flux effect". Specifically, in North
Atlantic, the distribution of sediment $^{231}$Pa/$^{230}$Th matches the distribution of
particle, especially opal, production: sediment $^{231}$Pa/$^{230}$Th is higher where opal
production is high, and vice versa (Fig. 4 and Fig. 1c). Quantitatively, the regression
coefficient between sediment $^{231}$Pa/$^{230}$Th in CTRL and observation in the Atlantic is
0.86, which is larger than in other basins. This suggests that sediment $^{231}$Pa/$^{230}$Th is
better simulated in the Atlantic than in other basins. One possible explanation is that
sediment $^{231}$Pa/$^{230}$Th in the Atlantic is controlled by both ocean circulation and
particle flux, while in other basins sediment $^{231}$Pa/$^{230}$Th is controlled almost only by
particle flux. With active AMOC, the north south gradient of sediment $^{231}$Pa/$^{230}$Th
can be simulated. However, for example, in the Southern Ocean, sediment
$^{231}$Pa/$^{230}$Th is dominantly controlled by opal flux, which varies on small scales and is
difficult for simulation. Therefore, model performance in simulating sediment
$^{231}$Pa/$^{230}$Th in the Southern Ocean is not as good as in the Atlantic.

4.2 Sensitivity on partition coefficient K

In this section, we show model sensitivity on partition coefficient by

increasing and decreasing the partition coefficient, K, by a factor of 5, but keeping
the relative ratio for different particles the same (Table 2). Our model shows similar
model sensitivity as in Siddall et al., (2005) as discussed below.

As stated in Siddall et al., (2005), the isotope decay term in Eq. (3) is three

orders of magnitude less than the production term. If we neglect the transport term
and the decay term in Eq. (3) and assume particulate phase activity at the surface as
0, when reach equilibrium, the activity of particulate phase will be as in Eq. (7). Eq.
(7) combined with Eq. (2) and $R_i = \frac{F}{w_s * \rho}$, we can obtain Eq. (8). Under the
assumption that there is isotope decay and ocean transport, Eq. (7) suggests that the
particulate isotope activity depends on the production rate and settling velocity and
will increase linearly with depth. Eq. (8) suggests that the dissolved isotope activity
depends on the production rate, partition coefficient K and particle flux and will also
increase linearly with depth. Any departure from this linear relationship with depth
is due to ocean transport, which is suggested by observations (Bacon and Anderson,
1982; Roy-Barman et al., 1996). Results of Eq. (7) and Eq. (8) can help to understand
the differences in Exp_1 and Exp_2.

Increasing K will decrease water column dissolved [231]Pa and [230]Th activities

but won't change particulate [231]Pa and [230]Th too much (Fig. 6). Magnitude of
dissolved [231]Pa and [230]Th in Exp_1 (smaller K) is at least one order larger than that
in Exp_2 (larger K), while magnitude of particulate [231]Pa and [230]Th in Exp_1 and
Exp_2 is in the same order. As suggested by Eq. (8), if there is no isotope decay and
no ocean transport, larger K will lead to smaller dissolved isotope activity but
unchanged particulate activity. Intuitively, larger K will lead to more [231]Pa and [230]Th
attached to particles and further buried into sediment, which increases the sink for
the [231]Pa and [230]Th budget. With the sources for [231]Pa and [230]Th staying the same,
dissolved [231]Pa and [230]Th will be reduced. Increasing K will also reduce the vertical
gradient of dissolved [231]Pa and [230]Th as reversible scavenging act as the vertical
transport and increase this vertical transport can decrease the vertical gradient.
However, changes in the particulate [231]Pa and [230]Th is relatively small (Fig. 6). Eq.
(7) suggests that particulate phase activity it is independent of K. Therefore,
changing K will have limited influence on particulate phase activity.

$$A_p^i(z) = \frac{\beta^i}{w_s} \cdot z \tag{7}$$


$$A_d^i(z) = \frac{\rho \beta^i}{K^i F} \cdot z \tag{8}$$


Increasing K will also reduce the spatial gradient in sediment [231]Pa/[230]Th

activity ratio and vice versa (Fig. 7). Larger K will decrease the [231]Pa and [230]Th
residence time and most isotopes produced in the water column are removed into
sediment locally (Table 2). Therefore, sediment [231]Pa/[230]Th ratio becomes more
homogeneous and approaching the production ration of 0.093 (Fig. 7b). The
deviation (the root mean squared error) of sediment [231]Pa/[230]Th is 0.0726 in CTRL,
0.0770 in Exp_1 and 0.0739 in Exp_2. The linear regression coefficients between
sediment [231]Pa/[230]Th in the model and the observations are listed in Table S1 in the
supplementary information. Although the performance of global sediment
$^{231}$Pa/$^{230}$Th in Exp_1 is better than CTRL, the performance of Atlantic $^{231}$Pa/$^{230}$Th in
Exp_1 is worse. We consider better simulating sediment $^{231}$Pa/$^{230}$Th in the Atlantic
is more important since the most important application of sediment $^{231}$Pa/$^{230}$Th is
using sediment $^{231}$Pa/$^{230}$Th in the North Atlantic to reconstruct past AMOC. In
addition, water column isotope activity is too large in Exp_1 compared with
observation. Therefore, the partition coefficient in CTRL is of the right order of
magnitude.

4.3. Sediment $^{231}$Pa/$^{230}$Th ratio in HOSING
Potential changes in the export of biogenic particles makes using $^{231}$Pa/$^{230}$Th
ratio to reconstructing AMOC strength under debate. In response to freshwater
perturbation in the North Atlantic, both biological productivity and AMOC strength
will change and will influence sediment $^{231}$Pa/$^{230}$Th in different ways. Our model
with p-fixed and p-coupled $^{231}$Pa and $^{230}$Th can help to detangle these two effects. In
this section, we examine the sediment $^{231}$Pa/$^{230}$Th (p-fixed and p-coupled) response
in the North Atlantic to idealized fresh water perturbation.
In HOSING, after applying freshwater forcing to the North Atlantic, AMOC
strength quickly decreases to a minimum of 2 Sv (AMOC_off) (Fig. 9a). During the
AMOC_off state, compared with CTRL with active AMOC (AMOC_on), p-fixed
sediment $^{231}$Pa/$^{230}$Th shows an overall increase in the North Atlantic and a decrease
in the South Atlantic (Fig. 10b) because of the reduced southward transport of $^{231}$Pa
from the North Atlantic by AMOC, consistent with paleo proxy evidence there (e.g.
Gherardi et al., 2005, 2009; McManus et al., 2004). The overall increase of sediment
$^{231}$Pa/$^{230}$Th ratio in the North Atlantic in response to the AMOC collapse can be seen
more clearly in the time evolution of the sediment $^{231}$Pa/$^{230}$Th ratio averaged from
20°N to 60°N in the North Atlantic (Fig.9b, green). Quantitatively, the $^{231}$Pa/$^{230}$Th
increases from 0.074 in AMOC_on to 0.098 in AMOC_off in the p-fixed version,
approaching the production ration of 0.093. This increase of $^{231}$Pa/$^{230}$Th is also in
the subtropical North Atlantic from the two sites near Bermuda Rise (Fig. 9e and f),
which is of comparable magnitude with the change from LGM to HS1 in
reconstructions there (McManus et al., 2004). In addition, the pattern of p-fixed
(Fig.10a) sediment $^{231}$Pa/$^{230}$Th ratio during the Atlantic in AMOC_off state is similar
to the opal distribution (Fig.1b) because, without active circulation, sediment
$^{231}$Pa/$^{230}$Th ratio is more controlled by particle flux effect, which is similar to the
Pacific in CTRL. It is further noted that our p-fixed sediment $^{231}$Pa/$^{230}$Th ratio in
HOSING behaves similarly to that in Siddall et al., (2007).

The overall increase in p-fixed sediment $^{231}$Pa/$^{230}$Th ratio in the North

Atlantic is not homogenous and the magnitude of the change between AMOC_on and
AMOC_off varies with location, depending on the distribution of particle flux,
especially the opal flux (Fig.9 and 10). The maximum increase in p-fixed sediment
$^{231}$Pa/$^{230}$Th ratio occurs near 40°N western Atlantic (Fig. 10a), where the opal
production in our model is maximum in North Atlantic (Fig. 1b). The sediment
$^{231}$Pa/$^{230}$Th ratio in this region during AMOC_on is larger than production ratio of
0.093 because opal maximum provides extra $^{231}$Pa to this region ("particle flux
effect"), which overwhelms the active ocean circulation transporting $^{231}$Pa
southward outside this region (Fig. 9d, green). During AMOC_off, without active
ocean circulation, the particle flux effect becomes even stronger because less $^{231}$Pa is
transported out of the North Atlantic and p-fixed sediment $^{231}$Pa/$^{230}$Th ratio
becomes even larger. It should be noted that the opal maximum in this region is not
in the observation (Fig. 7.2.5 in Sarmiento and Gruber 2006). However, our
sediment $^{231}$Pa/$^{230}$Th response in HOSING is self-consistent with the particle flux in
our model since the location of maximum $^{231}$Pa/$^{230}$Th increase matches the location
of opal flux in our model.

In most regions of the Atlantic, p-coupled sediment $^{231}$Pa/$^{230}$Th shows a

similar response to p-fixed $^{231}$Pa/$^{230}$Th in HOSING. The evolution of p-fixed and p-
coupled sediment $^{231}$Pa/$^{230}$Th activity ratio in HOSING are highly correlated (Fig.
11a). The change of sediment $^{231}$Pa/$^{230}$Th ratio from AMOC_on to AMOC_off are
similar in both p-fixed and p-coupled version (Fig.11b). The correlation between p-
fixed and p-coupled sediment $^{231}$Pa/$^{230}$Th ratio change from AMOC_on to AMOC_off
is 0.72 (1455points) and the linear regression coefficient is 0.71 ($R^2$ = 0.52). High
correlation between p-fixed and p-coupled response mainly happens over low
productivity regions (Fig.1, 10, and 11), where circulation effect on sediment
$^{231}$Pa/$^{230}$Th is more important than the particle flux change in HOSING.

In spite of these similarities discussed above,  the responses of p-fixed and p-
coupled sediment $^{231}$Pa/$^{230}$Th to the fresh water forcing can differ significantly in
high productivity regions because of  the productivity change. With persistent
freshwater forcing over the North Atlantic, most regions in the North Atlantic show
reduced production of CaCO$_3$, opal and POC (Fig. 8). Productivity in the North
Atlantic is suggested to be halved during AMOC collapse because of increased
stratification, which reduces nutrient supply from deep ocean (Schmittner, 2005). In
our model, the productivity in the mid-latitude North Atlantic is indeed greatly
reduced after the freshwater forcing is applied. For example, opal production from
30°N-50°N in the Atlantic at the end of HOSING is reduced by 50%~90% of its
original value in CTRL. However, opal production increases in high latitude North
Atlantic (north of 50°N).  The pattern of opal production changes with high opal
production region shifts northward in HOSING (Fig. 8 d, e and f). These particle flux
changes will influence sediment $^{231}$Pa/$^{230}$Th as discussed below.

North of 50°N in the Atlantic, the opal productivity increases during
AMOC_off (Fig. 8f) and will result an increase in sediment $^{231}$Pa/$^{230}$Th. The increase
caused by greater opal productivity enhances the sediment $^{231}$Pa/$^{230}$Th increase
caused by reduced AMOC. Therefore, the increase in p-coupled sediment $^{231}$Pa/$^{230}$Th
from AMOC_on to AMOC_off is larger than p-fixed sediment $^{231}$Pa/$^{230}$Th change
(Fig.9c).

In the mid-latitude North Atlantic, the opal productivity decreases during
AMOC_off (Fig.8 f) and will lead to a decrease in sediment $^{231}$Pa/$^{230}$Th, which is
opposite to the effect of reduced AMOC. P-coupled sediment $^{231}$Pa/$^{230}$Th shows an
initial decrease in first 200 years (Fig.9 d, e, and f, red dash lines) caused by the
reduced opal productivity. But this decrease trend is reversed eventually, suggesting
that the influence of particle flux change is overwhelmed by the effect of reduced
AMOC. It the long run, most regions in the subtropical and mid-latitude Atlantic
show increased sediment $^{231}$Pa/$^{230}$Th in HOSING (Fig.10 d), indicating the dominant
effect of reduced AMOC. However, sediment $^{231}$Pa/$^{230}$Th at 40°N west Atlantic,
where opal productivity is maximum during AMOC_on, show a decrease from
AMOC_on to AMOC_off (Fig.9 d and Fig.10 d). During AMOC_on, the opal productivity
maximum at 40°N west Atlantic lead to regional maximum sediment $^{231}$Pa/$^{230}$Th
because of the particle flux effect (Fig. 4). During AMOC_off, this opal productivity
maximum is eliminated (Fig.8 e) and there is no more extra $^{231}$Pa supplied by
surroundings to this region, which leads to a decrease in sediment $^{231}$Pa/$^{230}$Th. This
decrease in sediment $^{231}$Pa/$^{230}$Th caused by productivity change is greater than the
increase caused by the reduced AMOC. Therefore, sediment $^{231}$Pa/$^{230}$Th experiences
a decrease from AMOC_on to AMOC_off at this location (Fig.9 d and Fig.10 d). Our
results suggest that although the circulation effect is more dominant than the
particle flux change in controlling sediment $^{231}$Pa/$^{230}$Th on long time scale over
most of North Atlantic (Fig. 11), particle flux change can be important on short time
scale and in high productivity regions. With p-fixed and p-coupled $^{231}$Pa and $^{230}$Th,
our model can help to detangle the circulation effect and particle flux effect.

It has been suggested that the particulate $^{231}$Pa/$^{230}$Th response to the change

of AMOC depends on the location and depth. Above 2km and high latitude North
Atlantic, particulate $^{231}$Pa/$^{230}$Th decreases with the increased AMOC (Rempfer et al.,
2017). Our results are consistent with this finding (Fig. 12 a and b). Both p-fixed and
p-coupled particulate $^{231}$Pa/$^{230}$Th show similar patterns of change from AMOC_on to
AMOC_off: decrease in particulate $^{231}$Pa/$^{230}$Th at shallow depth and north of 60°N
and increase in particulate $^{231}$Pa/$^{230}$Th below 2km and south of 60°N during
AMOC_off. Therefore, sediment depth should also be taken into consideration when
interpreting sediment $^{231}$Pa/$^{230}$Th. Since the pattern in p-coupled is similar to the
pattern in p-fixed, the opposite particulate $^{231}$Pa/$^{230}$Th changes in shallow and deep
North Atlantic is associated with AMOC change. During AMOC_on, upper limb of
AMOC (about upper 1km) transport water northward, which provides extra $^{231}$Pa to
North Atlantic and particulate $^{231}$Pa/$^{230}$Th is larger than the production ratio of
0.093. In contrast, the lower limb of AMOC (2km-3km) features southward
transport, which transports $^{231}$Pa to the Southern Ocean and particulate $^{231}$Pa/$^{230}$Th
is smaller than the production ratio of 0.093 (Fig. 12 solid). Particulate $^{231}$Pa/$^{230}$Th
decreases with depth (Fig. 12 c solid). During AMOC_off, ocean transport of $^{231}$Pa is
greatly reduced. Therefore, shallow (deep) depth experiences a decrease (increase)
in particulate $^{231}$Pa/$^{230}$Th and the vertical gradient in the particulate $^{231}$Pa/$^{230}$Th is
also greatly reduced (Fig. 12 c dash). Our results support that the depth dependence
of particulate $^{231}$Pa/$^{230}$Th is mainly caused by lateral transport of $^{231}$Pa by
circulation (Gherardi et al., 2009; Lippold et al., 2011, 2012a; Luo et al., 2010;
Rempfer et al., 2017).

Overall, our model is able to simulate the correct magnitude of the sediment

$^{231}$Pa/$^{230}$Th ratio response to the freshwater forcing. Our experiments suggest that
the change of circulation is the dominant factor that influences   sediment
$^{231}$Pa/$^{230}$Th on long time scale over most of the globe  in the idealized hosing
experiment, although the detailed difference between p-fixed and p-coupled
sediment $^{231}$Pa/$^{230}$Th ratio response to freshwater forcing in different locations can
be complicated.


**5. Summary**

$^{231}$Pa and $^{230}$Th have been implemented in the ocean model of the CESM in

both the p-coupled and p-fixed forms. Our control experiment under present day
climate forcing is able to simulate most $^{231}$Pa and $^{230}$Th water column activity and
sediment $^{231}$Pa/$^{230}$Th activity ratio consistent with observations by using the
parameters that are suggested by Chase et al., (2002) and used in Siddall et al.
(2005). Our sensitivity experiments with varying parameters suggest that these
parameters are of the right order of magnitude.

Furthermore, our model is able to simulate the overall sediment $^{231}$Pa/$^{230}$Th

ratio change in the North Atlantic with a magnitude comparable to the
reconstruction in response to the collapse of AMOC, although the detailed response
can be complicated in different regions. Finally, the p-fixed form is able to capture
many major features of that of the p-coupled form over large ocean areas on long
time scale, although the two forms can also differ significantly in some regions,
especially the region with high opal productivity.

Much remains to be improved in our $^{231}$Pa and $^{230}$Th module in the future. For example, the model can be further improved by including nepheloid layers to better simulate water column $^{231}$Pa and $^{230}$Th activity as in Rempfer et al. (2017). In addition, partition coefficient for different particles can be further tuned , which can improve our understanding of the affinity of $^{231}$Pa and $^{230}$Th to different particles, complementing the limited observational studies available (e.g. Chase et al., 2002; Scholten et al., 2005; Walter et al., 1997). At present, as the first attempt to implement $^{231}$Pa and $^{230}$Th in the CESM with both p-fixed and p-coupled versions, our model can serve as a useful tool to improve our understanding of the processes of $^{231}$Pa and $^{230}$Th as well as interpretations of sediment $^{231}$Pa/$^{230}$Th reconstructions for past ocean circulation and climate changes.

**Code availability:**

The $^{231}$Pa and $^{230}$Th isotope source code of both p-fixed and p-coupled versions for CESM1.3 is included as supplementary material here.

**Acknowledgement:**
This work is supported by NSF P2C2 program (NSF 1401778 and NSF1600080), DOE DE-SC0006744 and NSFC 41630527 and 41130105. Computing resources (ark:/85065/d7wd3xhc) were provided by the Climate Simulation Laboratory at NCAR's Computational and Information Systems Laboratory, sponsored by the National Science Foundation and other agencies.

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

| Variable | Symbol | Value | Units |
|---|---|---|---|
| Production of $^{231}$Pa from U decay | $\beta^{Pa}$ | $2.33*10^{-3}$ | dpm m$^{-3}$ yr$^{-1}$ |
| Production of $^{230}$Th from U decay | $\beta^{Th}$ | $2.52*10^{-2}$ | dpm m$^{-3}$ yr$^{-1}$ |
| Decay constant of $^{231}$Pa | $\lambda^{Pa}$ | $2.13*10^{-5}$ | yr$^{-1}$ |
| Decay constant of $^{230}$Th | $\lambda^{Th}$ | $9.22*10^{-6}$ | yr$^{-1}$ |
| Index for $^{231}$Pa and $^{230}$Th | $i$ | | |
| Index for particle type | $j$ | | |
| Total isotope activity | $A_t$ | | dpm m$^{-3}$ |
| Dissolved isotope activity | $A_d$ | | dpm m$^{-3}$ |
| Particle associated activity | $A_p$ | | dpm m$^{-3}$ |
| Particle settling velocity | $w_s$ | 1000 | m yr$^{-1}$ |
| Particle concentration | $C$ | | kg m$^{-3}$ |
| Density of seawater | | 1024.5 | kg m$^{-3}$ |
| Ratio between particle concentration and density of seawater | $R$ | | |

Table 1. List of parameters, abbreviations and values.

| | CTRL | | EXP_1 | | EXP_2 | |
|---|---|---|---|---|---|---|
| | $^{231}$Pa | $^{230}$Th | $^{231}$Pa | $^{230}$Th | $^{231}$Pa | $^{230}$Th |
| $K_{CaCO_3}$ | $2.5*10^5$ | $1.0*10^7$ | $5*10^4$ | $2*10^6$ | $1.25*10^6$ | $5*10^7$ |
| $K_{opal}$ | $1.67*10^6$ | $5*10^5$ | $3.33*10^5$ | $1*10^5$ | $8.33*10^6$ | $2.5*10^6$ |
| $K_{POC}$ | $1.0*10^7$ | $1.0*10^7$ | $2*10^6$ | $2*10^6$ | $5*10^7$ | $5*10^7$ |
| $\tau$ (yr) | 118 | 33 | 501 | 143 | 27 | 9 |

Table 2. Partition coefficients for different particle types and residence time for
$^{231}$Pa and $^{230}$Th in different experiments. Partition coefficients used in CTRL follows
(Chase et al., 2002; Siddall et al., 2005). Both p-coupled and p-fixed versions are
enabled in CTRL, which yields identical results (discussed in section 4.1). Only p-
fixed version is enabled in Exp_1 and Exp_2. The residence time ($\tau$) is for p-fixed
version in each experiment.

| WATER COLUMN ACTIVITY | Holocene core-top $^{231}$Pa/$^{230}$Th |
|---|---|
| (Guo et al., 1995) | (Yu, 1994) |
| (Cochran et al., 1987) | (DeMaster, 1979) |
| (Nozaki et al., 1987) | (Bacon and Rosholt, 1982) |
| (Bacon and Anderson, 1982) | (Mangini and Diester-Hass, 1983) |
| (Bacon et al., 1989) | (Kumar, 1994) |

| | |
|---|---|
| (Huh and Beasley, 1987) | (Yang et al., 1986) |
| (Rutgers van der Loeff and Berger, 1993) | (Anderson et al., 1983) |
| (Nozaki et al., 1981) | (Anderson et al., 1994) |
| (Nozaki and Nakanishi, 1985) | (Ku, 1966) |
| (Mangini and Key, 1983) | (Ku et al., 1972) |
| (Nozaki and Horibe, 1983) | (Frank et al., 1994) |
| (Moore, 1981) | (Shimmield et al., 1986) |
| (Nozaki and Yamada, 1987) | (Frank, 1996) |
| (Roy-Barman et al., 1996) | (Yong Lao et al., 1992) |
| (Nozaki and Yang, 1987) | (Francois et al., 1993) |
| (Moran et al., 1995) | (Anderson et al., 1990) |
| (Luo et al., 1995) | (Mangini and Sonntag, 1977) |
| (Colley et al., 1995) | (Schmitz et al., 1986) |
| (Scholten et al., 1995) | (Shimmield and Price, 1988) |
| (Cochran et al., 1995) | (Yong-Liang Yang et al., 1995) |
| (Vogler et al., 1998) | (Müller and Mangini, 1980) |
| (Moran et al., 1997) | (Mangini and U., 1987) |
| (Edmonds et al., 1998) | (Scholten et al., 1995) |
| (Moran et al., 2001) | (Walter et al., 1997) |
| (Edmonds et al., 2004) | (Lippold et al., 2011) |
| (Okubo et al., 2007b) | (Lippold et al., 2012b) |
| (Coppola et al., 2006) | (Bradtmiller et al., 2007) |
| (Moran et al., 2002) | (Gherardi et al., 2005) |
| (Okubo et al., 2004) | (Gutjahr et al., 2008) |
| (Okubo et al., 2007a) | (Hall et al., 2006) |
| (Okubo et al., 2012) | (Lippold et al., 2011) |
| (Robinson et al., 2004) | (Roberts et al., 2014) |
| (Thomas et al., 2006) | (Bradtmiller et al., 2014) |
| (Trimble et al., 2004) | (Burckel et al., 2016) |
| (Venchiarutti et al., 2011) | (Hoffmann et al., 2013) |
| (Hsieh et al., 2011) | (Jonkers et al., 2015) |
| (Scholten et al., 2008) | (Negre et al., 2010) |
| (Luo et al., 2010) | |
| (Deng et al., 2014) | |
| (Hayes et al., 2013) | |
| (Hayes et al., 2015) | |

Table 3. References for observations of water column $^{231}$Pa and $^{230}$Th activity (left
column) and Holocene core-top $^{231}$Pa/$^{230}$Th (right column).



Figures:

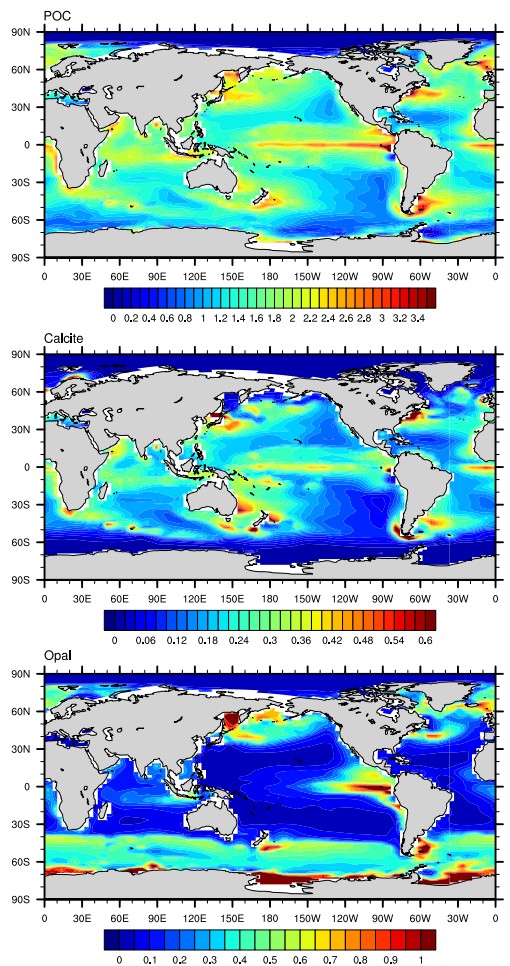


Figure 1. Annual mean particle fluxes in CESM. (a) CaCO$_3$ flux at 105m (mol m$^{-2}$ yr$^{-1}$).
(b) Opal flux at 105m (mol m$^{-2}$ yr$^{-1}$). (c) POC flux at 105m (mol m$^{-2}$ yr$^{-1}$).


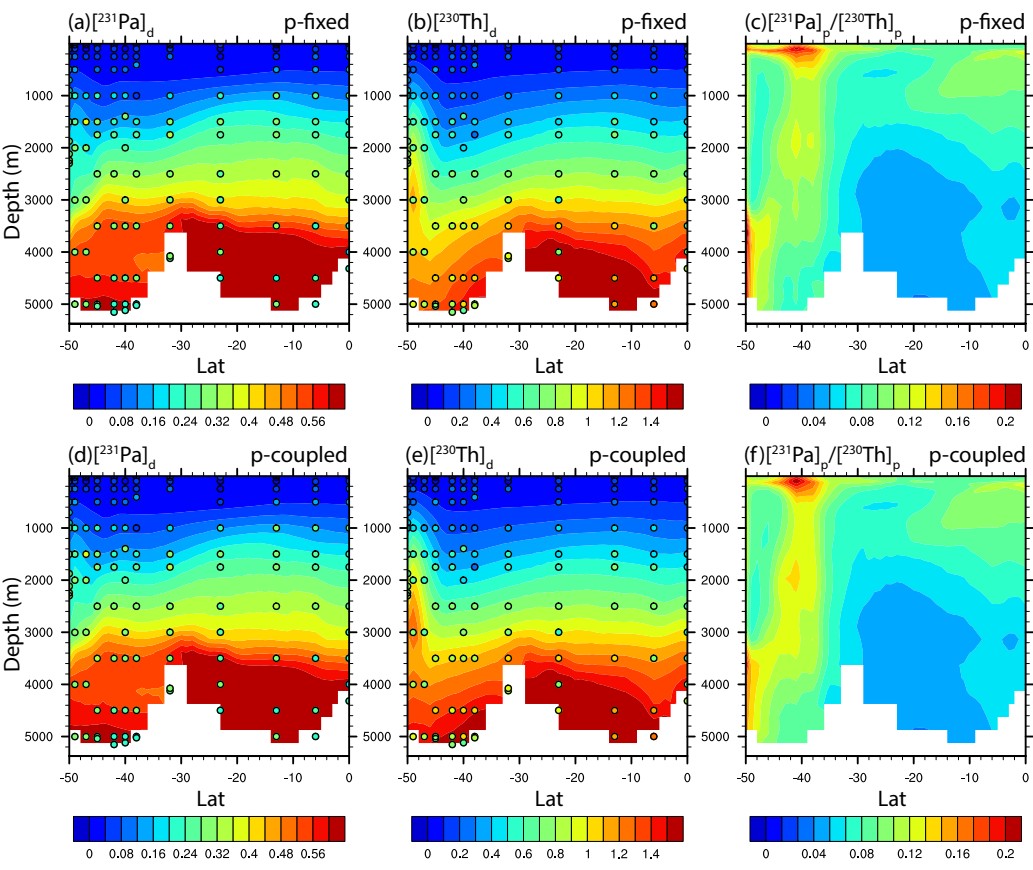


Figure 2. Dissolved ²³¹Pa, dissolved ²³⁰Th and particulate ²³¹Pa/²³⁰Th in CTRL along
GEOTRACES transect GA02S (Deng et al., 2014) (the track is indicated in Fig. S4) for
both p-fixed (top row) and p-coupled (bottom row) ²³¹Pa and ²³⁰Th (colored
contour). Observations of dissolved ²³¹Pa and ²³⁰Th activity are superimposed as
colored circles using the same color scale.

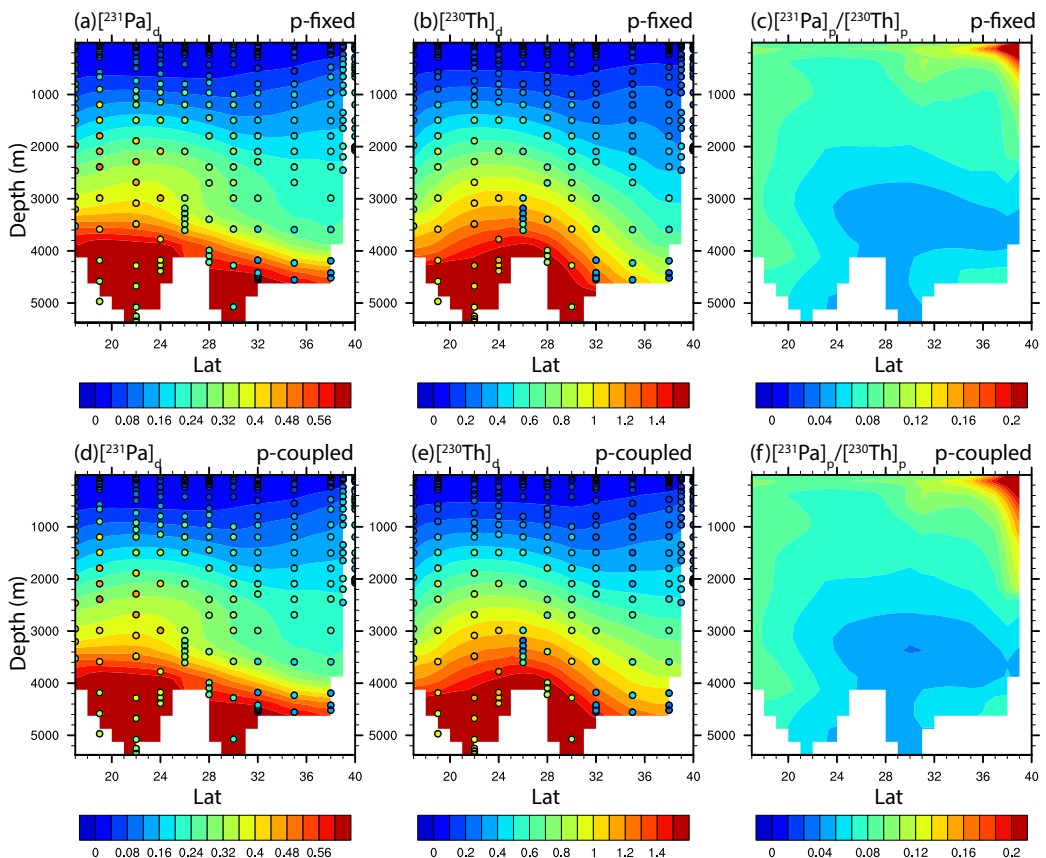

935

Figure 3. Dissolved $^{231}$Pa, dissolved $^{230}$Th and particulate $^{231}$Pa/$^{230}$Th in CTRL along GEOTRACES transect GA03 (Hayes et al., 2015) (the track is indicated in Fig. S4) for both p-fixed (top row) and p-coupled (bottom row) $^{231}$Pa and $^{230}$Th (colored contour). Observations of dissolved $^{231}$Pa and $^{230}$Th activity are superimposed as colored circles using the same color scale.

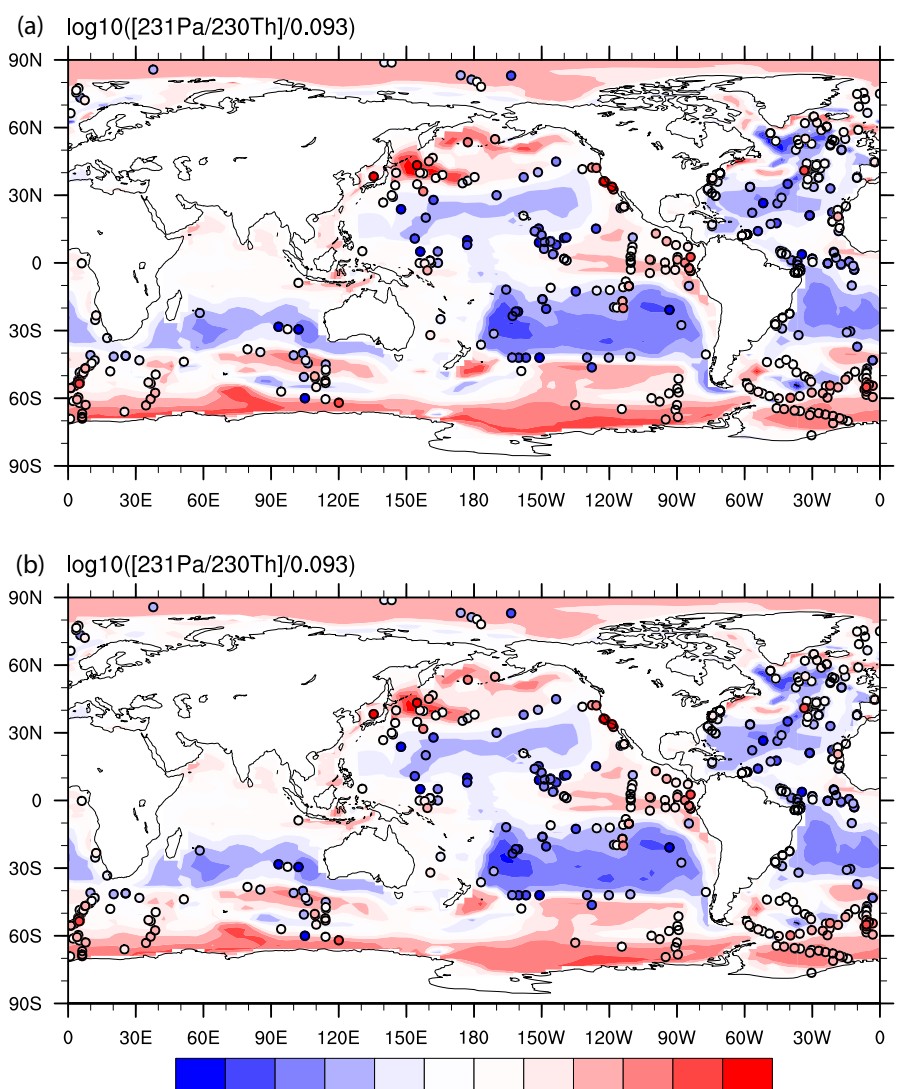


Figure 4. Sediment [231]Pa/[230]Th activity ratio in CTRL for both p-fixed (a) and p-
coupled version (b). Observations are attached as filled cycles using the same color
map. The [231]Pa/[230]Th activity ratio is plotted relative to the production ratio of
0.093 on a $\log_{10}$ scale.



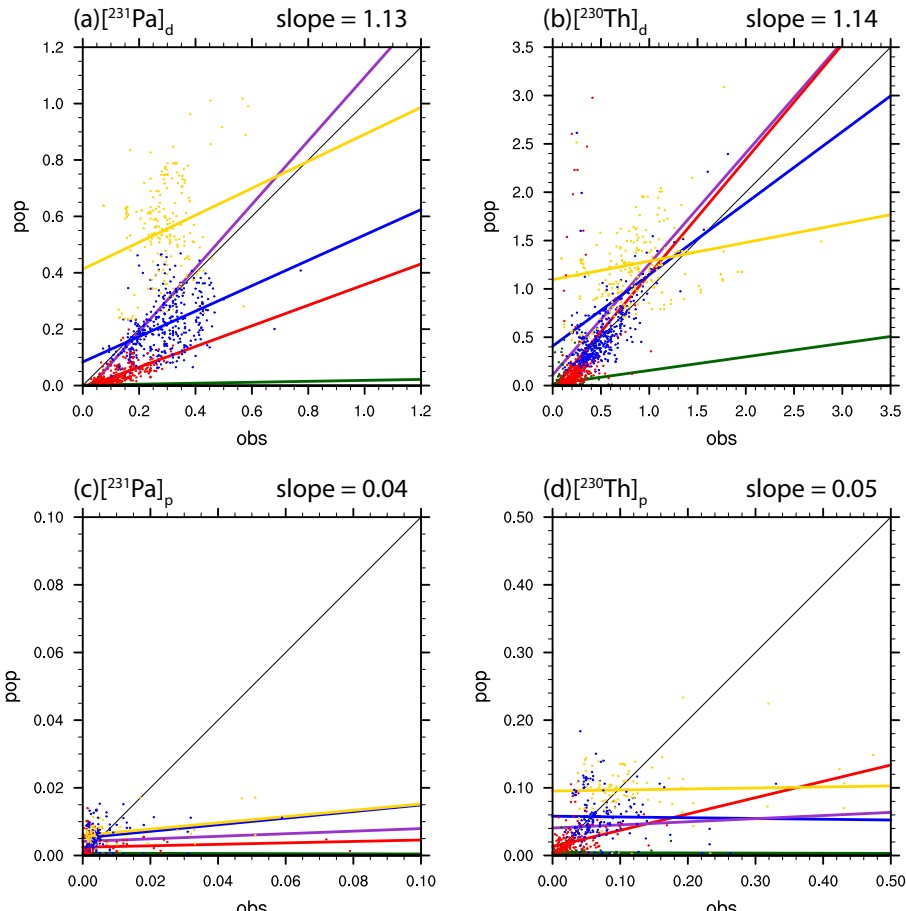



Figure 5. Scatter plot of global dissolved and particulate $^{231}$Pa and $^{230}$Th between observation and CTRL (p-fixed) (unit: dpm/m$^3$). (a) dissolved $^{231}$Pa; (b) particulate $^{231}$Pa; (c) dissolved $^{230}$Th; (d) particulate $^{230}$Th. Observations in different depth range are indicated by different colors: green for 0-100m; red for 100m-1,000m; blue for 1,000m-3,000m and yellow for deeper than 3,000m. Purple line is the least squared linear regression line for all depth range, the slope of which is indicated at the top right of each plot. Green line is the least squared linear regression line for depth from 0-100m. Red line is the least squared linear regression line for depth from 100m -1,000m. Blue line is the least squared linear regression line for depth from 1,000m-3,000m. Yellow line is the least squared linear regression line for depth deeper than 3,000m.

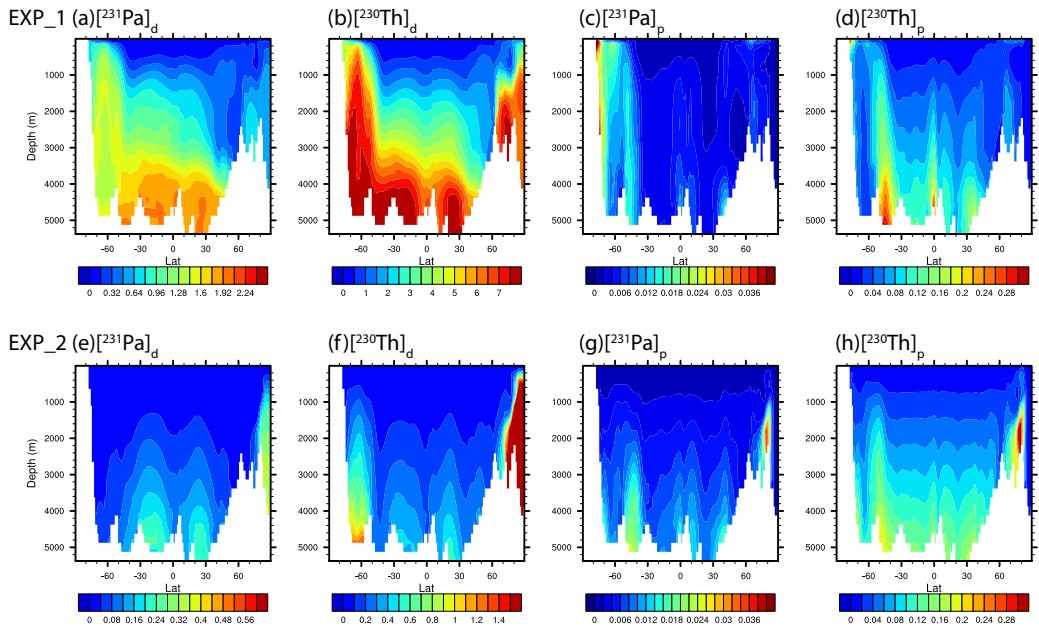


Figure 6. Atlantic zonal mean dissolved and particulate [231]Pa and [230]Th in EXP_1 and
EXP_2 (unit: dpm/m[3]). EXP_1: (a) dissolved [231]Pa; (b) dissolved [230]Th; (c)
particulate [231]Pa; (d) particulate [230]Th. EXP_2: (e) dissolved [231]Pa; (f) dissolved
[230]Th; (g) particulate [231]Pa; (h) particulate [230]Th.

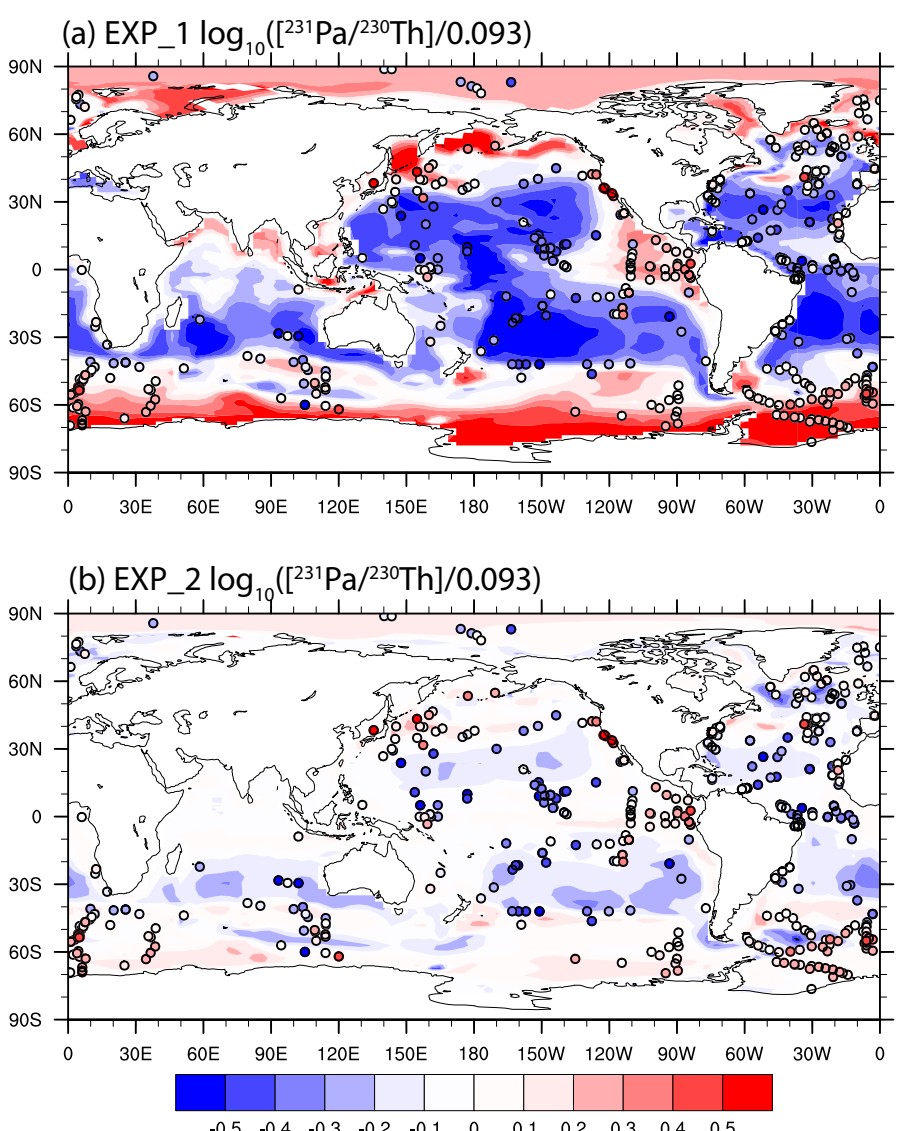


Figure 7. Sediment $^{231}$Pa/$^{230}$Th activity ratio in EXP_1 (a) and EXP_2 (b).

Observations are attached as filled cycles using the same color map. The $^{231}$Pa/$^{230}$Th

activity ratio is plotted relative to the production ratio of 0.093 on a $\log_{10}$ scale.


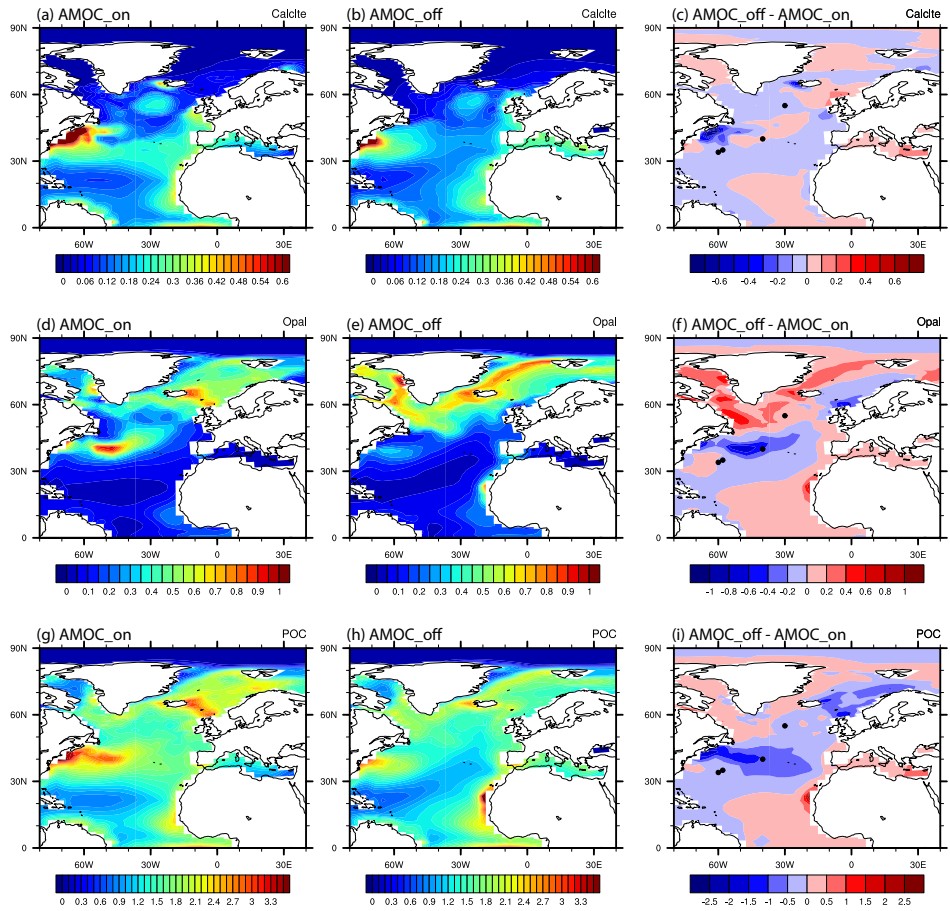


Figure 8. Comparison of particle fluxes between AMOC_on and AMOC_off. CaCO$_3$ flux
at 105m (mol m$^{-2}$ yr$^{-1}$) during AMOC_on (a), AMOC_off (b) and difference between
AMOC_off and AMOC_on. (b) Opal flux at 105m (mol m$^{-2}$ yr$^{-1}$) during AMOC_on (d),
AMOC_off (e) and difference between AMOC_off and AMOC_on (f). POC flux at 105m
(mol m$^{-2}$ yr$^{-1}$) during AMOC_on (g), AMOC_off (h) and difference between AMOC_off
and AMOC_on (i).


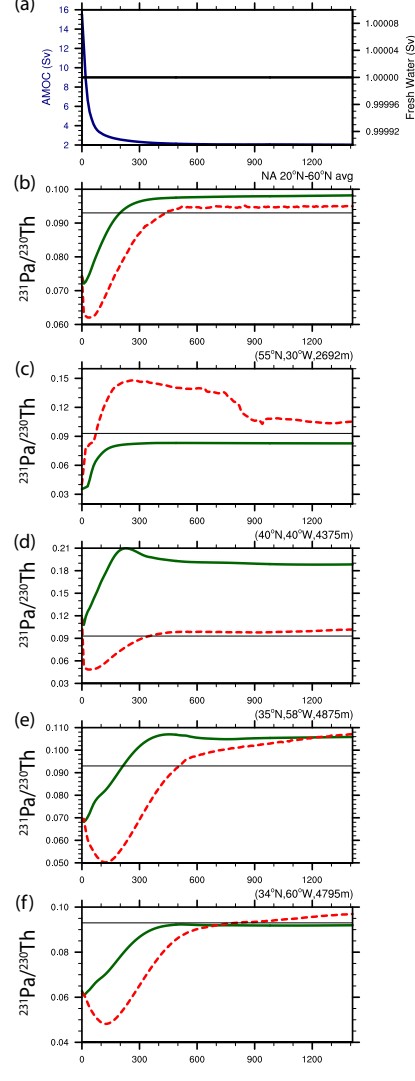



Figure 9. Time evolutions in HOSING. (a) Freshwater forcing (black) and AMOC
strength (navy), which is defined as the maximum of the overturning
streamfunction below 500m in the North Atlantic. (b) North Atlantic average
sediment $^{231}Pa/^{230}Th$ activity ratio from 20°N to 60°N: p-fixed (green) and p-
coupled (red). Production ratio of 0.093 is indicated by a solid black line (similar in
c, d, e and f). (c) Sediment $^{231}Pa/^{230}Th$ activity ratio at (55°N, 30°W). (d) Sediment
$^{231}Pa/^{230}Th$ activity ratio at (40°N, 40°W). (e) Sediment $^{231}Pa/^{230}Th$ activity ratio at
(35°N, 58°W). (f) Sediment $^{231}Pa/^{230}Th$ activity ratio at (34°N, 60°W). (e) and (f) are
near Bermuda Rise. Locations of each site are shown as dots in Fig. 8b.

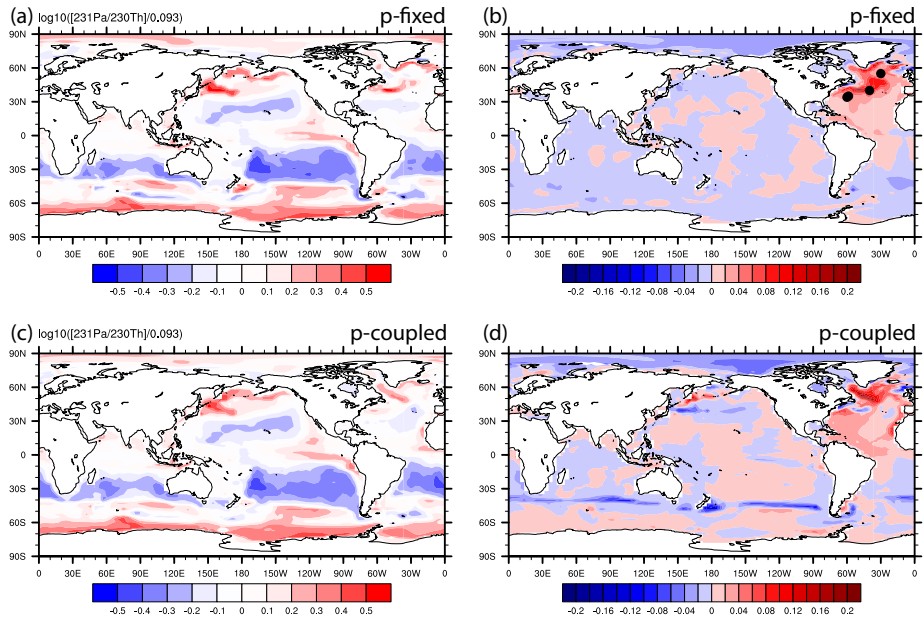



Figure 10. Sediment $^{231}$Pa/$^{230}$Th activity ratio during AMOC off state and the difference between AMOC off and CTRL. (a) P-fixed $\log_{10}([^{231}$Pa/$^{230}$Th]/0.093) in AMOC_off. (b) Difference of p-fixed sediment $^{231}$Pa/$^{230}$Th activity ratio between AMOC_off and AMOC_on. (c) and (d) are similar to (a) and (b) for p-coupled sediment $^{231}$Pa/$^{230}$Th activity ratio. Black dots in (b) shows the locations of sites in Fig. 9 from North to South.


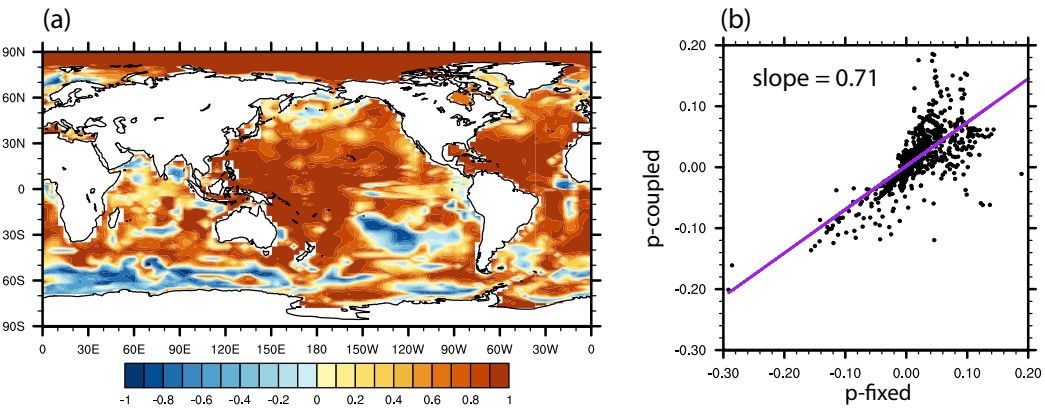


Figure 11. (a) Correlation of p-fixed and p-coupled evolution of sediment
$^{231}$Pa/$^{230}$Th activity ratio in HOSING. (b) Scatter plot of p-fixed and p-coupled
sediment $^{231}$Pa/$^{230}$Th activity ratio change from AMOC_on to AMOC_off in the
Atlantic and the Southern Ocean (70°W-20°E). Purple line is the least squared linear
regression line and slope is the linear regression coefficient.

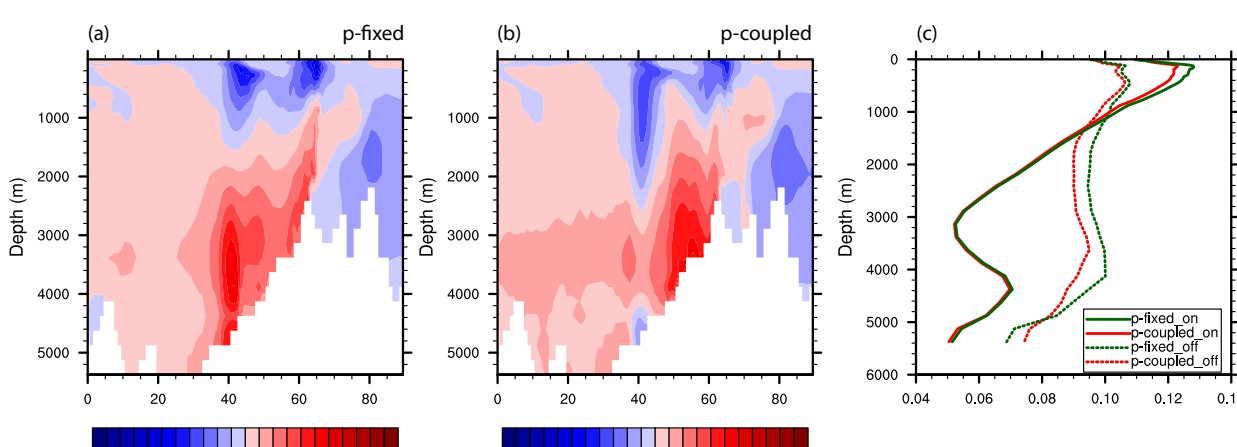


Figure 12. Difference of Atlantic zonal mean particulate $^{231}$Pa/$^{230}$Th between
AMOC_off and AMOC_on: (a) p-fixed and (b) p-coupled. (c) North Atlantic (20°N-
60°N) average profile during AMOC_on (solid) and AMOC_off (dash) for p-fixed
(green) and p-coupled (red) particulate $^{231}$Pa/$^{230}$Th.


