# Peer review of "231Pa and 230Th in the ocean model of the Community Earth System Model"

_Geoscientific Model Development, 2017_

## Short Comment (SC1) · 20 Apr 2017

Dear authors,

I really appreciate that you make your code available in the supplement of the article. Anyhow, the text of the code availability section reads weird, as "Nd isotopes" are never mentioned in your article. Most probably you copied the sentence from another paper and forgot to edit it?

Please correct this upon submission of the revised article

Best regards, Astrid Kerkweg

[Printer-friendly version]{.underline}

[Discussion paper]{.underline}

---

## Short Comment (SC2) · 25 Apr 2017

Dear Editor,

Thanks for pointing out this mistake. We will modify this in our revised version.

Best, Sifan

---

## Referee Comment (RC1) · Anonymous Referee #1 · 26 May 2017

The authors present the implementation of Pa and Th into the community model CESM1.3 in two variants, a biotic and biotic, and compare the control simulations to observations. A classical hosing experiment is carried out and some preliminary analyses are given. The implementation is an important step for later use of this model by the paleoceanographic community, but the description is not sufficiently detailed to be publishable at this stage. Major revisions are recommended.

Comments: 1) The authors seem unaware of the recent paper by Rempfer et al (2017, EPSL) which describes in detail how Pa and Th are implemented in their 3D ocean model. Their description is more comprehensive and complete in the sense that an interested reader has all information available to carry out the model development in another model. This comprehensiveness is also a hallmark of the earlier paper by Siddall et al (2005, EPSL). The paper here, however, does not provide the detail this

reviewer is expecting of a GSMD contribution. The paper needs to take the Rempfer study into consideration and describe carefully in which way the authors' approach is the same, or where it deviates, and why. In the latter case, all parameter values are to be given, as this is a contribution to GSMD (with emphasis on Development which means that a developer can take this paper and create a Pa, Th model component from this information). A the current stage, the paper does not provide this information.

2) Comment 1) does not only apply to the model description only but also to the one example Gu and Liu show, the effect of a collapse of the AMOC on Pa and Th. Rempfer et al (2017) carried out a water hosing experiment and analysed in detail how changes in the Pa/Th ratio inform about circulation changes in the North Atlantic. A critical comparison of the present results with Rempfer et al. is missing.

3) The authors state on line 134 that their implementation is based on Siddall et al (2005). Does this mean that it is identical, i.e. all the parameter values are the same? If not, a Table with the parameter values would be needed for complete information. As stated above, this would be a requirement fro GSMD; too many studies are published nowadays with incomplete information.

4) The text on lines 144ff does some forward referencing to the equations. This should be avoided. First set the context, then introduce the equations and describe every parameter and variable that occurs in these equations. This would ensure easier reading. For example, eq 4 shows many parameters whose values are not given. On line 167 the authors say that eq 4 can be derived from (1) and (2). This is not obvious from the formulations of (1) and (2). Rather eq 4 is a variant of eq 10 of Siddall et al (2005). Again more detail and clarity are needed here.

5) A central point of this paper is the implementation of Pa and Th in abiotic and biotic formulations. In order to appreciate this, more description and analysis should be provided. For example, the prescribed and simulated particle fluxes in different ocean provinces should be shown and compared. It should be quantified how and where

they differ in order to better understand the consequence of these choices for Pa and Th. Given the present level of information in the paper, one can be convinced that the agreement of the two approaches for the control simulation is satisfactory. However, in the transient experiments, differences are rather large depending on the location where the variations are analysed (Fig. 7). Without a more detailed description, the reader is unable to understand the differences. For example, it would be most useful in Fig. 2 below the first row to add panels of the biotic simulation for direct comparison. Implicitly, this information is provided in the scatter plots e)-h), but it would be easier for the reader to see the spatial distribution for the concentrations of the four constituents next to one another and to compare abiotic with biotic this way.

8) The authors follow the approach of Siddall et al (2015) and Rempfer et al (2017) to compare their control simulation with observations. Information is incomplete here as to which data has been used for this comparison. A table in the paper or in the supplementary material summarizing which data has been used would be helpful.

9) Further to 8) reference to the important effort of GEOTRACES is missing. GEO-TRACES offers a wealth of relevant new data. They were used in Rempfer et al (2017) and should also be incorporated into this study for a better and more comprehensive comparison.

10) Information is missing under what conditions Exp_1 and Exp_2 were run. Were these abiotic or biotic simulations? Also, this is not evident in Fig. 5.

11) In Fig. 2b high values of Th_d are noted in the Southern Ocean. This is in contrast to Siddall et al. (2005, their Fig.2) and should be discussed. Is this also occurring in the biotic simulations (see also comment 7. Might the opal fluxes be too high there?

12) Lines 237-240: This statement is not instructive, nor is it very useful. It is noted that the author have performed only one quite simple sensitivity experiment, and this is increasing or decreasing K which changes all partition coefficients simultaneously. This limited perspective does, of course, not shed too much light on this important

question. At least some more thoughts by the authors should be offered here, if not some more pertinent sensitivity tests with their model.

13) Section 4.3. Here, a deeper analysis is required, in particular a comparison with the recent paper of Rempfer et al (2017). They provide an interesting spatial consideration of correlation and Pa/Th-AMOC sensitivity in the North Atlantic Ocean in order to shed light on the controversy whether, and to what extent, Pa/Th changes reflect AMOC changes. The paper here would be able to make an important further contribution to this question, but this opportunity is missed. The authors may argue that this is a paper for GSMD, and hence addressing scientific questions is not the primary purpose. This reviewer might agree with this view if the necessary information for model developers. At this stage, unfortunately, neither is the case.

14) On line 307 the authors argue that the abiotic version captures the major features of the transient simulation. Considering Fig. 7c, d, e, f this statement seems overstating the agreement. Important differences in the transient signal are evident. This should be discussed and explained.

15) From Table 1 it is evident that dust input was not considered in these experiments, although this is not explicitly stated in the text. It would be important to inform the reader why this choice was made, or better, quantify the effect on the Pa and Th concentrations if dust input is included in the simulations.

16) line 383-385. The authors seemed to copy this part from another of their GSMD papers.

17) Throughout the paper, the English should be carefully revisited, in particular in section 4.3. In that section, more paragraphs would ease the reading.

---

## Short Comment (SC3) · 6 Jun 2017

Dear Sifan Gu and Zhengyu Liu,

In the manuscript, you mention that for the purpose of Pa/Th computation (beginning of section 2.2) you need four particles field as follow:

"The implementation of 231Pa and 230Th requires four particle fields: CaCO3, opal, particulate organic carbon (POC) and dust. These particle fields can be obtained from the ecosystem driver from the ecosystem module (Jahn et al., 2015)."

However, Jahn et al., 2015 does not describe anything related to the opal cycle, unless I overlooked it. Hence, though you show the particle flux field in the manuscript, there is no reference to how the opal flux is computed in relation to the Si cycle.

[Figure]

Could you please update this part to clarify how the opal flux is computed?

Best wishes,

Didier Roche

---

## Short Comment (SC4) · 13 Jun 2017

Dear Didier Roche,

Sorry for the confusion. Ocean model of CESM has the biogeochemical component (BGC), which simulates different biological variables. This BGC is described in Moore et al., 2013 and validated against observations in different studies as stated in our manuscript in section 2.2 as "POP2 has incorporated a marine ecosystem module that simulates biological variables (Moore et al., 2013). The marine ecosystem module has been validated against present day observations extensively (Doney et al., 2009; Long et al., 2013; Moore and Braucher, 2008; Moore et al., 2002, 2004)."

For the simulation of Pa and Th cycle, we need to get different particle concentrations

simulated in the BGC module. Jahn et al., 2015 develops an ecosystem driver in CESM which can pass variables simulated in the BGC out. We use this ecosystem driver to get the concentration of particles in the BGC module. This ecosystem driver acts as a media to help different modules in the ocean model to communicate. How the BGC is developed in the CESM can be referred to the references mentioned above.

Hope this helps.

Best,

Sifan Gu

---

## Referee Comment (RC2) · Anonymous Referee #2 · 7 Jul 2017

The paper Âń 231Pa and 230Th in the ocean model of the community Earth system model (CESM1.3)" by S. Gu and Z. Liu is presenting the implementation of 231Pa and 230Th in their general circulation model. It is mainly following the procedure defined by previous work Siddall et al (2005) and Dutay et al (2009). The implementation of the tracers in the model is described and results are compared to observations. However some severe weaknesses are found in the manuscript. The comparison with observation is insufficient, it is strictly following the analysis performed by Siddall et al in 2009, while It now exists , thanks to the GEOTRACES project, new data set. Moreover, the paper do not only show the implementation of the tracer in the model and its validation, which is the scope of the GMD journal, It also propose the response to hosing experiments that is paleoclimate studies that are application that are not

devoted to this journal, Climate of the past would be a more appropriate journal if this study was more correctly analysed. For all these reasons I propose to reject this paper from publication in GMD.

Specific comments: Page 4 section 2.2. The authors show particle flux surface horizontal distribution without concrete comparison with observation. This diagnostic is interesting but it is not sufficient for the proposed study. The model uses particle concentrations and results are strongly dependent to the quality of these fields. It now exist observations to validate the particle fields (Lam et al, 2015) that were not available for Siddall et al (2005) and Dutay et al (2009). A more detailed analysis of the vertical particle concentration distribution at large scale is required. Page 5 section 2.3 Abiotic and Biotic name for simulations are not appropriate. These names suggest that the tracers are subject to different processes while it is not the case. The two approaches are the same except that the particles fields are fixed in the Abiotic run. None biogeochemical process affects the tracer except adsorption and desorption onto particles, so the appellation Biotic run seems exaggerated. Line 162: No validation of particle fields is preformed while it affect strongly the model results. Observations are now available (see for instance lam et al 2015)

Pages 7 and 8 section 4, results Definition and way of estimation of the residence time given for the tracers should be explained.

Comparison of Atlantic zonal averaged model results with observations is no more adequate. It is strictly following analysis performed by Siddall et al (2005) and Dutay et al (2009) a decade ago, but now many new observations are available in the different basins thanks to the GEOTRACES program. This validation is not appropriate any more. Discussion concerning the ratio 231Pa/230Th is very poor. More detailed analysis must be given. For instance what causes low ratio in the north atlantics south of Grennland: convection?

Page10 and 11. This part is already an attempt to use the model development for
scientific question. It is not the purpose of GMD papers. This part should be more deeply analysed and submitted to another more appropriate journal (eg climate of the past)

In conclusion, GMD journal propose to publish model development, and authors can follow procedure previously published with other model. However it can not accept copy of papers published a decade ago and following same analysis while new appropriate observations are now available

---

## Referee Comment (RC3) · Anonymous Referee #3 · 21 Jul 2017

Review of
"$^{231}$Pa and $^{230}$Th in the ocean model of the Community Earth System Model (CESM1.3)" by
Sifan Gu and Zhengyu Liu (MS hereafter).

The aim of this MS is the examination $^{231}$Pa/$^{230}$Th in order to reconstruct past AMOC strengths. The $^{231}$Pa/$^{230}$Th method is somewhat under debate due to the fact that (at least) two competing factors control sedimentary $^{231}$Pa/$^{230}$Th: AMOC and particle fluxes. Although the AMOC seems to be of first order control on $^{231}$Pa/$^{230}$Th the question is still valid when and where particle fluxes may take over control or may at least influence the circulation signal significantly. From this point of view I welcome very much this new attempt of deciphering the behaviour of $^{231}$Pa/$^{230}$Th in the ocean and to implement both key isotopes in a climate model.

The authors build up on previous work by (Siddall et al., 2005). They have improved the CESM by implementing a coupled biogeochemical module. The output of the model is compared to observations in order to "reality-check" the model and the chosen parameters. Further they compare the output of the coupled marine ecosystem model with a prescribed particle field in order to gain knowledge about the sensitivity of the applied particle flux forcing.

The main point of criticism I have here is their comparison to observational data, which I find is too nebulous and not supported by newer data. There is an obvious lack of consideration of recent papers. More recent studies would provide a much better basis for comparison and reality-checks of the model. The references for the observational data given in the MS are quite old holding mostly data obtained by the noisy counting-method resulting in large analytical uncertainties. Instead the model should be cross-checked with newer sedimentary and water column data. I don't see much benefit from comparing "biotic" against "abiotic" $^{231}$Pa and $^{230}$Th particle-fluxes (Fig. 2), as long as the absolute values have not been tested against new observational data. The authors urgently need to test the output of the model versus recent sedimentary data (e.g. (Böhm et al., 2015; Bradtmiller et al., 2014; Burckel et al., 2016; Henry et al., 2016; Hoffmann et al., 2013; Jonkers et al., 2015; Lippold et al., 2011; Lippold et al., 2016; Lippold et al., 2012; Luo et al., 2015; Negre et al., 2010; Roberts et al., 2014; Rutgers van der Loeff et al., 2016)),
water data (e.g. (Deng et al., 2014; Hayes et al., 2014; Hayes et al., 2013; Hayes et al., 2015a; Hayes et al., 2015b; Kretschmer et al., 2011))
and most importantly other modelling studies (e.g. (Dutay et al., 2015; Lippold et al., 2011; Rempfer et al., 2017)).
I find the terms "biotic $^{231}$Pa/$^{230}$Th" and "abiotic $^{231}$Pa/$^{230}$Th" quite confusing. Since there is no biotic $^{231}$Pa and $^{230}$Th these terms should be used only to distinguish between the usage of particle fields in the model.
Given that (Rempfer et al., 2017) recently provided insights into an upgraded approach by (Siddall et al., 2005) and (Siddall et al., 2007), including a bio-geochemical-module in the model, I do not see much advance provided by the here presented MS. I did not find a reference to (Rempfer et al., 2017), maybe because this is a very recent publication, but I don't think the authors should neglect this paper in a new version.
Although I welcome very much the provision of the Fortran code the reader is left alone with the comparison between model and observations (Fig.3) without sufficient information about the values, observational error bars and references. The color code in Fig. 3 may hold some information about the water depths, but since (already) older publications demandingly have shown, that the correlation of $^{231}$Pa/$^{230}$Th with water-depth seems to be a manifested pattern of AMOC in the $^{231}$Pa/$^{230}$Th distribution (Burckel et al., 2016; Gherardi et al., 2009; Gherardi et

al., 2010; Hoffmann et al., 2013; Luo et al., 2010; Luo et al., 2015) this feature is required to be reproduced by a meaningful model. But I'm not able to see this from the provided figures.

By the way, the diagrams are way too detailed (in terms of graphic resolution) demanding a lot of computer resources and slowing down even my reasonably new computer just by scrolling down.

The table for the K values (Table 1) needs to be accompanied by references, because these values vary within a wide range according to the studies by (Chase et al., 2002, 2004; Hayes et al., 2013; Hayes et al., 2015b; Kretschmer et al., 2011; Kretschmer et al., 2008; Luo et al., 1999, 2003, 2004) and others. I think, a well selected digest of values can be found at the new study by (Rempfer et al., 2017).

Besides the shortcomings of the MS regarding the observational data, I also find patterns in the model output, which are not observed in reality to my knowledge. E.g. the appearance of a high opal/POC field in the NW-Atlantic. Further, I see an obvious mismatch of model and observations in Fig. 5, which is not explained.

In summary, it is hard for me to see that the here presented model approach provides any new insights on the $^{231}Pa/^{230}Th$ method. Due to the lack of information about the model-data comparison it is not possible to assess the quality of the model and the applied parameters. Consequently I suggest revising both the model runs and the MS thoroughly before publication can be considered.

References:

Böhm, E., et al., **2015**. Strong and deep Atlantic Meridional Overturning Circulation during the last glacial cycle. Nature 517.

Bradtmiller, L., et al., **2014**. $^{231}Pa/^{230}Th$ evidence for a weakened but persistent Atlantic meridional overturning circulation during Heinrich Stadial 1. Nature Communications 5.

Burckel, P., et al., **2016**. Changes in the geometry and strength of the Atlantic Meridional Overturning Circulation during the last glacial (20-50 ka). Climate of the Past 12.

Chase, Z., et al., **2002**. The influence of particle composition and particle flux on scavenging of Th, Pa and Be in the ocean. Earth and Planetary Science Letters 204.

Chase, Z., et al., **2004**. Comment on "On the importance of opal, carbonate and lithogenic clays in scavenging and fractionating $^{230}Th$ $^{231}Pa$ and $^{10}Be$ in the ocean". Earth and Planetary Science Letters 220.

Deng, F., et al., **2014**. Controls on seawater $^{231}Pa$, $^{230}Th$ and $^{232}Th$ concentrations along the flow paths of deep waters in the Southwest Atlantic. Earth and Planetary Science Letters 390.

Dutay, J.C., et al., **2015**. Modelling the role of marine particle on large scale $^{231}Pa$, $^{230}Th$, Iron and Aluminium distributions. Progress in Oceanography 133.

Gherardi, J., et al., **2009**. Glacial-interglacial circulation changes inferred from $^{231}Pa/^{230}Th$ sedimentary record in the North Atlantic region. Paleoceanography 24.

Gherardi, J., et al., **2010**. Reply to comment by S. Peacock on "Glacial-interglacial circulation changes inferred from $^{231}Pa/^{230}Th$ sedimentary record in the North Atlantic region". Paleoceanography 25.

Hayes, C., et al., **2014**. Biogeography in $^{231}Pa/^{230}Th$ ratios and a balanced $^{231}Pa$ budget for the Pacific Ocean. Earth and Planetary Science Letters 391.

Hayes, C., et al., **2013**. A new perspective on boundary scavenging in the North Pacific Ocean. Earth and Planetary Science Letters 369–370.

Hayes, C., et al., **2015a**. $^{230}Th$ and $^{231}Pa$ on GEOTRACES GA03, the U.S. GEOTRACES North Atlantic transect, and implications for modern and paleoceanographic chemical fluxes. Deep Sea Research Part II: Topical Studies in Oceanography 116.

Hayes, C., et al., **2015b**. Intensity of Th and Pa scavenging partitioned by particle chemistry in the North Atlantic Ocean. Marine Chemistry 170.

Henry, L.G., et al., **2016**. North Atlantic ocean circulation and abrupt climate change during the last glaciation. Science 353.

Hoffmann, S., et al., **2013**. Persistent export of $^{231}$Pa from the deep central Arctic Ocean over the past 35,000 years. Nature 497.

Jonkers, L., et al., **2015**. Deep circulation changes in the central South Atlantic during the past 145 kyrs reflected in a combined $^{231}$Pa/$^{230}$Th, Neodymium isotope and benthic record. Earth and Planetary Science Letters 419.

Kretschmer, S., et al., **2011**. Fractionation of $^{230}$Th, $^{231}$Pa, and $^{10}$Be induced by particle size and composition within an opal-rich sediment of the Atlantic Southern Ocean. Geochimica et Cosmochimica Acta 75.

Kretschmer, S., et al., **2008**. Distribution of $^{230}$Th, $^{10}$Be and $^{231}$Pa in Sediment Particle Classes. Geochimica et Cosmochimica Acta 72.

Lippold, J., et al., **2011**. Testing the $^{231}$Pa/$^{230}$Th paleocirculation proxy - A data versus 2D model comparison. Geophysical Research Letters 38.

Lippold, J., et al., **2016**. Deep water provenance and dynamics of the (de)glacial Atlantic meridional overturning circulation. Earth and Planetary Science Letters 445.

Lippold, J., et al., **2012**. Strength and geometry of the glacial Atlantic Meridional Overturning Circulation. Nature Geoscience 5.

Luo, S., et al., **1999**. Oceanic $^{231}$Pa/$^{230}$Th ratio influenced by particle composition and reminmeralization. Earth and Planetary Science Letters 167.

Luo, S., et al., **2003**. On the importance of opal, carbonate and lithogenic clays in scavenging and fractionating $^{230}$Th $^{231}$Pa and $^{10}$Be in the ocean. Earth and Planetary Science Letters 220.

Luo, S., et al., **2004**. Reply to Comment on ''On the importance of opal, carbonate, and lithogenic clays in scavenging and fractionating $^{230}$Th, $^{231}$Pa and $^{10}$Be in the ocean''. Earth and Planetary Science Letters 220.

Luo, Y., et al., **2010**. Sediment $^{231}$Pa/$^{230}$Th as a recorder of the rate of the Atlantic meridional overturning circulation: insights from a 2-D model. Ocean Science 6.

Luo, Y., et al., **2015**. Controls on $^{231}$Pa and $^{230}$Th in the Arctic Ocean. Geophysical Research Letters 42.

Negre, C., et al., **2010**. Reversed flow of Atlantic deepwater during the Last Glacial Maximum. Nature 468.

Rempfer, J., et al., **2017**. New insights into cycling of 231Pa and 230Th in the Atlantic Ocean. Earth and Planetary Science Letters 468.

Roberts, N., et al., **2014**. Advection and scavenging controls of Pa/Th in the northern NE Atlantic. Paleoceanography 29.

Rutgers van der Loeff, M., et al., **2016**. Meridional circulation across the Antarctic Circumpolar Current serves as a double $^{231}$Pa and $^{230}$Th trap. Earth and Planetary Science Letters.

Siddall, M., et al., **2005**. $^{231}$Pa/$^{230}$Th fractionation by ocean transport, biogenic particle flux and particle type. Earth and Planetary Science Letters 237.

Siddall, M., et al., **2007**. Modelling the relationship between $^{231}$Pa/$^{230}$Th distribution in North Atlantic sediment and Atlantic meridional overturning circulation. Paleoceanography 22.

---

## Author Comment (AC1) · 11 Aug 2017

We thank the reviewer for his/her time for constructing the comments.

In the following, we have addressed all comments.

"Comments: 1) The authors seem unaware of the recent paper by Rempfer et al (2017, EPSL) which describes in detail how Pa and Th are implemented in their 3D ocean model. Their description is more comprehensive and complete in the sense that an interested reader has all information available to carry out the model development in another model. This comprehensiveness is also a hallmark of the earlier paper by Siddall et al (2005, EPSL). The paper here, however, does not provide the detail this reviewer is expecting of a GSMD contribution. The paper needs to take the Rempfer

study into consideration and describe carefully in which way the authors' approach is the same, or where it deviates, and why. In the latter case, all parameter values are to be given, as this is a contribution to GSMD (with emphasis on Development which means that a developer can take this paper and create a Pa, Th model component from this information). A the current stage, the paper does not provide this information."

Thanks for pointing out the paper by Rempfer et al., (2017). We have made substantial changes describing how Pa and Th are implemented in our model and the difference between Rempfer et al., (2017). We add a short review on previous modeling efforts (Line 80-89) and the similarity and difference between our method and previous studies (Line 200-207). Eq. (3) is the conservation equation for Pa and Th, which is how Pa and Th calculated in the model. The calculations of Pa and Th are based on this equation. In section 2.3, we explicitly describe each term in Eq. (3) and the values of different parameters, which includes all the information for reproduce the model development in another model. Also, we add Table 1 and Table 2 to show the abbreviation and values of different parameters used in text.

"2) Comment 1) does not only apply to the model description only but also to the one example Gu and Liu show, the effect of a collapse of the AMOC on Pa and Th. Rempfer et al (2017) carried out a water hosing experiment and analysed in detail how changes in the Pa/Th ratio inform about circulation changes in the North Atlantic. A critical comparison of the present results with Rempfer et al. is missing."

In the revised version, we compare our results with Rempfer et al. 2017. We get similar particulate Pa/Th response in North Atlantic (their Fig. 8 and our Fig. 12) and we add a discussion in the text (Line 423-444).

"3) The authors state on line 134 that their implementation is based on Siddall et al (2005). Does this mean that it is identical, i.e. all the parameter values are the same? If not, a Table with the parameter values would be needed for complete information. As stated above, this would be a requirement fro GSMD; too many studies are published

nowadays with incomplete information."

Sorry we did not make this clear enough. Yes, the parameters in the implementation is the same as Siddall et al., (2005). Values of different parameters are given in the text when it first appears. To make it clearer, we add a table to summarize the parameters and variable in Table 1 and Table 2.

"4) The text on lines 144ff does some forward referencing to the equations. This should be avoided. First set the context, then introduce the equations and describe every parameter and variable that occurs in these equations. This would ensure easier reading. For example, eq 4 shows many parameters whose values are not given. On line 167 the authors say that eq 4 can be derived from (1) and (2). This is not obvious from the formulations of (1) and (2). Rather eq 4 is a variant of eq 10 of Siddall et al (2005). Again more detail and clarity are needed here."

Thanks for pointing out the problems in the equations. We have rearranged the context and the equations as suggested. We explicitly show how can be calculated from Eq. 1 and 2 (Eq. 4, 5 and 6).

"5) A central point of this paper is the implementation of Pa and Th in abiotic and biotic formulations. In order to appreciate this, more description and analysis should be provided. For example, the prescribed and simulated particle fluxes in different ocean provinces should be shown and compared. It should be quantified how and where they differ in order to better understand the consequence of these choices for Pa and Th. Given the present level of information in the paper, one can be convinced that the agreement of the two approaches for the control simulation is satisfactory. However, in the transient experiments, differences are rather large depending on the location where the variations are analysed (Fig. 7). Without a more detailed description, the reader is unable to understand the differences. For example, it would be most useful in Fig. 2 below the first row to add panels of the biotic simulation for direct comparison. Implicitly, this information is provided in the scatter plots e)-h), but it would be easier for

the reader to see the spatial distribution for the concentrations of the four constituents next to one another and to compare abiotic with biotic this way."

First of all, the term abiotic and biotic seems to be not appropriate since Pa and Th are not actually involved in the biological activity. We change the term to "p-fixed" and "p-coupled" which clearly indicates the difference between two versions.

Thanks for suggesting ways to show that the p-fixed and p-coupled versions give almost identical results in CTRL. We follow this advice and show directly the results of these two versions in Fig. 2, 3 and 4. Clearly, readers can see that p-fixed and p-coupled are similar in CTRL.

For the HOSING experiment, we also add Fig. 8 to show the differences in particle production during AMOC_on and AMOC_off, which will help the discussion about the p-fixed and p-coupled Pa/Th differences in HOSING.

"8) The authors follow the approach of Siddall et al (2015) and Rempfer et al (2017) to compare their control simulation with observations. Information is incomplete here as to which data has been used for this comparison. A table in the paper or in the supplementary material summarizing which data has been used would be helpful."

Thanks for suggesting to use a table to show the data used. We have added Table 3 to show the references used in model data comparison. Most of the data are also used in Rempfer et al., (2017).

"9) Further to 8) reference to the important effort of GEOTRACES is missing. GEO-TRACES offers a wealth of relevant new data. They were used in Rempfer et al (2017) and should also be incorporated into this study for a better and more comprehensive comparison."

Thanks for pointing out available GEOTRACES data. We have included those in the observation (Table 3). We show model results along two GEOTRACES transects as in Rempfer et al., (2017) (Fig. 2 and 3) for direct comparision. Our results are similar as

the case Re3d in Rempfer et al. (2017), which does not include boundary scavenging and sediment resuspensions.

"10) Information is missing under what conditions Exp_1 and Exp_2 were run. Were these abiotic or biotic simulations? Also, this is not evident in Fig. 5."

Sensitivity experiment Exp 1 and Exp 2 are abiotic simulations for computational efficiency (Line 212-213). Exp1 and Exp2 are carried under the same forcing as CTRL (Line 221-222).

"11) In Fig. 2b high values of Th_d are noted in the Southern Ocean. This is in contrast to Siddall et al. (2005, their Fig.2) and should be discussed. Is this also occurring in the biotic simulations (see also comment 7. Might the opal fluxes be too high there?"

Since the GEOTRACE transects are more appropriate for model data comparison, we replace the zonal mean figure with the GEOTRACE transects and move the zonal mean figure to the supplementary information (Fig. S3). The high values of Th_d in the Southern Ocean around 60S in the model is consistent observations (Fig. S3b) since observations of Th_d from 60S-55S are much larger than Th_d from 55S-40S. In addition, our model is in much higher resolution than Siddall et al., (2005). The maximum Th_d locates at around 60S, decreasing if further southward in our model. Similar pattern also appears in Siddall et al., (2005). Their Th_d maximum is at around 55S, decreasing southward (but only two grid available in their model).

"12) Lines 237-240: This statement is not instructive, nor is it very useful. It is noted that the author have performed only one quite simple sensitivity experiment, and this is increasing or decreasing K which changes all partition coefficients simultaneously. This limited perspective does, of course, not shed too much light on this important question. At least some more thoughts by the authors should be offered here, if not some more pertinent sensitivity tests with their model."

Thanks for pointing this out. We have removed this part. The poor performance in

simulating particulate Pa and Th is also in Siddall et al., (2005) and Dutay et al., (2009). Rempfer et al., (2017) only shows Pa_p/Th_p and does not show individual Pa_p and Th_p. It's possible the performace is limited by our choice of modeling scheme since the process in controlling Pa and Th activities are essentially the same among our study, Siddall et al., (2005) and Dutay et al., (2009). Although individual Pa_p and Th_p do not agree well with the observations, the ratio of Pa_p/Th_p in our CTRL experiment show similar results as in Rempfer et al. (2017) and sediment Pa_p/Th_p distribution agrees with available observations. And the ratio of Pa/Th is what we are interested in.

"13) Section 4.3. Here, a deeper analysis is required, in particular a comparison with the recent paper of Rempfer et al (2017). They provide an interesting spatial consideration of correlation and Pa/Th-AMOC sensitivity in the North Atlantic Ocean in order to shed light on the controversy whether, and to what extent, Pa/Th changes reflect AMOC changes. The paper here would be able to make an important further contribution to this question, but this opportunity is missed. The authors may argue that this is a paper for GSMD, and hence addressing scientific questions is not the primary purpose. This reviewer might agree with this view if the necessary information for model developers. At this stage, unfortunately, neither is the case."

Thanks for pointing out the interesting spatial dependence behavior of Pa/Th in the hosing experiment in Rempfer et al., 2017. Our model, with much higher resolution, shows similar spatial pattern as theirs (Fig. 12). We add discussion of this spatial dependence in Line 423-444. This spatial dependence is mainly caused by AMOC, since the pattern in p-fixed and p-coupled are similar.

"14) On line 307 the authors argue that the abiotic version captures the major features of the transient simulation. Considering Fig. 7c, d, e, f this statement seems overstating the agreement. Important differences in the transient signal are evident. This should be discussed and explained."

We agree that there are many differences between p-fixed and p-coupled response to freshwater forcing. If we compare Fig. 10 b and d, in North Atlantic, the sediment Pa/Th overall show increase in both p-fixed and p-coupled (except opal maximum region). In Fig. 9, the transient evolution figure, if we neglect the initial drop in p-coupled (red), the long-term trend between p-coupled and p-fixed are the same. Therefore, over low productivity and long time scale, the p-fixed capture the major features of sediment Pa/Th change and suggest that AMOC change is dominant. But on short time scale and over high productivity region, p-coupled response behaves quite differently from p-fixed. We discuss the differences in the revised manuscript (Line 381-420).

"15) From Table 1 it is evident that dust input was not considered in these experiments, although this is not explicitly stated in the text. It would be important to inform the reader why this choice was made, or better, quantify the effect on the Pa and Th concentrations if dust input is included in the simulations."

Thanks for the suggestions. Dust is not included in the calculation. We use the parameters used in the control experiment in Siddall et al., (2005), which the partition coefficient for dust is 0. They also did sensitivity experiment and find dust flux is unimportant for Pa/Th fractionation. We have modified Fig.1, Table 1 and text (line135-137) accordingly.

"16) line 383-385. The authors seemed to copy this part from another of their GSMD papers."

Sorry for the mistake in the code availability part. We have fixed the error.

"17) Throughout the paper, the English should be carefully revisited, in particular in section 4.3. In that section, more paragraphs would ease the reading."

Follow this suggestion, we re-write section 4.3. In the revised version, we first discuss the p-fixed sediment Pa/Th response in the Atlantic, which generally increase during AMOC_off (Line 339-358) and the magnitude of increase is related to particle

distribution (Line 359-370). Then we discuss the p-coupled response. The change in sediment Pa/Th between AMOC_on and AMOC_off in p-coupled are similar to p-fixed in most North Atlantic (Line 371-380), but there are differences especially on short time scale and over high productivity region (Line 381-420). At last, we discuss the change in particulate Pa/Th in North Atlantic and show the depth dependence of the change (Line 423-444).

Please also note the supplement to this comment:
https://www.geosci-model-dev-discuss.net/gmd-2017-82/gmd-2017-82-AC1-supplement.pdf

**Supplement:**

**1** Coherent response of Antarctic Intermediate Water and Atlantic Meridional

Overturning Circulation during the last deglaciation: reconciling contrasting
 neodymium isotope reconstructions from the tropical Atlantic

4

**5 Sifan Gua,\*, Zhengyu Liua,\*, Jiaxu Zhanga, Johannes Rempferb, Fortunat Joosb,**

- 6 Delia W. Oppoc
- 7 aDepartment of Atmospheric and Oceanic Sciences and Center for Climatic Research,
- 8 University of Wisconsin-Madison, Madison, WI, USA;
- 9 bClimate and Environmental Physics, Physics Institute and Oeschger Center for Climate
- 10 Change Research, University of Bern, Bern, Switzerland
- cDepartment of Geology and Geophysics, Woods Hole Oceanographic Institution, Woods
   Hole, MA, USA
- 13
- 14 Corresponding author: Sifan Gu (sgu28@wisc.edu) and Zhengyu Liu (zliu3@wisc.edu)

**15 Key Points:**

- Antarctic Intermediate Water northward penetration is controlled by the Atlantic
   Meridional Overturning Circulation strength.
- Atlantic Intermediate Water becomes deeper and thicker during weaker Atlantic
   Meridional Overturning Circulation period.
- The contradictory  $\varepsilon_{Nd}$  reconstructions from the tropical Atlantic are due to the site location and depth and the influence of different water masses.

**22 Abstract**

Antarctic Intermediate Water (AAIW) plays important roles in the global climate system 23 and the global ocean nutrient and carbon cycles. However, it is unclear how AAIW 24 25 responds to global climate changes. In particular, neodymium isotopic composition ( $\varepsilon_{Nd}$ ) reconstructions from different locations from the tropical Atlantic, have led to a debate on 26 the relationship between northward penetration of AAIW into the tropical Atlantic and 27 the Atlantic Meridional Overturning Circulation (AMOC) variability during the last 28 deglaciation. We resolve this controversy by studying the transient oceanic evolution 29 during the last deglaciation using a neodymium-enabled ocean model. Our results suggest 30 a coherent response of AAIW and AMOC: when AMOC weakens, the northward 31 penetration and transport of AAIW decreases while its depth and thickness increase. Our 32 33 study highlights that as part of the return flow of the North Atlantic Deep Water (NADW), the northward penetration of AAIW in the Atlantic is determined 34 predominately by AMOC intensity. Moreover, the inconsistency among different tropical 35 Atlantic  $\varepsilon_{Nd}$  reconstructions is reconciled by considering their corresponding core 36 locations and depths, which were influenced by different water masses and ocean 37 currents in the past. The very radiogenic water from the bottom of the Gulf of Mexico 38 and the Caribbean Sea, which was previously overlooked in the interpretations of 39 deglacial  $\varepsilon_{Nd}$  variability, can be transported to shallow layers during active AMOC, and 40 modulates  $\varepsilon_{Nd}$  in the tropical Atlantic. Changes in the AAIW core depth must also be 41 considered. Thus, interpretation of  $\varepsilon_{Nd}$  reconstructions from the tropical Atlantic is more 42 complicated suggested previous studies. 43 than in

**44 **1 Introduction**

Antarctic Intermediate Water (AAIW) is a key component of the global ocean 45 circulation. Large volume northward flowing AAIW plays an important role in the northward 46 nutrient transport to sustain primary production in the North Atlantic [Sarmiento et al., 2004; 47 *Palter and Lozier*, 2008]. It also contributes to the anthropogenic carbon sink [*Sabine*, 2004; 48 Gruber et al., 2009] and the ocean acidification [Ito et al., 2010; Resplandy et al., 2013]. 49 However, how AAIW responds to global climate changes has remained poorly understood. In 50 particular, how AAIW interacts with the Atlantic Meridional Overturning Circulation (AMOC) 51 52 remains highly controversial. The last deglaciation presents an ideal target to test our understanding of the relation between AAIW and AMOC. Some previous observational studies 53 of the last deglaciation suggested that the northward penetration of AAIW in the tropical Atlantic 54 should be positively correlated with the AMOC strength [*Came et al.*, 2008; Xie et al., 2012; 55 *Huang et al.*, 2014]. This positive correlation seems to be consistent with the notion that AAIW. 56 as part of the North Brazil Current (NBC), contributes to the return branch of North Atlantic 57 Deep Water (NADW) [Rintoul, 1991; Schmitz and McCartney, 1993; Lumpkin and Speer, 2003; 58 Zhang et al., 2011]. However, other observational studies infer an enhanced AAIW penetration 59 into the tropical Atlantic with a collapsed AMOC during the last deglaciation, or a negative 60 correlation between the AAIW penetration and AMOC intensity [Zahn and Stüber, 2002; 61 Rickaby and Elderfield, 2005; Pahnke et al., 2008]. This negative correlation appears to be 62 consistent with some other modeling studies, which simulate an increased AAIW transport into 63 the North Atlantic in a counterclockwise shallow AAIW cell after the initial collapse of AMOC 64 [Saenko et al., 2003; Weaver et al., 2003; Stouffer et al., 2007]. The different relationship 65 between AAIW northward penetration in the Atlantic and the AMOC strength suggests different 66

roles of AAIW in AMOC: a positive correlation implies the AAIW penetration as a subsequent 67 response to the AMOC reorganization while a negative correlation indicates that the AAIW 68 penetration may provide a positive feedback or a trigger for AMOC reorganization as more fresh 69 water is transported to the North Atlantic by AAIW when AMOC is weaker [Pahnke et al., 70 2008]. In addition, understanding the relationship between the AAIW northward penetration in 71 72 the Atlantic and AMOC also helps to understand the mechanisms of nutrient supply change in low latitude Atlantic across the deglaciation, which is also under debate [Meckler et al., 2013; 73 Hendry et al., 2016]. 74

In studying the AAIW evolution during the last deglaciation, we will pay particular 75 attention to neodymium (Nd) isotopic composition ( $\varepsilon_{Nd}$ ), which has emerged as a promising 76 quasi-conservative tracer for water masses [Goldstein and Hemming, 2003]. ENd is defined as 77  $[(^{143}Nd/^{144}Nd)_{sample}/(^{143}Nd/^{144}Nd)_{CHUR} - 1]*10^4$ , where  $(^{143}Nd/^{144}Nd)_{CHUR}$  is 0.512638, which is 78 the bulk earth composition defined by the Chondritic Uniform Reservoir [Jacobsen and 79 *Wasserburg*, 1980]. The  $\varepsilon_{Nd}$  exhibits distinct values geographically, with the most radiogenic 80 (highest) values in the North Pacific  $(0 \sim -5)$ , intermediate values in the Southern Ocean and the 81 Indian Ocean  $(-7 \sim -10)$  and the least radiogenic (lowest) values in the North Atlantic  $(-10 \sim -14)$ . 82 This strong  $\epsilon_{Nd}$  gradient has motivated using  $\epsilon_{Nd}$  as a tracer for Northern versus Southern water 83 mass mixing. Unlike tracers such as  $\delta^{13}$ C and Cd/Ca, which are highly influenced by biological 84 processes in addition to ocean circulation, biological or chemical fractionation of  $\varepsilon_{Nd}$  is 85 negligible [Goldstein and Hemming, 2003]. Furthermore,  $\varepsilon_{Nd}$  is relatively insensitive to potential 86 87 Nd source changes as unrealistically extreme changes in Nd sources are required in the model to produce the magnitude of  $\varepsilon_{Nd}$  changes comparable to reconstructions [*Rempfer et al.*, 2012b]. 88 89 Variations of  $\varepsilon_{Nd}$  is able to reflect the strength of overturning circulation in idealized fresh water hosing experiments [*Rempfer et al.*, 2012a]. Therefore,  $\varepsilon_{Nd}$  appears to be an effective tracer for water masses and has been increasingly used in paleoceanographic studies.

In the tropical Atlantic, a more radiogenic  $\varepsilon_{Nd}$  at the AAIW depth would imply a stronger 92 AAIW influence (from the Southern Ocean) with an enhanced AAIW northward penetration, and 93 vice versa, if the end-member  $\varepsilon_{Nd}$  values are stable. Although the North Atlantic water mass  $\varepsilon_{Nd}$ 94 end-member is complicated by NADW source waters, which are distinct in  $\varepsilon_{Nd}$  [van de Flierdt et 95 al., 2016], end-member  $\varepsilon_{Nd}$  of northern-sourced water is suggested to be stable on glacial-96 interglaical to millennial timescales [van de Flierdt et al., 2006; Foster et al., 2007]. ENd from the 97 southern Brazil margin at intermediate depth also shows no changes across the last deglaciation 98 [Howe et al., 2016]. Furthermore, a modeling study [Rempfer et al., 2012a] suggests that effect 99 of end-member  $\varepsilon_{Nd}$  changes are much smaller than the effect of changes in water mass 100 distribution on the millennial time scale. 101

The controversy on the relationship between the AMOC intensity and the northward extent of AAIW arises in part from  $\varepsilon_{Nd}$  reconstructions at intermediate depths from the tropical Atlantic, which show two opposite evolution behaviors: from the Last Glacial Maximum (LGM, 22 kyr Before Present, B.P.) to the Heinrich Stadial 1 (HS1, 17.5-14.7 kyr B.P.),  $\varepsilon_{Nd}$  decreases (becomes less radiogenic) in some cores [*Xie et al.*, 2012; *Huang et al.*, 2014], but increases (becomes more radiogenic) in some others [*Pahnke et al.*, 2008]. Understanding these opposite responses is critical for understanding the response of AAIW to deglacial AMOC variability.

To better understand the evolution of AAIW and the opposite  $\varepsilon_{Nd}$  changes in different tropical Atlantic records, we performed a transient ocean simulation for the last deglaciation (iPOP2-TRACE) [*Zhang*, 2016] under realistic climate forcings using a Nd-enabled ocean model. We find that the AAIW northward penetration in the tropical Atlantic is dominated by

AMOC strength but interpreting  $\varepsilon_{Nd}$  reconstructions is not as simple as suggested in previous 113 studies because both the AAIW core depth and the influence of radiogenic bottom water from 114 115 the Gulf of Mexico and the Caribbean Sea respond to variations in AMOC strength, influencing  $\varepsilon_{Nd}$  values in the tropical Atlantic. We describe the Nd implementation and experiments in 116 section 2. We examine the deglacial AAIW evolution in our simulation and the associated 117 physical mechanism in section 3. Section 4 discusses how the inconsistency in  $\varepsilon_{Nd}$ 118 reconstructions can be understood in terms of the different depth and influence of the radiogenic 119 water from the Gulf of Mexico and the Caribbean Sea. Finally, we summarize our findings in 120 section 5. 121

122

**123 **2 Methods**

**124 **2.1 Nd implementation**

The Nd module is implemented in the ocean model (POP2) of Community Earth System 125 Model (CESM) [Hurrell et al., 2013] following Rempfer et al., [2011]. Nd has three sources: 126 riverine input, dust deposition and boundary source from continental margins. Dust and river 127 sources enter the ocean at the surface ocean while the boundary source enters through the 128 continental margins above 3,000m. Dust flux is prescribed using a model composite from 129 Mahowald et al., [2005]. We use global mean Nd concentration of 20 ug/g in the dust [Goldstein] 130 et al., 1984; Grousset et al., 1988, 1998] and 2% of which is released into the ocean [Greaves et 131 al., 1994]. River discharge is taken from the coupler of the model instead of being prescribed as 132 in *Rempfer et al.*, [2011]. Nd concentration in river discharge is prescribed following *Goldstein* 133 and Jacobsen, [1987] and 70% of the dissolved Nd in rivers is removed in estuaries [Goldstein 134 and Jacobsen, 1987]. Nd flux from the continental margins is assumed to be a globally uniform 135

value and we use  $5.5 \times 10^9$  g/yr for the global total Nd source from the continental margins [*Rempfer et al.*, 2011]. 143Nd and 144Nd are simulated separately as two passive tracers and the fluxes for individual 143Nd and 144Nd are obtained by using prescribed isotopic ratio (IR = 143Nd/144Nd): IRdust is prescribed following *Tachikawa et al.*, [2003] and IRriver and IRboundary are prescribed following *Jeandel et al.*, [2007].

The sink of Nd in the ocean is the reversible scavenging process. It describes the 141 adsorption of Nd onto particles (particulate organic carbon (POC), opal, calcium carbonate 142 (CaCO3) and dust), settling downward along with these particles and the desorption from 143 particles due to particle dissolution. In the bottom layer in the water column, if particles still 144 exist, the Nd associated to these particles will be removed from the ocean. The balance between 145 the dissolved Nd ( $[Nd]_d$ ) and the particle related Nd ( $[Nd]_p$ ) is described by equilibrium 146 scavenging coefficient which is also prescribed following *Rempfer et al.*, [2011]. Therefore, the 147 conservation equation for 143Nd and 144Nd is as follows: 148

149
$$\frac{\partial [Nd]_t^j}{\partial t} = S_{tot} - \frac{\partial (v \cdot [Nd]_p^j)}{\partial z} + T([Nd]_t^j)(j = 143, 144)$$

The three terms on the right-hand side represent the total sources, the reversible scavenging, and the ocean transport, respectively. The settling velocity of particles, v, is chosen as 1000 m/yr as in *Rempfer et al.*, [2011]. Detailed description and parameterization are given in *Rempfer et al.*, [2011]. Our Nd module is not coupled with a marine biogeochemical model. We use export production of POC, opal and CaCO3 from the biogeochemical component from Bern3D model and prescribe the remineralization profile following Rempfer et al. [2011]. Overall, our Nd concentration and  $\varepsilon_{Nd}$  capture the major features in the observations (in Section 2.2).

**157 **2.2 Nd module validation**

Our Nd-enabled CESM can simulate the global distribution of both Nd concentration and 158  $\varepsilon_{Nd}$  reasonably well under present day climate forcing. We first run a present day control 159 experiment (CTRL) forced by 1948-2007 atmospheric data from Coordinated Ocean-ice 160 Reference Experiments [Large and Yeager, 2008]. Nd concentrations (both 143Nd and 144Nd) 161 were initialized from zero. CTRL has been integrated for more than 4,000 model years until the 162 Nd inventory has reached equilibrium. The Nd global inventory in CTRL is  $3.64 \times 10^{12}$ g, which is 163 comparable to the observational estimates of  $4.2 \times 10^{12}$ g [Tachikawa et al., 2003]. The mean 164 residence time is 508 years, which is in the range reported previously [Tachikawa et al., 2003]. 165 Both simulated Nd concentration and  $\varepsilon_{Nd}$  in CTRL are also in reasonable agreement with a 166 compilation of available observations [van de Flierdt et al., 2016] (Figs.1, Fig.2 and Fig. S1) as 167 discussed below. 168

Our model can simulate 64% of the Nd concentration observational points within  $\pm 10$ 169 pmol/kg (70% in *Rempfer et al.*, [2011]) and 83% of the  $\varepsilon_{Nd}$  observational points with  $\pm 3 \varepsilon_{Nd}$ 170 unit (83% in Rempfer et al., [2011]). Nd concentration in CTRL captures the general feature of 171 increasing with depth and also increasing along with the circulation pathway, consistent with 172 observations (Fig. 1B and Fig. S1). Similar to observations [Goldstein and Hemming, 2003], ENd 173 values exhibits an inter-basin gradient as the North Pacific has the most radiogenic  $\varepsilon_{Nd}$  values, 174 the North Atlantic has the least radiogenic values and the Indian and Southern Oceans have 175 intermediate values (Fig. 1C and Fig.2). The linear regression coefficient between model  $\varepsilon_{Nd}$  and 176 observational  $\epsilon_{Nd}$  is 0.67 ( $r^2 = 0.7$ , N = 1699). 177

Since our study focuses on the Atlantic basin, especially the tropical Atlantic, we show several  $\varepsilon_{Nd}$  vertical profiles in the Atlantic (Fig.2). Overall, our model can simulate the vertical 180 structure of  $\varepsilon_{Nd}$ , indicating the influences of water mass from different origins. For example, the zig-zag pattern in observations [Goldstein and Hemming, 2003] are successfully simulated in our 181 model (Fig.2 profile 9 and 10), as AAIW and Antarctic Bottom Water (AABW) carry radiogenic 182  $\varepsilon_{Nd}$  northward and NADW carries unradiogenic  $\varepsilon_{Nd}$  southward. In particular, our model 183 successfully captures the relative magnitude among different water masses, suggesting it can be 184 used to study the relative changes of different water masses during the deglaciation. Another 185 important feature is that our model is able to simulate the very radiogenic water from the 186 Caribbean Sea (Fig. 2 profile 7) [Osborne et al., 2014]. This turns out to be an important water 187 mass that is the source of some of the discrepancies in the  $\varepsilon_{Nd}$  reconstructions, as will be 188 discussed later in Section 4. 189

In spite of the overall agreement of the model simulation and the observations, there are 190 also some deficiencies in the model. The Nd concentration at shallow depth is lower in the model 191 than in observations and the vertical gradient is larger in the model than the observations (Fig.1B 192 193 and D, Fig.S1), as in the case of *Rempfer et al.*, [2011]. These deficiencies in simulating surface Nd is due partly to our choice of model parameters that optimize  $\epsilon_{Nd}$  instead of Nd, as in 194 195 Rempfer et al., [2011]. With extensive sensitivity experiments, Rempfer et al., [2011] shows that 196 it is impossible to optimize the simulation for both Nd concentration and  $\varepsilon_{Nd}$  simultaneously. They chose the parameters that yield the best  $\varepsilon_{Nd}$  simulation, since  $\varepsilon_{Nd}$  is the proxy used for 197 reconstructing past circulations. These parameter values are also used in our model setting. 198 199 Overall, our model can simulate the major  $\varepsilon_{Nd}$  features of the main water masses over both global scale and local scale of the tropical Atlantic and therefore should help us interpret  $\varepsilon_{Nd}$ 200 reconstructions in the tropical Atlantic in the past. 201

203 2.3 Transient deglacial simulation

The transient simulation (iPOP2-TRACE) is carried out using Nd-enabled ocean-alone 204 model CESM-POP2 to simulate the global ocean evolution from the LGM (21ka) to the late 205 Bølling-Allerød Interstadial (13ka) under realistic surface forcings. The model was first spun up 206 207 under LGM condition and then integrated to the present under surface climate forcing taken from a transient simulation in a fully coupled climate model (TRACE21k, using CCSM3), which 208 reproduced many features in last deglaciation [Liu et al., 2009; He, 2011]. The horizontal 209 resolution is nominally 3° and it has 60 vertical layers with a 10-m resolution in the upper 200m, 210 increasing to 250m below 3000m. Detailed experiment descriptions are described in Zhang, 211 [2016]. 212

We keep Nd sources and  $\varepsilon_{Nd}$  in Nd sources unchanged during the deglacial simulation 213 iPOP2-TRACE. Surface dust flux and origin [Grousset et al., 1998; Wolff et al., 2006; Lupker et 214 al., 2010] and river runoff magnitude and origin [Harris and Mix, 1999; Burton and Vance, 215 2000; Nurnberg and Tiedemann, 2004; Lézine et al., 2005; Stoll et al., 2007; Rincon-Martinez et 216 al., 2010] were reported to be changing throughout time. Boundary source of Nd is not well 217 constrained [Amakawa et al., 2000; Johannesson and Burdige, 2007; Rickli et al., 2010], 218 therefore it is hard to estimate the change in the past, although it is highly likely to happen due to 219 changes in different processes such as groundwater discharge [Zektser and Loaiciga, 1993; 220 Johannesson and Burdige, 2007] and continental erosion [Tütken et al., 2002]. Results from a 221 modeling study suggest that changes in the sources are unlikely to be important, as the 222 magnitude of the reconstructed glacial-deglacial  $\varepsilon_{Nd}$  variations is hard to obtain by only changing 223 the Nd sources and/or  $\varepsilon_{Nd}$  in Nd sources [*Rempfer et al.*, 2012b]. We also keep the particle fields 224 as the present, with no change throughout the simulation. This choice, although is not very 225

realistic [*Kohfeld et al.*, 2005], is limited by our model capability which is not fully coupled with a marine ecosystem model. This limitation will be addressed in a future study when an active marine ecosystem model is enabled. Here, our simplified model has the advantage that the change of the ocean circulation is the only factor that affects  $\varepsilon_{Nd}$  distribution, enabling us to focus on the influence of ocean circulation.

**3 Coherent AAIW response and AMOC strength**

**3.1 Reduced AAIW northward penetration but increased depth and thickness of AAIW water mass during weaker AMOC**

In the modern ocean, AAIW can be identified by a low salinity (or radiogenic  $\varepsilon_{Nd}$ ) tongue 234 originating from the subantarctic surface ocean extending northward at the intermediate depth 235 [*Talley*, 1996] (Fig. 3). Here, consistent with convention, we define  $\sigma_{AAIW}$  as the potential 236 237 density at the salinity minimum point in the South Atlantic mean potential temperature-salinity ( $\theta$ -S) diagrams. For convenience, the AAIW depth is defined as the zonal mean depth of  $\sigma_{AAIW}$  at 238 the equatorial Atlantic. The AAIW  $\varepsilon_{Nd}$  is defined as the zonal mean  $\varepsilon_{Nd}$  value at  $\sigma_{AAIW}$  (or AAIW 239 depth) at the equatorial Atlantic. The  $\sigma_{AAIW}$  in CTRL is 27.36 kg/m3, which is comparable to the 240 observation value of 27.3 kg/m3 [*Talley*, 1996]. The isopycnal line of  $\sigma_{AAIW}$  is also consistent 241 with the low salinity and the high  $\varepsilon_{Nd}$  tongue in the Atlantic (Fig. 3, green line), suggesting that 242 243 this is a good approximation for the location of AAIW core layer. The AAIW depth in CTRL is 778 meters, which is also in the range of modern observations [Talley, 1996]. 244

iPOP2-TRACE simulates the key oceanic changes during the last deglaciation. The simulated AMOC collapses during HS1 in response to freshwater forcing in the North Atlantic and then recovers rapidly in the Bølling-Allerød warming (BA, ~14.5 kyr B.P.) (Fig. 4B, black), consistent with 231Pa/230Th records from Bermuda Rise [*McManus et al.*, 2004](Fig. 4B, green)
and the original coupled model simulation [*Liu et al.*, 2009].

In iPOP2-TRACE, the northward penetration of AAIW in the Atlantic is closely linked to 250 the change of AMOC. During LGM and HS1,  $\sigma_{AAIW}$  surface also tends to follow the low salinity, 251 252 or the radiogenic  $\varepsilon_{Nd}$ , tongue of AAIW (green lines in Figs. 5C and D), as in CTRL. To better quantify the northward penetration of AAIW in the Atlantic, we estimate the AAIW northward 253 penetration latitude using Atlantic zonal mean  $\varepsilon_{Nd}$ : we first calculate the maximum  $\varepsilon_{Nd}$  value in 254 the South Atlantic above 1,200 meters, then we find the latitude that  $\varepsilon_{Nd}$  value of 1.3  $\varepsilon_{Nd}$  unit less 255 than the maximum can reach above 1,200 meters. The AAIW northward extent varies over an 256 approximately 15° latitude range during the deglaciation (Fig. 4C blue dots), with a high positive 257 correlation with the AMOC intensity (Fig. 4B black). AAIW in the Atlantic reaches 2°N during 258 the LGM, and withdraws southward after 19ka, when the AMOC starts to decrease in response 259 to the meltwater input in the North Atlantic. By late HS1, the AAIW retreats to its southernmost 260 latitude of 17°S, followed by a rapid intrusion during the BA to 1°N, in response to the AMOC 261 recovery. This HS1 southward retreat of the AAIW tongue is also obvious in the Atlantic zonal 262 mean salinity or  $\varepsilon_{Nd}$  (Fig. 5 C and D) and the horizontal distribution of  $\varepsilon_{Nd}$  at  $\sigma_{AAIW}$  surface (Fig. 263 5E and F). 264

265 Physically, the change of latitudinal extent is also consistent with that of the cross-266 equator transport of the AAIW (Fig. 4B red), which is defined as the northward transport 267 between the isopycnal surfaces of  $\sigma_{AAIW}\pm 0.5$ , and more generally, the subsurface component of 268 the NBC, in the model. The AAIW transport is reduced during the HS1 and increased again 269 during the BA, also following the AMOC [*Nace et al.*, 2014]. This result is insensitive to the choice of density interval (d), between  $\sigma_{AAIW}$  – d and  $\sigma_{AAIW}$  + d, because similar results are produced with density intervals (d) ranging from 0.1 to 0.4 (Fig. S2).

The equatorial Atlantic  $\varepsilon_{Nd}$  at the AAIW depth (AAIW  $\varepsilon_{Nd}$ ) also varies closely with the 272 AAIW northward penetration, as hypothesized in previous  $\varepsilon_{Nd}$  reconstructions [Pahnke et al., 273 2008; Xie et al., 2012; Huang et al., 2014]. Our model shows an almost linear relationship 274 between the equatorial AAIW  $\varepsilon_{Nd}$  (Fig. 4D solid black, which follows  $\sigma_{AAIW}$  and varies with 275 depth) and the northward penetration latitude of AAIW (Fig. 4C navy dot), with decreased  $\varepsilon_{\rm Nd}$ 276 during HS1 and its subsequent increase during BA corresponding to the southward withdraw and 277 the subsequent northward re-advance in the penetration latitude, respectively. In the model, we 278 calculate the  $\varepsilon_{Nd}$  of the AAIW southern end-member, which is the average  $\varepsilon_{Nd}$  in the AAIW 279 280 production region. It remains unchanged at -8.3 during the deglaciation prior to BA and shifts abruptly to -9.1 during BA due to the quick AMOC recovery during BA, which brings 281 unradiogenic  $\epsilon_{Nd}$  water from the North Atlantic to the Southern Ocean. The evolution of the  $\epsilon_{Nd}$ 282 283 difference between the equatorial Atlantic and its southern end-member (Fig. 4D, red) is similar to the evolution of the  $\varepsilon_{Nd}$  in the equatorial Atlantic (Fig. 4D, solid black). Therefore,  $\varepsilon_{Nd}$  in the 284 equatorial Atlantic at AAIW depth can indeed be used as an indicator for AAIW northward 285 penetration in the Atlantic. 286

Another important feature of AAIW is that its depth changes significantly during the last deglaciation in iPOP2-TRACE. The AAIW depth is also closely linked to the AMOC evolution, deepening from around 230-m during LGM to around 670-m during HS1, shoaling back to 240m during BA (Fig.4C red) and deepening again slowly to ~530-m in the Holocene (Fig. 4C triangle on right Y axis), which is consistent with the present day observation [*Talley*, 1996]. This deepening of AAIW from LGM to HS1 has been illustrated in previous modeling studies

[e.g. Vallis, 2000; Wolfe and Cessi, 2010]. When the surface density in the source region of 293 NADW is between the surface density in the source region of AAIW and AABW ( $\sigma_{AAIW}$  < 294  $\sigma_{\text{NADW}} < \sigma_{\text{AABW}}$ , which is the case during LGM in our simulation (Fig. 4A), NADW fills the 295 mid-depth and AAIW is shallow and partially entrained in the main thermocline. However, 296 when the surface density in the source region of NADW is less than AAIW, which is the case 297 during HS1 in our simulation, as no NADW is produced due to the melt water input to the North 298 Atlantic (Fig. 4A), AAIW fills the middepth between abyssal and main thermocline. Therefore, 299 AAIW becomes deeper and thicker during HS1. In addition, this magnitude of deepening of 300 301 middepth water during HS1 has also been suggested by the deglacial attmospheric radiocarbon decline [Hain et al., 2014]. Finally, the Holocene deepening compared with the glacial period 302 may be caused partly by the sea ice retreat in the Southern Ocean [Ferrari et al., 2014]. 303

The depth change of AAIW core layer may also contribute to  $\varepsilon_{Nd}$  change at a fixed depth. 304 As the AAIW deepens, any site above (below) AAIW core layer would experience a less (more) 305 radiogenic  $\varepsilon_{Nd}$  shift, which may complicate the interpretation of  $\varepsilon_{Nd}$  evolution as AAIW 306 northward penetration. However, the  $\varepsilon_{Nd}$  in the western boundary of equatorial Atlantic shows a 307 change of about 1 unit  $\epsilon_{Nd}$  change from the LGM to the HS1 at a fixed intermediate depth of 308 309 1000m (Fig. 4D black dash) (similar at 500m and 800m, not shown), and this change at fixed depth is comparable with the  $\varepsilon_{Nd}$  change at the AAIW core depth that changes with time (Fig.4D 310 black solid). Therefore, the  $\varepsilon_{Nd}$  change from the tropical Atlantic is dominated by the change in 311 312 the AAIW northward penetration change rather than AAIW depth change.

Overall, our model shows a coherent response between the AMOC intensity and the AAIW northward penetration latitude, northward transport, AAIW  $\varepsilon_{Nd}$  value and AAIW depth in iPOP2-TRACE. These relationships are robust in the model and have been reproduced in several idealized hosing experiments (Fig. S4 and S5). Our simulation is also consistent with a climate model of intermediate complexity [*Rempfer et al.*, 2012a] (their Figure 12a), where the zonalmean  $\varepsilon_{Nd}$  becomes more radiogenic with a maximum increases of 4  $\varepsilon_{Nd}$  units in the upper 1,200 meters of the equatorial Atlantic and decreases at greater depths for a transitions from an NADW-on state to an NADW-off state.

321

**322 3.2 Mechanism**

How does a weaker AMOC reduce the AAIW northward penetration in the Atlantic? 323 Intuitively, one might think the AAIW northward penetration of AAIW is determined mainly by 324 its production rate: a larger AAIW production rate would favor a stronger northward penetration 325 towards the North Atlantic. This is not the case in iPOP2-TRACE: AAIW northward penetration 326 is not controlled by upstream AAIW production. We compare the AAIW subduction rate, which 327 is the subduction across the base of the ocean mixed layer in the South Atlantic AAIW formation 328 region [Goes et al., 2008]. The AAIW subduction rate is 4.6 Sv during LGM and 6.0 Sv during 329 330 HS1 in iPOP2-TRACE, indicating the upstream AAIW production during HS1 is not lower but 331 even higher. This stronger HS1 AAIW production rate during HS1 also occurs in the fully coupled experiment TRACE21k, which shows a subduction rate of 16 Sv during LGM ( 332 333 consistent with Wainer et al., [2012]) and 19 Sv during HS1, although the overall magnitudes of the subduction rate are different. The relatively smaller magnitude of AAIW subduction in the 334 ocean-alone simulation (iPOP2-TRACE) than in the fully coupled simulation (TRACE21k) is 335 336 because the AAIW subduction rate depends on the mixed layer depth, which is much smaller in iPOP2-TRACE than in TRACE21k, probably because that iPOP2-TRACE is forced by monthly 337 atmospheric forcings, in which the high frequency signals are filtered out. Regardless of these 338

**Confidential manuscript submitted to Paleoceanography**

differences, the results from both simulations indicate that the retreat of AAIW northward
 penetration during HS1 cannot be caused by AAIW formation in the Southern Ocean.

Since the meltwater flux to the North Atlantic can reverse the density contrast between 341 AAIW and NADW such that AAIW becomes heavier than NADW, it could encourage the 342 northward penetration of AAIW and the southward compensating flow from the North Atlantic 343 above AAIW, forming a reversed counterclockwise shallow overturning cell that circulates in the 344 opposite direction to the modern AMOC [Keeling and Stephens, 2001; Saenko et al., 2003; 345 Weaver et al., 2003]. In our model, the higher surface density in the NADW formation region 346 during LGM ( $\sigma_{\text{NADW}}$ =28.5 kg/m3 >  $\sigma_{\text{AAIW}}$  =28.2 kg/m3) is indeed reduced to lower than that of 347 AAIW during HS1 ( $\sigma_{AAIW}=28.0 \text{ kg/m}^3 > \sigma_{NADW}=26.8 \text{ kg/m}^3$ ) (Fig. 4A). However, no reversed 348 349 AAIW cell is generated (Fig. 5B). The detailed mechanism of the reversed AAIW cell remains to be fully understood in future studies. Here, we note that, during LGM, the AAIW lies above 350 351 NADW, contributing to the return flow of NADW as in modern observation [Lumpkin and 352 Speer, 2003]; in response to the freshwater input during HS1, the southward export of NADW at depth collapses, which then reduces the compensating flow in the upper ocean, including AAIW. 353 354 As such, the AAIW retreats to south of the equator during HS1 (Fig. 5 B, D and F). This 355 response is consistent with the present day observational [Zhang et al., 2011] and modeling studies of the multi-decadal variability of the NBC, which is found to be determined 356 predominantly by the changes of the AMOC and NADW formation [Rühs and Getzlaff, 2015]. 357

Our study suggests a remote dynamical control on the AAIW northward penetration from the North Atlantic, as opposed to a local control of AAIW production and transport from the Southern Ocean. Typically, the AAIW is transported northward first through the southern subtropical gyre circulation and then across the equator by the western boundary current, as in

modern observations [Schmid et al., 2000]. During the LGM, the AAIW flows northwestward to 362  $\sim 20^{\circ}$ S in a broad interior pathway, following the counterclockwise subtropical gyre in the South 363 Atlantic at intermediate depth (Fig. 6A); most of the AAIW water, however, recirculates back 364 through the southward Brazil Current along the western boundary (Fig. 6B). A small residual of 365 AAIW advances beyond 20°S northward along the western boundary into the tropical Atlantic; 366 this part of AAIW then crosses the equator as a part of the subsurface component of the NBC 367 along the western boundary, generating a low salinity/high  $\varepsilon_{Nd}$  tongue there. The AAIW 368 penetrates across the equator only in the western boundary current because the cross-equator 369 370 penetration is largely prohibited in the interior ocean due to the conservation of potential vorticity [McCreary and Lu, 2001]. During HS1, there is little AAIW transported across the 371 equator (Fig. 6D), confining the low salinity/high  $\varepsilon_{Nd}$  tongue south of the equator (Fig. 5D). 372 Upstream in the subantarctic South Atlantic, however, the northward transport of AAIW is 373 actually increased relative to the LGM (Figs. 6B and 6D); this increased AAIW transport, 374 however, is returned southward almost entirely in the Brazil Current, leaving little AAIW 375 penetrating into the equatorial Atlantic (Fig.6D). Thus, the deglacial evolution of the AAIW 376 penetration to the tropical Atlantic appears to be determined predominantly by the remote 377 processes in the North Atlantic, rather than by the local forcing in the South Atlantic subantarctic 378 region. This remote control of AAIW in the Atlantic is similar to that in the Pacific, where the 379 cross-equator penetration of AAIW is caused predominantly by the opening of the Indonesia 380 381 Throughflow, rather than the climate forcing in the South Pacific subantarctic region [McCreary and Lu, 2001]. We also did an idealized hosing experiment (not shown), in which constant fresh 382 water forcing of 1Sv is added to North Atlantic for the first 100 years and then removed. It 383

shows similar equatorial  $\varepsilon_{Nd}$  response as in iPOP2-TRACE and  $\varepsilon_{Nd}$  lags AMOC change for 30-40 years.

- 386
- 387

**4 Reconciling $\varepsilon_{Nd}$ reconstructions controversy with core depth**

388 As noted above, available tropical  $\varepsilon_{Nd}$  reconstructions show contradictory  $\varepsilon_{Nd}$  evolutions across the last deglaciation. The  $\varepsilon_{Nd}$  reconstruction from the Tobago Basin (MD99-2198, 389 12.09°N, 61.23°W, 1330m) [Pahnke et al., 2008] shows an increase (becomes more radiogenic) 390 during the HS1 (Fig. 4F), which was interpreted as enhanced northward advection of AAIW. 391 However,  $\varepsilon_{Nd}$  records from the Florida Strait (KNR166-2-26JPC, 24°19.62'N, 83°15.14'W, 392 546m) [Xie et al., 2012] (Fig. S3C) and the Demerara Rise (KNR197-3-46CDH, 7.836°N, 393 53.663°W, 947m) [Huang et al., 2014] (Fig. 4E) show decreases (become less radiogenic) 394 during the HS1, and were interpreted to indicate decreased penetration of AAIW into tropical 395 396 North Atlantic. The controversy may be due to deficiencies of each data site. On the one hand, it was argued that MD99-2198 lies beneath the modern AAIW depth range and fails to record the 397 AAIW northward penetration signals [Xie et al., 2012]. On the other hand, present day 398 399 hydrographic data from the Gulf of Mexico shows much warmer and saltier water mass than AAIW, suggesting that if any AAIW has arrived at this site, it has already been modified by 400 other water masses. Therefore, site KNR166-2-26JPC from the Florida Strait has been suggested 401 not ideally situated to record the deglacial AAIW changes [Pena et al., 2013; Osborne et al., 402 2014]. 403

404 Our model reproduces the  $\varepsilon_{Nd}$  evolutions at different sites from intermediate depth. The 405  $\varepsilon_{Nd}$  from the Demerara Rise (~950m) (Fig. 4E and S3 A, B) and from the Florida Strait (~540m) (Fig.S3 C) exhibit less radiogenic excursion during HS1, while  $\varepsilon_{Nd}$  from the Tobago Basin (~1330m) shows a more radiogenic shift during HS1 (Fig. 4F). Our model is able to simulate the diverse  $\varepsilon_{Nd}$  evolutions consistent with the reconstructions at these three tropical North Atlantic sites and suggest that the opposite  $\varepsilon_{Nd}$  evolutions at these locations are physically consistent with a common deglacial ocean circulation change. The interpretation, however, is more complex than suggested in previous studies because it involves both the change of the AAIW depth and the radiogenic water from the Gulf of Mexico and the Caribbean Sea, as discussed below.

Our model simulation shows that the less radiogenic shift of  $\varepsilon_{Nd}$  from the Florida Strait 413 site (KNR166-2-26JPC) during HS1 [Xie et al., 2012] is due to the reduced influence of the 414 radiogenic water from the bottom in the Gulf of Mexico and the Caribbean Sea. Deep water from 415 the Gulf of Mexico and the Caribbean Sea features very radiogenic  $\varepsilon_{Nd}$  sources from boundary 416 exchange as discussed in Section 2.2 [Jeandel et al., 2007; Osborne et al., 2014]. During LGM, 417 active AMOC drives strong upwelling in this region (Fig. 7A black contour), which, in turn, 418 influences the shallow layers with very radiogenic  $\varepsilon_{Nd}$  water in this region and the nearby open 419 ocean in the subtropical North Atlantic. The influence of this regional radiogenic  $\varepsilon_{Nd}$  source can 420 also be seen in the Atlantic zonal mean  $\varepsilon_{Nd}$  as a high  $\varepsilon_{Nd}$  center located at 600m-900m from 20°N 421 to 40°N (Fig. 5C) (also in Fig. 3 in modern CTRL). During HS1, however, this radiogenic  $\varepsilon_{Nd}$ 422 bottom water is trapped in the bottom locally because of reduced upwelling (Fig. 7A black 423 contour). This leads to a great reduction in the transport of radiogenic  $\varepsilon_{Nd}$  water from bottom to 424 shallow layers and therefore, a unradiogenic  $\varepsilon_{Nd}$  shift in the upper 1,500 m in the Gulf of Mexico 425 and the Caribbean Sea (Fig. 7 A color contour) and, eventually, in the upper 1,000 m in 426 subtropical North Atlantic as there is no more a radiogenic  $\varepsilon_{Nd}$  center in subtropical North 427 Atlantic in the zonal mean  $\varepsilon_{Nd}$  (Fig. 5D). Furthermore, the  $\varepsilon_{Nd}$  from the Florida Strait site is 428

dominated by radiogenic horizontal advection (Fig. S7 A) by an eastward flow from the Gulf of 429 Mexico (Fig. S7 B).  $\varepsilon_{Nd}$  at this site experiences an unradiogenic shift during HS1 because with 430 reduced input of deep radiogenic waters, the upper ocean in the Gulf of Mexico becomes less 431 radiogenic and at the same time, the eastward flow also becomes weaker (Fig. S7 B). Thus,  $\varepsilon_{Nd}$ 432 variations in the Florida Strait are not due to variations in AAIW as previously suggested [Xie et 433 al., 2012]. Overall, the relationship between the weakened AMOC and the weakened influence 434 from the regional radiogenic  $\varepsilon_{Nd}$  influence from the Gulf of Mexico and the Caribbean Sea is 435 also robust in our idealized hosing experiment (Fig. S5 C and D), although detailed dynamics 436 that relates the weakened AMOC and the reduced upwelling in the Gulf of Mexico and 437 Caribbean Sea remains to be further studied. 438

Our model simulation further suggests that the opposite  $\varepsilon_{Nd}$  behaviors at two nearby sites 439 from the Demerara Rise and the Tobago Basin discussed above are caused by the different 440 depths of the sediment cores as well as the influence of radiogenic  $\epsilon_{Nd}$  water from the Caribbean 441 Sea. Both locations experience similar  $\varepsilon_{Nd}$  change in the upper 2,000m (Fig. 7 C and D). During 442 the LGM, the Demerara Rise site is located in the lower limb of AMOC (as shown in southward 443 meridional velocity in Fig. 8A and 9C), with water transported from the subtropical North 444 445 Atlantic and the Caribbean Sea. Starting from 19ka, AMOC begins to decrease in response to the fresh water forcing applied to the North Atlantic,  $\varepsilon_{Nd}$  in the subtropical North Atlantic becomes 446 less radiogenic due to the reduced influence of the radiogenic source water from the bottom of 447 the Gulf of Mexico and the Caribbean Sea as discussed above. In the meantime, the meridional 448 velocity also begins to decrease (Fig. 9C), leading to a decrease in the radiogenic  $\varepsilon_{Nd}$  advection 449 term (Fig. 9A). During HS1, the flow is almost stagnant (Fig. 9C) and all the  $\varepsilon_{Nd}$  tendency terms 450 are greatly reduced compared with LGM (Fig. 9A). Therefore, the less radiogenic shift in  $\varepsilon_{Nd}$ 451

**Confidential manuscript submitted to Paleoceanography**

during HS1 from the Demerara Rise is due to the reduced influence of radiogenic water from bottom of the Gulf of Mexico and the Caribbean Sea as well as the reduced southward flow, instead of the retreat of northward advection of AAIW suggested in *Huang et al.*, [2014].

The Tobago Basin site is about 400 meters deeper than the Demerara Rise site and is 455 mainly influenced by the NADW from the north, which features unradiogenic  $\epsilon_{Nd}$  values. 456 Although NADW  $\varepsilon_{Nd}$  is complicated by distinct west and east NADW source waters [van de 457 Flierdt et al., 2016], in our simulation, changes in the relative contribution from west versus east 458 NADW formation does not have much influence on the NADW  $\varepsilon_{Nd}$  value (SI. text 2), which is 459 consistent with the finding that the influence of the endmember  $\varepsilon_{Nd}$  change is rather small 460 compared with  $\varepsilon_{Nd}$  changes due to changes in watermass distribution [*Rempfer et al.*, 2012a]. 461 During LGM, strong southward western boundary current contributes to the unradiogenic  $\varepsilon_{Nd}$ 462 advections at the Tobago Basin site (Fig. 8B and Fig. 9B). When AMOC collapsed during HS1, 463 this unradiogenic  $\varepsilon_{Nd}$  advection of NADW is also reduced (Fig. 9B and D), which then 464 contributes to the more radiogenic shift of  $\varepsilon_{Nd}$  during HS1 as in the  $\varepsilon_{Nd}$  reconstruction. In 465 addition, circulation change in the Caribbean Sea also contributes to the more radiogenic  $\varepsilon_{Nd}$ 466 shift in the Tobago Basin during HS1. During LGM, flow at the location where the Caribbean 467 Sea connects with the Atlantic (12°N, 75°W, 1330m) is westward and therefore leads to a less 468 radiogenic  $\varepsilon_{Nd}$  advection into the the Caribbean Sea (Fig. 8B and Fig. S6A). During HS1, 469 however, the westward flow is changed to eastward flow out of the Caribbean Sea, because of 470 the reduced deep west boundary current (Fig. 8D and Fig. S6B). This eastward flow out of the 471 Caribbean Sea transports radiogenic  $\varepsilon_{Nd}$  water from the Caribbean Sea out to influences the 472 Tobago Basin site. Therefore, the more radiogenic  $\varepsilon_{Nd}$  shift during HS1 in Tobago Basin site is 473 474 caused by both the retreat of the unradiogenic  $\varepsilon_{Nd}$  NADW and the leak of radiogenic  $\varepsilon_{Nd}$  water

from the Caribbean Sea. Again, variations in the northward extent of AAIW did not control the  $\epsilon_{Nd}$  evolution in this Tobago Basin site, contrary to what was suggested previously [*Pahnke et al.*, 2008].

The discussion above suggests that deglacial  $\varepsilon_{Nd}$  in the low latitude North Atlantic at the 478 479 depth of modern AAIW can be complicated by the radiogenic  $\varepsilon_{Nd}$  end-member form the Gulf of 480 Mexico and the Caribbean Sea. From LGM to HS1, our model  $\varepsilon_{Nd}$  exhibits an unradiogenic shift above around 1,100-m and a more radiogenic shift from 1,100-m to 2,000-m at both the 481 482 Demerara Rise and the Tobago Basin (Fig.7 C and D), consistent with the respective proxy records. Above 1,100-m, low latitude North Atlantic  $\varepsilon_{Nd}$  can be influenced by both southern 483 sourced water of AAIW in the upper layers and northern sourced water from the Caribbean Sea, 484 both of which become weaker and lead to an unradiogenic shift of  $\varepsilon_{Nd}$  when AMOC strength is 485 reduced. Below 1,100-m, water is influenced mainly by the NADW as well as water from the 486 Caribbean Sea. The retreat of NADW and the advance of the Caribbean Sea water both lead to a 487 radiogenic shift of  $\varepsilon_{Nd}$  during reduced AMOC. Therefore, radiogenic  $\varepsilon_{Nd}$  water from the Gulf of 488 Mexico and the Caribbean Sea provides effectively the third  $\varepsilon_{Nd}$  end-member in addition to the 489 radiogenic  $\varepsilon_{Nd}$  south sourced AAIW and unradiogenic  $\varepsilon_{Nd}$  north sourced water. This third source 490 should be taken into consideration when interpreting  $\varepsilon_{Nd}$  reconstructions from low latitude North 491 Atlantic at modern intermediate depth. 492

It should also be pointed out that the interpretation of the deglacial  $\varepsilon_{Nd}$  records from the tropical Atlantic can also be complicated by the changing depth of the AAIW during the deglaciation. Our model shows a much shallower AAIW during LGM than the present day (Fig. 4C). Sites located at modern AAIW depth may not be influenced by AAIW in the past. In iPOP2-TRACE, in the western boundary of equatorial Atlantic, for the upper 900 meters, flow is northward which contributes to a radiogenic  $\varepsilon_{Nd}$  advection, indicating an AAIW influence. Therefore, we suggest that  $\varepsilon_{Nd}$  reconstructions shallower than 900 meters from equatorial and tropical Atlantic are more suitable to reconstruct past AAIW northward penetration change. The complicated mechanisms controlling  $\varepsilon_{Nd}$  reconstruction at different sites from the tropical North Atlantic, however, also indicates that more reconstructions from different locations and depths are needed to infer past circulation changes as suggested by *van de Flierdt et al.*, [2016].

504

**505 **5 Conclusions**

Overall, our transient Nd-enabled ocean model simulation suggests a coherent AAIW 506 response to the change of AMOC strength. The northward AAIW penetration in the tropical 507 Atlantic is determined predominantly by the AMOC intensity or climate in the high latitude of 508 the North Atlantic remotely, with a stronger AMOC enhancing AAIW northward penetration 509 (Fig. 10 A and B). In addition, AAIW water mass sinks to a greater depth and dominates a wider 510 water depth range in response to the freshening of NADW. Our results suggest that AAIW is a 511 critical part of the return flow of the southward flowing NADW and, in turn, the global 512 thermohaline circulation, and therefore can contribute significantly to the global climate change. 513 Also, monitoring changes of AAIW can contribute to our understanding of climate changes in 514 the past and help future projections. 515

516 During HS1, the reduced AMOC strength is caused by fresh water forcing in the North 517 Atlantic. Under this North Atlantic buoyancy forcing scenario, we find that AAIW becomes 518 deeper when AMOC is weaker. *Toggweiler and Samuels*, [1995] suggests that NADW formation 519 in the North Atlantic is also controlled by wind forcing in the Southern Ocean: weaker winds 520 over Drake Passage will lead to weaker NADW formation. Interestingly, the pycnocline depth 521 becomes shallower under weaker Southern Ocean wind forcing. This relationship between 522 pycnocline and AMOC strength under Southern Ocean wind forcing is opposite to our finding 523 under North Atlantic buoyancy forcing. Therefore, the response of the circulation at middepth to 524 the forcings from the North Atlantic and the Southern Ocean needs to be further studied.

In addition,  $\varepsilon_{Nd}$  reconstructions from the tropical and subtropical North Atlantic from 525 within and near modern AAIW depths do not inform us about northward AAIW extent as 526 previously assumed. Our simulation reproduces the contrasting deglacial  $\varepsilon_{Nd}$  evolutions at three 527 intermediate-depth sites in the tropical North Atlantic. The inconsistency among reconstructions 528 relates to the individual site locations and depths. With the AAIW depth changing in the past, 529 530 core sites bathed by AAIW in present day, such as the Demerara Rise site, may not be influenced by AAIW in the past. In addition, our results point out the importance the radiogenic  $\varepsilon_{Nd}$  water 531 from the Gulf of Mexico and the Caribbean Sea as the third end-member for regulating  $\varepsilon_{Nd}$ 532 533 values at intermediate depth in tropical North Atlantic, which complicates the interpretation of  $\varepsilon_{Nd}$  reconstruction in the tropical North Atlantic. During the AMOC-on state (LGM), upwelling 534 535 in the Gulf of Mexico and the Caribbean Sea brings very radiogenic water from the bottom to shallow depth, influencing the upper 1,000 m of the tropical and subtropical Atlantic (Fig. 10 C). 536 During the AMOC-off state (HS1), this upwelling is greatly reduced and the upper 1,000 m 537 subtropical and tropical Atlantic  $\varepsilon_{Nd}$  experience an unradiogenic shift (Fig. 10 D), which, 538 539 combined with a weak deep western boundary current, lead to the unradiogenic shift in reconstruction of the Demerara Rise site (Fig. 10C and D). The radiogenic shift in the 540 reconstruction of the Tobago Basin site during HS1 is due to the reduced deep western boundary 541 current as well as leakage of radiogenic water from the Caribbean Sea (Fig. 10E and F). 542

Therefore, we cannot interpret  $\varepsilon_{Nd}$  reconstructions from the tropical Atlantic within and near modern AAIW depth without taking the influence of radiogenic water from the Gulf of Mexico and the Caribbean Sea into consideration. Eventually, more reconstructions from different depths and latitudes, and comparison of these records to simulations using Nd-enabled models, will help

- 547 to improve our understanding of past circulation.
- 548

Acknowledgements: We are especially grateful to two reviewers and the editor for their 549 constructive comments. We thank B. Otto-Bliesner and E. Brady for their support of the work, 550 and we thank Dr J. Zhu for helpful discussions. This study was supported by the US NSF P2C2 551 552 (NSF 1401778 and NSF1401802), DOE DE-SC0006744 and NSFC 41630527 and 41130105. F.J. acknowledges support by the Swiss National Science Foundation. D. O. acknowledges 553 support from the WHOI Investing in Science Program and the US NSF. Computing resources 554 (ark:/85065/d7wd3xhc) were provided by the Climate Simulation Laboratory at NCAR's 555 Computational and Information Systems Laboratory (CISL), sponsored by the National Science 556 Foundation and other agencies. Data used to produce the results in this study can be obtained 557 558 from HPSS at CISL: /home/sgu28/iPOP TRACE.

**559 **References**

- Amakawa, H., D. S. Alibo, and Y. Nozaki (2000), Nd isotopic composition and REE pattern in
   the surface waters of the eastern Indian Ocean and its adjacent seas, *Geochim. Cosmochim. Acta*, 64(10), 1715–1727, doi:10.1016/S0016-7037(00)00333-1.
- Burton, K. W., and D. Vance (2000), Glacial-interglacial variations in the neodymium isotope
   composition of seawater in the Bay of Bengal recorded by planktonic foraminifera, *Earth Planet. Sci. Lett.*, 176(3–4), 425–441, doi:10.1016/S0012-821X(00)00011-X.
- Came, R. E., D. W. Oppo, W. B. Curry, and J. Lynch-Stieglitz (2008), Deglacial variability in
   the surface return flow of the Atlantic meridional overturning circulation,
   *Paleoceanography*, 23(1), doi:10.1029/2007PA001450.
- Ferrari, R., M. F. Jansen, J. F. Adkins, A. Burke, A. L. Stewart, and A. F. Thompson (2014),
   Antarctic sea ice control on ocean circulation in present and glacial climates., *Proc. Natl. Acad. Sci. U. S. A.*, *111*(24), 8753–8, doi:10.1073/pnas.1323922111.
- van de Flierdt, T., L. F. Robinson, J. F. Adkins, S. R. Hemming, and S. L. Goldstein (2006),
   Temporal stability of the neodymium isotope signature of the Holocene to glacial North
   Atlantic, *Paleoceanography*, *21*(4), doi:10.1029/2006PA001294.
- van de Flierdt, T., A. M. Griffiths, M. Lambelet, S. H. Little, T. Stichel, and D. J. Wilson (2016),
   Neodymium in the oceans: a global database, a regional comparison and implications for

- palaeoceanographic research, *Philos. Trans. R. Soc. A Math. Phys. Eng. Sci.*, *374*(2081),
  20150293, doi:10.1098/rsta.2015.0293.
- Foster, G. L., D. Vance, and J. Prytulak (2007), No change in the neodymium isotope
  composition of deep water exported from the North Atlantic on glacial-interglacial time
  scales, *Geology*, 35(1), 37, doi:10.1130/G23204A.1.
- Goes, M., I. Wainer, P. R. Gent, and F. O. Bryan (2008), Changes in subduction in the South
  Atlantic Ocean during the 21st century in the CCSM3, *Geophys. Res. Lett.*, 35(6), L06701,
  doi:10.1029/2007GL032762.
- Goldstein, S., and S. Hemming (2003), Long-lived isotopic tracers in oceanography,
   paleoceanography, and ice-sheet dynamics, *Treatise on geochemistry*, 6(6), 453–489.
- Goldstein, S. J., and S. B. Jacobsen (1987), The Nd and Sr isotopic systematics of river-water
   dissolved material: Implications for the sources of Nd and Sr in seawater, *Chem. Geol. Isot. Geosci. Sect.*, 66(3–4), 245–272, doi:10.1016/0168-9622(87)90045-5.
- Goldstein, S. L., R. K. O'Nions, and P. J. Hamilton (1984), A Sm-Nd isotopic study of
  atmospheric dusts and particulates from major river systems, *Earth Planet. Sci. Lett.*, 70(2),
  221–236, doi:10.1016/0012-821X(84)90007-4.
- Greaves, M., P. Statham, and H. Elderfield (1994), Rare earth element mobilization from marine
   atmospheric dust into seawater, *Mar. Chem.*, 46(3), 255–260.
- Grousset, F. E., P. E. Biscaye, a. Zindler, J. Prospero, and R. Chester (1988), Neodymium
   isotopes as tracers in marine sediments and aerosols: North Atlantic, *Earth Planet. Sci. Lett.*, 87(4), 367–378, doi:10.1016/0012-821X(88)90001-5.
- Grousset, F. E., M. Parra, A. Bory, P. Martinez, P. Bertrand, G. Shimmield, and R. M. Ellam
  (1998), Saharan wind regimes traced by the Sr-Nd isotopic composition of subtropical
  Atlantic sediments: Last Glacial Maximum vs Today, *Quat. Sci. Rev.*, *17*(4–5), 395–409,
  doi:10.1016/S0277-3791(97)00048-6.
- Gruber, N. et al. (2009), Oceanic sources, sinks, and transport of atmospheric CO2, *Global Biogeochem. Cycles*, 23(1), doi:10.1029/2008GB003349.
- Hain, M. P., D. M. Sigman, and G. H. Haug (2014), Distinct roles of the Southern Ocean and
   North Atlantic in the deglacial atmospheric radiocarbon decline, *Earth Planet. Sci. Lett.*,
   394, 198–208, doi:10.1016/j.epsl.2014.03.020.
- Harris, S., and A. Mix (1999), Pleistocene Precipitation Balance in the Amazon Basin Recorded
   in Deep Sea Sediments, *Quat. Res.*, 26(1999), 14–26, doi:10.1006/qres.1998.2008.
- He, F. (2011), SIMULATING TRANSIENT CLIMATE EVOLUTION OF THE LAST
   DEGLACIATION WITH CCSM3.
- Hendry, K. R., X. Gong, G. Knorr, J. Pike, and I. R. Hall (2016), Deglacial diatom production in

- the tropical North Atlantic driven by enhanced silicic acid supply, *Earth Planet. Sci. Lett.*,
   438, 122–129, doi:10.1016/j.epsl.2016.01.016.
- Howe, J. N. W., A. M. Piotrowski, D. W. Oppo, K.-F. Huang, S. Mulitza, C. M. Chiessi, and J.
   Blusztajn (2016), Antarctic Intermediate Water circulation in the South Atlantic over the
   past 25,000 years, *Paleoceanography*, doi:10.1002/2016PA002975.
- Huang, K.-F., D. W. Oppo, and W. B. Curry (2014), Decreased influence of Antarctic
  intermediate water in the tropical Atlantic during North Atlantic cold events, *Earth Planet*. *Sci. Lett.*, 389, 200–208, doi:10.1016/j.epsl.2013.12.037.
- Hurrell, J. W. et al. (2013), The community earth system model: A framework for collaborative
  research, *Bull. Am. Meteorol. Soc.*, *94*(9), 1339–1360, doi:10.1175/BAMS-D-12-00121.1.
- Ito, T., M. Woloszyn, and M. Mazloff (2010), Anthropogenic carbon dioxide transport in the
  Southern Ocean driven by Ekman flow., *Nature*, *463*(7277), 80–83,
  doi:10.1038/nature08687.
- Jacobsen, S. B., and G. J. Wasserburg (1980), Sm-Nd isotopic evolution of chondrites, *Earth Planet. Sci. Lett.*, 50(1), 139–155, doi:10.1016/0012-821X(80)90125-9.
- Jeandel, C., T. Arsouze, F. Lacan, P. Techine, and J. Dutay (2007), Isotopic Nd compositions
   and concentrations of the lithogenic inputs into the ocean: A compilation, with an emphasis
   on the margins, *Chem. Geol.*, 239(1–2), 156–164, doi:10.1016/j.chemgeo.2006.11.013.
- Johannesson, K. H., and D. J. Burdige (2007), Balancing the global oceanic neodymium budget:
  Evaluating the role of groundwater, *Earth Planet. Sci. Lett.*, 253(1–2), 129–142,
  doi:10.1016/j.epsl.2006.10.021.
- Keeling, R., and B. Stephens (2001), Antarctic sea ice and the control of Pleistocene climate
   instability, *Paleoceanography*, *16*(1), 112–131.
- Kohfeld, K. E., S. P. Harrison, C. Le Que, and R. F. Anderson (2005), Role of Marine Biology in
   Glacial-Interglacial CO2 Cycles, *Oceans*, 308(2005), 74–78, doi:10.1126/science.1105375.
- Large, W. G., and S. G. Yeager (2008), The global climatology of an interannually varying air–
  sea flux data set, *Clim. Dyn.*, 33(2–3), 341–364, doi:10.1007/s00382-008-0441-3.
- Lézine, A. M., J. C. Duplessy, and J. P. Cazet (2005), West African monsoon variability during
  the last deglaciation and the Holocene: Evidence from fresh water algae, pollen and isotope
  data from core KW31, Gulf of Guinea, *Palaeogeogr. Palaeoclimatol. Palaeoecol.*, *219*(3–
  4), 225–237, doi:10.1016/j.palaeo.2004.12.027.
- Liu, Z. et al. (2009), Transient simulation of last deglaciation with a new mechanism for Bolling-Allerod warming., *Science*, *325*(5938), 310–4, doi:10.1126/science.1171041.
- Lumpkin, R., and K. Speer (2003), Large-scale vertical and horizontal circulation in the North
  Atlantic Ocean, J. Phys. Oceanogr., 33(9), 1902–1920.

- Lupker, M., S. M. Aciego, B. Bourdon, J. Schwander, and T. F. Stocker (2010), Isotopic tracing
  (Sr, Nd, U and Hf) of continental and marine aerosols in an 18th century section of the Dye3 ice core (Greenland), *Earth Planet. Sci. Lett.*, 295(1–2), 277–286,
- doi:10.1016/j.epsl.2010.04.010.
- Mahowald, N. M., A. R. Baker, G. Bergametti, N. Brooks, R. a. Duce, T. D. Jickells, N. Kubilay,
   J. M. Prospero, and I. Tegen (2005), Atmospheric global dust cycle and iron inputs to the
   ocean, *Global Biogeochem. Cycles*, 19(4), doi:10.1029/2004GB002402.
- McCreary, J. P., and P. Lu (2001), Influence of the Indonesian Throughflow on the Circulation
  of Pacific Intermediate Water, *J. Phys. Oceanogr.*, *31*(4), 932–942, doi:10.1175/15200485(2001)031<0932:IOTITO>2.0.CO;2.
- McManus, J., R. Francois, and J. Gherardi (2004), Collapse and rapid resumption of Atlantic
   meridional circulation linked to deglacial climate changes, *Nature*, 428(6985), 834–837.
- Meckler, A. N., D. M. Sigman, K. a Gibson, R. François, A. Martínez-García, S. L. Jaccard, U.
  Röhl, L. C. Peterson, R. Tiedemann, and G. H. Haug (2013), Deglacial pulses of deepocean silicate into the subtropical North Atlantic Ocean., *Nature*, 495(7442), 495–8,
  doi:10.1038/nature12006.
- Nace, T. E., P. a. Baker, G. S. Dwyer, C. G. Silva, C. a. Rigsby, S. J. Burns, L. Giosan, B. OttoBliesner, Z. Liu, and J. Zhu (2014), The role of North Brazil Current transport in the
  paleoclimate of the Brazilian Nordeste margin and paleoceanography of the western tropical
  Atlantic during the late Quaternary, *Palaeogeogr. Palaeoclimatol. Palaeoecol.*, *415*, 3–13,
  doi:10.1016/j.palaeo.2014.05.030.
- Nurnberg, D., and R. Tiedemann (2004), Environmental change in the Sea of Okhotsk during the
   last 1.1 million years, *Paleoceanography*, *19*(4), 1–23, doi:10.1029/2004PA001023.
- Osborne, A. H., B. A. Haley, E. C. Hathorne, S. Flögel, and M. Frank (2014), Neodymium
  isotopes and concentrations in Caribbean seawater: Tracing water mass mixing and
  continental input in a semi-enclosed ocean basin, *Earth Planet. Sci. Lett.*, 406, 174–186,
  doi:10.1016/j.epsl.2014.09.011.
- Pahnke, K., S. L. Goldstein, and S. R. Hemming (2008), Abrupt changes in Antarctic
  Intermediate Water circulation over the past, *Nat. Geosci.*, *1*(12), 870–874,
  doi:10.1038/ngeo360.
- Palter, J. B., and M. S. Lozier (2008), On the source of Gulf Stream nutrients, *J. Geophys. Res.*,
   *113*(C6), C06018, doi:10.1029/2007JC004611.
- Pena, L. D., S. L. Goldstein, S. R. Hemming, K. M. Jones, E. Calvo, C. Pelejero, and I. Cacho
  (2013), Rapid changes in meridional advection of Southern Ocean intermediate waters to
  the tropical Pacific during the last 30kyr, *Earth Planet. Sci. Lett.*, *368*, 20–32,
  doi:10.1016/j.epsl.2013.02.028.
- Rempfer, J., T. F. Stocker, F. Joos, J.-C. Dutay, and M. Siddall (2011), Modelling Nd-isotopes

- with a coarse resolution ocean circulation model: Sensitivities to model parameters and
- source/sink distributions, *Geochim. Cosmochim. Acta*, 75(20), 5927–5950,
   doi:10.1016/j.gca.2011.07.044.
- Rempfer, J., T. F. Stocker, F. Joos, and J.-C. Dutay (2012a), On the relationship between Nd
   isotopic composition and ocean overturning circulation in idealized freshwater discharge
   events, *Paleoceanography*, 27(3), doi:10.1029/2012PA002312.
- Rempfer, J., T. F. Stocker, F. Joos, and J.-C. Dutay (2012b), Sensitivity of Nd isotopic
   composition in seawater to changes in Nd sources and paleoceanographic implications, *J. Geophys. Res.*, *117*(C12), C12010, doi:10.1029/2012JC008161.
- Resplandy, L., L. Bopp, J. C. Orr, and J. P. Dunne (2013), Role of mode and intermediate waters
  in future ocean acidification: Analysis of CMIP5 models, *Geophys. Res. Lett.*, 40(12),
  3091–3095, doi:10.1002/grl.50414.
- Rickaby, R. E. M., and H. Elderfield (2005), Evidence from the high-latitude North Atlantic for
   variations in Antarctic Intermediate water flow during the last deglaciation, *Geochemistry*,
   *Geophys. Geosystems*, 6(5), doi:10.1029/2004GC000858.
- Rickli, J., M. Frank, A. R. Baker, S. Aciego, G. de Souza, R. B. Georg, and A. N. Halliday
  (2010), Hafnium and neodymium isotopes in surface waters of the eastern Atlantic Ocean:
  Implications for sources and inputs of trace metals to the ocean, *Geochim. Cosmochim. Acta*, 74(2), 540–557, doi:10.1016/j.gca.2009.10.006.
- Rincon-Martinez, D., F. Lamy, S. Contreras, G. Leduc, E. Bard, C. Saukel, T. Blanz, A.
  MacKensen, and R. Tiedemann (2010), More humid interglacials in Ecuador during the past
  500 kyr linked to latitudinal shifts of the equatorial front and the Intertropical Convergence
  Zone in the eastern tropical Pacific, *Paleoceanography*, 25(2), 1–15,
  doi:10.1029/2009PA001868.
- Rintoul, S. R. (1991), South Atlantic interbasin exchange, J. Geophys. Res., 96(C2), 2675,
   doi:10.1029/90JC02422.
- Rühs, S., and K. Getzlaff (2015), On the suitability of North Brazil Current transport estimates
  for monitoring basin-scale AMOC changes, *Geophys. Res. Lett.*, 8072–8080,
  doi:10.1002/2015GL065695.Received.
- Sabine, C. L. (2004), The Oceanic Sink for Anthropogenic CO2, *Science (80-. ).*, *305*(5682),
   367–371, doi:10.1126/science.1097403.
- Saenko, O., A. Weaver, and J. Gregory (2003), On the link between the two modes of the ocean
   thermohaline circulation and the formation of global-scale water masses, *J. Clim.*, *16*(17),
   2797–2801.
- Sarmiento, J. L., N. Gruber, M. A. Brzezinski, and J. P. Dunne (2004), High-latitude controls of
   thermocline nutrients and low latitude biological productivity., *Nature*, 427(6969), 56–60,
   doi:10.1038/nature10605.

- Schmid, C., G. Siedler, and W. Zenk (2000), Dynamics of Intermediate Water Circulation in the
   Subtropical South Atlantic, *J. Phys. Oceanogr.*, 3191–3211.
- Schmitz, W., and M. McCartney (1993), On the north Atlantic circulation, *Rev. Geophys.*, (92),
   29–49.
- Stoll, H. M., D. Vance, and A. Arevalos (2007), Records of the Nd isotope composition of
  seawater from the Bay of Bengal: Implications for the impact of Northern Hemisphere
  cooling on ITCZ movement, *Earth Planet. Sci. Lett.*, 255(1–2), 213–228,
  doi:10.1016/j.epsl.2006.12.016.
- Stouffer, R., D. Seidov, and B. Haupt (2007), Climate response to external sources of freshwater:
   North Atlantic versus the Southern Ocean, *J. Clim.*, *30*(3), 436–448.
- Tachikawa, K., V. Athias, and C. Jeandel (2003), Neodymium budget in the modern ocean and
  paleo-oceanographic implications, *J. Geophys. Res.*, *108*(C8), 3254,
  doi:10.1029/1999JC000285.
- Talley, L. (1996), Antarctic Intermediate Water in the South Atlantic.
- Toggweiler, J. R., and B. Samuels (1995), Effect of drake passage on the global thermohaline
   circulation, *Deep. Res. Part I*, 42(4), 477–500, doi:10.1016/0967-0637(95)00012-U.
- Tütken, T., A. Eisenhauer, B. Wiegand, and B. T. Hansen (2002), Glacial-interglacial cycles in
  Sr and Nd isotopic composition of Arctic marine sediments triggered by the
  Svalbard/Barents Sea ice sheet, *Mar. Geol.*, *182*(3–4), 351–372, doi:10.1016/S00253227(01)00248-1.
- Vallis, G. (2000), Large-scale circulation and production of stratification: Effects of wind,
   geometry, and diffusion, *J. Phys. Oceanogr.*, (1962), 933–954.
- Wainer, I., M. Goes, L. N. Murphy, and E. Brady (2012), Changes in the intermediate water
   mass formation rates in the global ocean for the Last Glacial Maximum, mid-Holocene and
   pre-industrial climates, *Paleoceanography*, 27(3), n/a-n/a, doi:10.1029/2012PA002290.
- Weaver, A. J., O. A. Saenko, P. U. Clark, and J. X. Mitrovica (2003), Meltwater pulse 1A from
  Antarctica as a trigger of the Bølling-Allerød warm interval., *Science (80-. ).*, 299(5613),
  1709–13, doi:10.1126/science.1081002.
- Wolfe, C. L., and P. Cessi (2010), What Sets the Strength of the Middepth Stratification and
   Overturning Circulation in Eddying Ocean Models?, *J. Phys. Oceanogr.*, 40(7), 1520–1538,
   doi:10.1175/2010JPO4393.1.
- Wolff, E. W. et al. (2006), Southern Ocean sea-ice extent, productivity and iron flux over the
  past eight glacial cycles., *Nature*, 440(7083), 491–496, doi:10.1038/nature06271.
- Xie, R. C., F. Marcantonio, and M. W. Schmidt (2012), Deglacial variability of Antarctic
   Intermediate Water penetration into the North Atlantic from authigenic neodymium isotope

- ratios, *Paleoceanography*, 27(3), doi:10.1029/2012PA002337.
- Zahn, R., and A. Stüber (2002), Suborbital intermediate water variability inferred from paired
   benthic foraminiferal Cd/Ca and δ 13 C in the tropical West Atlantic and linking with North
   Atlantic, *Earth Planet. Sci. Lett.*, 200, 191–205.
- Zektser, I. S., and H. A. Loaiciga (1993), Groundwater fluxes in the global hydrologic cycle:
   past, present and future, *J. Hydrol.*, *144*(1–4), 405–427, doi:10.1016/0022-1694(93)90182 9.
- Zhang, D., R. Msadek, M. J. McPhaden, and T. Delworth (2011), Multidecadal variability of the
   North Brazil Current and its connection to the Atlantic meridional overturning circulation,
   *J. Geophys. Res.*, *116*(C4), C04012, doi:10.1029/2010JC006812.
- Zhang, J. (2016), Understanding the deglacial evolution of deep Atlantic water masses in an
   isotope-enabled ocean model.
- 768

770 Figures

---

## Author Comment (AC2) · 11 Aug 2017

We thank the reviewer for his/her time for constructing the comments.

In the following, we have addressed all comments, with the original review text underlined in italics and red.

"The paper 231Pa and 230Th in the ocean model of the community Earth system model (CESM1.3)" by S. Gu and Z. Liu is presenting the implementation of 231Pa and 230Th in their general circulation model. It is mainly following the procedure defined by previous work Siddall et al (2005) and Dutay et al (2009). The implementation of the tracers in the model is described and results are compared to observations. However some severe weaknesses are found in the manuscript. The comparison with
observation is insufficient, it is strictly following the analysis performed by Siddall et al in 2009, while It now exists , thanks to the GEOTRACES project, new data set. Moreover, the paper do not only show the implementation of the tracer in the model and its validation, which is the scope of the GMD journal, It also propose the response to hosing experiments that is paleoclimate studies that are application that are not devoted to this journal, Climate of the past would be a more appropriate journal if this study was more correctly analysed. For all these reasons I propose to reject this paper from publication in GMD."

Thanks for pointing out the new data set provide by GEOTRACES. In our revised manuscript, we include this new data set. A recent study by Rempfer et al., (2017) shows 231Pa and 230Th in Bern3D model. We also compare our results with theirs. The results in the hosing experiment is an example to show the advantages of our model. The interpretation of sediment 231Pa/230Th as a paleo proxy for reconstructing AMOC has been questioned because it will also be influenced by particle flux change. Our model includes two versions of 231Pa and 230Th, which can help to detangle these two effects. The hosing experiment is an example to show that with these two versions of 231Pa and 230Th, our model is able to help the interpretation of paleo 231Pa/230Th reconstructions. GMD encourage submissions with "tangible and potentially useful advance related to model development" (Editorial 1.1, Introduction) and we think the content in the hosing experiment fits this scope.

"Specific comments: Page 4 section 2.2. The authors show particle flux surface horizontal distribution without concrete comparison with observation. This diagnostic is interesting but it is not sufficient for the proposed study. The model uses particle concentrations and results are strongly dependent to the quality of these fields. It now exist observations to validate the particle fields (Lam et al, 2015) that were not available for Siddall et al (2005) and Dutay et al (2009). A more detailed analysis of the vertical particle concentration distribution at large scale is required."

The particle fields used in this study is generated from the ecosystem module of the

CESM, which has been validated extensively in previous studies (e.g. Doney et al., 2009; Long et al., 2013; Moore et al., 2002, 2004; Moore and Braucher, 2008). The export production is similar to satellite observations in both pattern and magnitude (Sarmiento and Gruber 2006). Global average POC concentration is 2.6*10-6 kgC/m3; CaCO3 is 1.1*10-6 kgC/m3 and opal is 3.9*10-6 kgSi/m3, consistent with Rempfer et al., (2011). Therefore, the particle fields in CESM is more or less right, although regional discrepancies from observation may exist. We appreciate the reviewer's suggestion to validate the performance of the ecosystem module of the CESM with new data. But our focus of study is the Pa/Th in the model.

Also, we show the distribution of particle fields to help the discussion of sediment 231Pa/230Th, which is influenced largely by particle distribution. Compare with Siddall et al., 2005, Dutay et al., 2009, and Rempfer et al., 2017, all models use particle fields generate from different models (but the general patterns are the same) but yields similar 231Pa and 230Th results.

"Page 5 section 2.3 Abiotic and Biotic name for simulations are not appropriate. These names suggest that the tracers are subject to different processes while it is not the case. The two approaches are the same except that the particles fields are fixed in the Abiotic run. None biogeochemical process affects the tracer except adsorption and desorption onto particles, so the appellation Biotic run seems exaggerated. Line 162: No validation of particle fields is preformed while it affect strongly the model results. Observations are now available (see for instance lam et al 2015)"

Thanks for pointing out this inappropriate usage. We have renamed the version which is coupled to the ecosystem model as "p-coupled" and the version which uses prescribed particle fields as "p-fixed".

"Pages 7 and 8 section 4, results Definition and way of estimation of the residence time given for the tracers should be explained."

The residence time is calculated as the ratio of global average total isotope activity and

the radioactive ingrowth of the isotope. The way of calculated is used in Rempfer et al., (2017) and Yu et al., (1996). We add this in the revised manuscript (line 248-249).

"Comparison of Atlantic zonal averaged model results with observations is no more adequate. It is strictly following analysis performed by Siddall et al (2005) and Dutay et al (2009) a decade ago, but now many new observations are available in the different basins thanks to the GEOTRACES program. This validation is not appropriate any more. Discussion concerning the ratio 231Pa/230Th is very poor. More detailed analysis must be given. For instance what causes low ratio in the north atlantics south of Grennland: convection?"

With the new GEOTRACES data, we update the model data comparison with two GEO-TRACES transects in the Atlantic (Fig.2 and 3). This is a more appropriate comparison than Atlantic zonal mean figure.

The large-scale feature of sediment 231Pa/230Th is small value in North Atlantic and large value in the Southern Ocean discussed in line 282-293. Regionally, the distribution of sediment 231Pa/230Th is controlled by particle distribution (especially opal) due to the particle flux effect (line 56-58). The low values south of Greenland at about 50N is because of this particle flux effect (line 293-296). Opal production is larger in both south and north of this region. Therefore, the particle flux effect will transport 231Pa out of this region, resulting lower sediment 231Pa/230Th in this region and higher sediment 231Pa/230Th north and south of this region.

"Page10 and 11. This part is already an attempt to use the model development for scientific question. It is not the purpose of GMD papers. This part should be more deeply analysed and submitted to another more appropriate journal (eg climate of the past)"

The purpose of implementing 231Pa and 230Th in CESM is to provide a tool to better interpret sediment 231Pa/230Th reconstructions. The advance of our modelling study compared with previous studies is that we have two version of 231Pa and 230Th

to separate the circulation effect and particle effect, both of which will change in response to freshwater forcing. Section 4.3 is to examine this model feature and show that although circulation effect dominates sediment 231Pa/230Th over low productivity regions in the North Atlantic and on long time scale, particle effect can be important over high productivity region and on short time scale. This part is an example to show the model advantage to detangle these two effects and therefore we think it is important to include this part to demonstrate our model advantage.

---

## Author Comment (AC3) · 11 Aug 2017

We thank the reviewer for his/her time for constructing the comments.

In the following, we have addressed all comments, with the original review text underlined in italics and red.

"The main point of criticism I have here is their comparison to observational data, which I find is too nebulous and not supported by newer data. There is an obvious lack of consideration of recent papers. More recent studies would provide a much better basis for comparison and reality-checks of the model. The references for the observational data given in the MS are quite old holding mostly data obtained by the noisy counting-method resulting in large analytical uncertainties. Instead the model should be cross-

checked with newer sedimentary and water column data. I don't see much benefit from comparing "biotic" against "abiotic" 231Pa and 230Th particle-fluxes (Fig. 2), as long as the absolute values have not been tested against new observational data. The authors urgently need to test the output of the model versus recent sedimentary data (e.g. (Böhm et al., 2015; Bradtmiller et al., 2014; Burckel et al., 2016; Henry et al., 2016; Hoffmann et al., 2013; Jonkers et al., 2015; Lippold et al., 2011; Lippold et al., 2016; Lippold et al., 2012; Luo et al., 2015; Negre et al., 2010; Roberts et al., 2014; Rutgers van der Loeff et al., 2016)), water data (e.g. (Deng et al., 2014; Hayes et al., 2014; Hayes et al., 2013; Hayes et al., 2015a; Hayes et al., 2015b; Kretschmer et al., 2011)) and most importantly other modelling studies (e.g. (Dutay et al., 2015; Lippold et al., 2011; Rempfer et al., 2017))."

Thanks for pointing recent available observations. We have updated our analysis with more complete data. The references for observations are listed in Table 3, which includes all the references used for model data comparison in Rempfer et al., (2017). Unfortunately, there is no intercalibrated dataset available.

In the revised manuscript, we replace the zonal mean figure with the GEOTRACE transects (Fig. 2 and 3), which seems to be more appropriate for direct model-data comparison. These two GEOTRACES transects are also shown in Rempfer et al. 2017. Our modelling scheme is essentially the same as Siddall et al., (2005) and the experiment Re3d in Rempfer et al., (2017), which does not include boundary scavenging and sediment resuspensions. Our results along the two GEOTRACES transects are similar to the Re3d in Rempfer et al., (2017). For dissolved 231Pa and 230Th, our model can simulate the right magnitude as in observations (Fig. 2 and 3) except in the abyssal. The larger values in the abyssal compared with observations is because we do not include boundary scavenging and sediment resuspensions in our model. As shown in Rempfer et al., (2017), if boundary scavenging and sediment resuspensions are added, the model performance in simulating the dissolved 231Pa and 230Th will be much improved (their Fig. 2 and 3 top and bottom row). This is discussed in the

revised manuscript (Line 255-263).

Rempfer et al., (2017) suggests that boundary scavenging and sediment resuspensions are unimportant for particulate 231Pa/230Th. Our particulate 231Pa/230Th (Fig. 2c and Fig. 3c) in the Atlantic show similar results as Rempfer et al., (2017). Most importantly, our sediment 231Pa/230Th compares well with available observations (Fig. 4): low values in North Atlantic and high values in the Southern Ocean; high values in high productivity regions (Line 281-296). In addition, we show side by side comparison between "abiotic" and "biotic" version in revised Fig. 2, 3 and 4 to directly show that the two versions give identical results in CTRL (Line 237-246). Although these two are similar in CTRL, they do vary differently in the HOSING experiment. Therefore, we find it may be clearer for readers to directly see the comparison between the two version in both CTRL and HOSING.

"I find the terms "biotic 231Pa/230Th" and "abiotic 231Pa/230Th" quite confusing. Since there is no biotic 231Pa and 230Th these terms should be used only to distinguish between the usage of particle fields in the model."

Thanks for pointing out this inappropriate usage. We have renamed the version which is coupled to the ecosystem model as "p-coupled" and the version which uses prescribed particle fields as "p-fixed" as suggested.

"Given that (Rempfer et al., 2017) recently provided insights into an upgraded approach by (Siddall et al., 2005) and (Siddall et al., 2007), including a bio-geochemical-module in the model, I do not see much advance provided by the here presented MS. I did not find a reference to (Rempfer et al., 2017), maybe because this is a very recent publication, but I don't think the authors should neglect this paper in a new version."

Thanks for referring to Rempfer et al., (2017). We add comparison with their results in the revised manuscript. In CTRL, our water column dissolved 231Pa and 230Th is similar as Re3d in Rempfer et al., (2017) which do not include boundary scavenging and sediment resuspensions. The particulate 231Pa/230Th in the Atlantic is also similar to Rempfer et al., (2017). In the hosing experiment, our model produces the similar spatial dependence of particulate 231Pa/230Th in the Atlantic (our Fig. 12 and their Fig.8). The text referring to Rempfer et al., 2017 are in line 86-89, 202-207, 255-263, 423-444.

"Although I welcome very much the provision of the Fortran code the reader is left alone with the comparison between model and observations (Fig.3) without sufficient information about the values, observational error bars and references. The color code in Fig. 3 may hold some information about the water depths, but since (already) older publications demandingly have shown, that the correlation of 231Pa/230Th with water-depth seems to be a manifested pattern of AMOC in the 231Pa/230Th distribution (Burckel et al., 2016; Gherardi et al., 2009; Gherardi et al., 2010; Hoffmann et al., 2013; Luo et al., 2010; Luo et al., 2015) this feature is required to be reproduced by a meaningful model. But I'm not able to see this from the provided figures."

Thanks for pointing out the important depth dependence of 231Pa/230Th. In our revised Fig. 2 and 3, particulate 231Pa/230Th in the Atlantic transects are shown. 231Pa/230Th increases with depth as suggested by previous studies (Line 277-280). We also show North Atlantic average particulate 231Pa/230Th profile in Fig.12. We further discuss this depth dependence in the HOSING experiment (Line 423-444). Our results supports the argument that this depth dependence is caused by the lateral transport of 231Pa by ocean circulation (Gherardi et al., 2009; Lippold et al., 2011, 2012; Luo et al., 2010).

"By the way, the diagrams are way too detailed (in terms of graphic resolution) demanding a lot of computer resources and slowing down even my reasonably new computer just by scrolling down."

Sorry the resolution of figure is too large. We have compress this figure in the revised manuscript.

"The table for the K values (Table 1) needs to be accompanied by references, because

these values vary within a wide range according to the studies by (Chase et al., 2002, 2004; Hayes et al., 2013; Hayes et al., 2015b; Kretschmer et al., 2011; Kretschmer et al., 2008; Luo et al., 1999, 2003, 2004) and others. I think, a well selected digest of values can be found at the new study by (Rempfer et al., 2017)."

The K values used in our control experiment are the same as what used in Siddall et al., (2005), which is from Chase et al., (2002). We have added these references in the Table 2 (originally Table 1) caption in the revised manuscript.

"Besides the shortcomings of the MS regarding the observational data, I also find patterns in the model output, which are not observed in reality to my knowledge. E.g. the appearance of a high opal/POC field in the NW-Atlantic. Further, I see an obvious mismatch of model and observations in Fig. 5, which is not explained."

The particle fields are produced by the marine ecosystem module in CESM. This ecosystem module is have been discussed in many previous studies (e.g. Doney et al., 2009; Long et al., 2013; Moore et al., 2002, 2004; Moore and Braucher, 2008) (Line 122-123). The general pattern globally is similar to the satellite observations (Sarmiento and Gruber 2006). For example, low production in subtropical gyre; high opal in the Southern Ocean. Regionally, the mismatch can be caused by many different aspects, such as modelling scheme, model resolution and biases in boundary conditions. How to improve the performance of the marine ecosystem module is beyond the scope this study.

The Fig. 6 (originally Fig. 5) shows the results of sensitivity experiments. The discussion is in line 303-310. The mismatch of model and observation is reasonable since we change the partition coefficients K in these two experiments. Take EXP_1 for example, the simulated dissolved 231Pa and 230Th (Fig. 6 and b) are much larger than observations because in EXP_1, K is decreased from CTRL by a factor of 5. Smaller K means smaller sink for 231Pa and 230Th, with the source kept the same, dissolved 231Pa and 230Th will increase. The mismatch of model and observations also suggest

that K is in the correct magnitude in CTRL.

"In summary, it is hard for me to see that the here presented model approach provides any new insights on the 231Pa/230Th method. Due to the lack of information about the model-data comparison it is not possible to assess the quality of the model and the applied parameters. Consequently I suggest revising both the model runs and the MS thoroughly before publication can be considered."

In our revised manuscript, we compare our model results with new GEOTRACES data and also compare with the recent modelling study by Rempfer et al., (2017). Overall, our model can simulate the general features in water column 231Pa and 230Th and sediment 231Pa/230Th. Different from Rempfer et al., (2017), we have two versions of 231Pa and 230Th: p-fixed and p-coupled, which have the advantage to detangle the circulation effect and particle effect in controlling sediment 231Pa/230Th. In our hosing experiment, these two version of 231Pa and 230Th do show different responses. Therefore, our model is a useful tool to improve the interpretations of 231Pa/230Th reconstructions.
* * *

---

## Referee Report (RR1)

Here the authors provide a revised version of the manuscript (MS). The new version has overcome some points of criticism from the last round of reviews. However, there is still more work needed in order to increase consistency, readability and traceability of the manuscript, in my opinion.

General comments:

The readability of the MS could be improved by proofreading by a native-speaker. There are several passages in the text which are hard to understand, simply because grammar issues sometimes obscure the logic and line of argumentation behind the words.

I appreciate that the implementation of $^{231}Pa/^{230}Th$ into CESM and the provision of the source code is a reasonable step forward. Still I would have welcomed if the examined scenarios and parameter sets would have been more realistic in a sense that they can be actually used for testing (pale)oceanographic hypotheses (see comments below).

Specific comments:

Line 17: "p-coupled" and "p-fixed" are not generally known terms. They should not be used without explicit definition. Please consider rephrasing, e.g.: In addition to the fully coupled implementation of the scavenging behaviour of $^{231}Pa$ and $^{230}Th$ with the active marine ecosystem module (p-coupled), another form of $^{231}Pa$ and $^{230}Th$ scavenging have also been implemented with prescribed particle flux fields of the present climate (p-fixed).

Line 96: Please explain in more detail: how can the effects of circulation on $^{231}Pa/^{230}Th$ be separated from the effects of particle fluxes simply by using two different non-confirmed particle schemes?

Table 2: Please add p-fixed or p-coupled to the scenarios respectively.

Table 3: Is there a reason for the iterating and non-iterating grey layers? Some references appear twice.

Line 216: I do not agree the parameter set used by (Siddall et al., 2005) is a reasonable choice, only because "[…] the control experiment in Siddall et al., (2005) is able to simulate major features of 231Pa and 230Th distributions […]". Choosing the parameter set more carefully and based on more recent approaches may help yielding more realistic simulations. (Rempfer, Stocker, Joos, Lippold, & Jaccard, 2017) listed different experimental studies suggesting a more balanced choice on K values. With the upper limit K value used for opal by (Siddall et al., 2005) the particle effect are inevitably overestimated.

Line 230 and Fig. 9: a freshwater input of 1 Sv for 1.2 ka is way too high in order to simulate any past fresh-water flux (Carlson & Clark, 2012). If the authors want to show that AMOC and $^{231}Pa/^{230}Th$ are a function of fresh-water flux then their study is presented approx. 20 years too late. But if they want to improve our knowledge on the reaction of $^{231}Pa/^{230}Th$ on realistic fresh-water fluxes of the past, they should lower the fresh-water input. I think the authors miss an opportunity here.

They also miss an opportunity by not-implementing bottom-scavenging. There are new GEOTRACES data out, which suggest non-negligible effects from nepheloid layers on $^{231}Pa/^{230}Th$. I would expect that (at least for p-fixed) this would be very laborious.

Lien 242: I don't understand this sentence at all. There is a reference to statistical values in Fig. 4a which are not there.

Figure 5: I cannot follow the statistics provided here. The yellow points in 5a hardly lead to a slope close to 1.

Line 259: I think I cannot accept that the dissolved fractions are simulated so utterly bad, simply "[…] because boundary scavenging and sediment resuspensions are not included in our model […]". I suggest first that the authors re-examine the observational data. Which of the outliers (e.g. Fig. 5c) are reliable values with reasonable errors? Because what can we learn from a parameter set and model which is not able to reconstruct the magnitude of the particulate fraction. If this was already a problem in the studies by (Dutay, Lacan, Roy-Barman, & Bopp, 2009) and (Siddall et al., 2005), why not recalibrate the model? How did (Rempfer et al., 2017) cope with this problem?

Line 281: "The sediment $^{231}Pa/^{230}Th$ in CTRL is overall consistent with observations […]". Wouldn't it be interesting to go into more detail here? Where are they consistent? Which basin, which water depth? Is margin distance an issue? By carving out which region is worse represented than others a lot could be learned about and from the model. E.g. Southern Ocean: because opal fluxes are so high $^{231}Pa/^{230}Th$ can vary a lot (much more than in the Atlantic). Simulating correct absolute values is almost impossible because opal flux varies on very small spatial scales, which cannot be captured by any model. Thus, the quality of the model run assessed by observations from this area will inevitably lead to bad agreement.

Line296: Where is the statement given here shown/demonstrated? Figure?

Line 303, Fig.6: I cannot follow the argumentation here. It would be necessary to increase the scale on Fig. 6and b in order to better resolve the high values. At the moment any variations are hidden within the red colour. The finding, that K influences dissolved fractions but not particulate fractions needs much more explanation. The simplification with reference to Eq. 3 and 7 does not help much.

Table 2: More realistic values for EXP1 and 2 would be appreciated in order to derive helpful insights from the model runs.

Line 329: This statement should be proved statistically (like Fig. 5).

Line 360: In the following paragraph the effects of opal on $^{231}Pa/^{230}Th$ is discussed. However, the model generates opal fluxes not in agreement with reality. In the response to the reviewer the authors claim that the large scale global opal production is reflected well in the model (e.g. high in SO). I agree. They also claim that the question, why the models produces a "fake-bloom" of opal production in the Western North Atlantic, is beyond the scope of this study. I may accept this (but then one may questioning the validity of the model approach), however in this case the paragraph following line 360 needs to be written more carefully and with a clear statement, that opal is not well represented on smaller spatial scales. Same with line 409.

Line 419: Of course studies on AMOC reconstructions need to cross check opal fluxes, but this sentence spreads a way too negative message when based on unrealistic opal fluxes and hence I do not agree. Please rephrase.

Fig 9: the difference between coupled and fixed are partly so big, that I wonder how both methods did agree so well before. Differences in the range of $\Delta^{231}Pa/^{230}Th>0.1$ (e.g. 9d) are not increasing my confidence in the model. Observations are much more constrained. Again I plead for applying realistic model parameters only. Further, I could not find information on water depth and longitude of the values shown in Fig. 9 diagrams, which are essential for the interpretation.

Line 424: Why is there a decrease of $^{231}Pa/^{230}Th$ above 2 km only? To my understanding and as stated in line 442 the decrease affects all of the NADW seized water depths.

Line 460: Yes, the parameters are somewhere in the range of the right magnitude, but not more. It would be great if this study would help to represent $^{231}Pa/^{230}Th$ in a realistic model, not only somewhere in the range of a factor of 25.

Fig10b: site locations are not visible.

Fig12c: Please explain the change of direction of $^{231}Pa/^{230}Th$ with depth at about 4000m for ON

Carlson, A., & Clark, P. 2012. Ice sheet sources of sea level rise and freshwater discharge during the last deglaciation. *Reviews of Geophysics*, 50: RG4007.

Dutay, J., Lacan, F., Roy-Barman, M., & Bopp, L. 2009. Influence of particle size and type on $^{231}Pa$ and $^{230}Th$ simulation with a global coupled biogeochemical-ocean general circulation model: A first approach. *Geochemistry Geophysics Geosystems*, 10(1).

Rempfer, J., Stocker, T. F., Joos, F., Lippold, J., & Jaccard, S. L. 2017. New insights into cycling of 231Pa and 230Th in the Atlantic Ocean. *Earth and Planetary Science Letters*, 468: 27-37.

Siddall, M., Henderson, G., Edwards, N., Frank, M., Müller, S., Stocker, T., & Joos, F. 2005. $^{231}Pa/^{230}Th$ fractionation by ocean transport, biogenic particle flux and particle type. *Earth and Planetary Science Letters*, 237: 135-155.

---

## Author Response (AR2)

Reply to referee's comments

Dear editor,

We thank your and the reviewer's time for constructing the comments.

In the following, we have addressed all the comments, with the original review text
underlined in italics and red.
*Line 17: "p-coupled" and "p-fixed" are not generally known terms. They should not be*
*used without explicit definition. Please consider rephrasing, e.g.: In addition to the fully*
*coupled implementation of the scavenging behaviour of 231Pa and 230Th with the active*
*marine ecosystem module (p-coupled), another form of 231Pa and 230Th scavenging*
*have also been implemented with prescribed particle flux fields of the present climate (p-*
*fixed).*
Thanks for this advice. We have rephrased as suggested (line 17-20).
*Line 96: Please explain in more detail: how can the effects of circulation on*
*231Pa/230Th be separated from the effects of particle fluxes simply by using two*
*different non-confirmed particle schemes?*
We have modified the text to make it clearer. For p-fixed Pa/Th, the particle flux is fixed
at present values and the only thing affect p-fixed Pa/Th is ocean circulation. For p-
coupled Pa/Th, it is coupled to ecosystem, therefore, is influenced by both ocean
circulation and particle flux. For example, during HS1, both AMOC and productivity is
suggested to be changed which will influence Pa/Th. Therefore, it is hard to detangle
these two effects. Our model can help to solve this problem. For example, in our model,
if we add freshwater forcing to North Atlantic, both productivity and AMOC changes
will influence p-coupled Pa/Th. But in p-fixed Pa/Th is only influenced by AMOC
change. Therefore, the effect of particle flux can be approximately estimated as p-couple
minus p-fixed (line 86-93).
*Table 2: Please add p-fixed or p-coupled to the scenarios respectively.*
We add clarification as suggested (lin 888-891). As stated in Section 3 (line 207-209),
both p-fixed and p-coupled are in CTRL, but only p-fix is available in Exp_1 and Exp_2
for computational efficiency. The p-fixed and p-coupled results in CTRL are identical
(Line 235-244).
*Table 3: Is there a reason for the iterating and non-iterating grey layers? Some*
*references appear twice.*
There is no particular reason for the iterating and non-iterating grey layers. The left
column is references for water column activity and the right column is for Holocene core
top Pa/Th. Some references have both column activity and Pa/Th, therefore appear twice.
*Line 216: I do not agree the parameter set used by (Siddall et al., 2005) is a reasonable*
*choice, only because "[…] the control experiment in Siddall et al., (2005) is able to*
*simulate major features of 231Pa and 230Th distributions […]". Choosing the parameter*

*set more carefully and based on more recent approaches may help yielding more realistic simulations. (Rempfer, Stocker, Joos, Lippold, & Jaccard, 2017) listed different experimental studies suggesting a more balanced choice on K values. With the upper limit K value used for opal by (Siddall et al., 2005) the particle effect are inevitably overestimated.*

In recent studies by Rempfer et al., 2017, they include bottom scavenging and boundary scavenging. In their study, they fix the fractionation factor (f, in their study, table A1 in their supplementary information, fractionation of $^{231}$Pa and $^{230}$Th by a certain particle type) and use scavenging efficiency as a tuning parameter ($\sigma_0$, in their study), keeping the fractionation factor the same. As they have pointed out, information about fractionation by different particles are still very limited (Chase et al., 2002; Scholten et al., 2005; Walter et al., 1997). The fractionation factor for opal used in our study is 0.3, while it is 1 in Rempfer et al., 2017. Observations suggests 0.2 from Luo & Ku, 2004, 0.3 from Chase et al., 2002 and 2.8 from Geibert & Usbeck, 2004. The fractionation factor for CaCO3 in our study is 40, while it is 10 in Rempfer et al., 2017. Observations suggests 3.8 from Roberts et al., 2009, 10 from Luo & Ku, 2004, 2.3-37 from Geibert & Usbeck, 2004 and 42 from Chase et al., 2002. Fractionation factor suggested by observations varies and our choice is in the range of observations. We agree that more sensitivity experiments will definitely help to improve the model performance. Our study is the first step trying to implementing $^{231}$Pa and $^{230}$Th into CESM. The parameters can be improved in the future with more observations available (line 489-492).

*Line 230 and Fig. 9: a freshwater input of 1 Sv for 1.2 ka is way too high in order to simulate any past fresh-water flux (Carlson & Clark, 2012). If the authors want to show that AMOC and 231Pa/230Th are a function of fresh-water flux then their study is presented approx. 20 years too late. But if they want to improve our knowledge on the reaction of 231Pa/230Th on realistic fresh-water fluxes of the past, they should lower the fresh-water input. I think the authors miss an opportunity here.*

Thanks for pointing this out. We agree 1 Sv is too high for realistic fresh water forcing. However, in our idealized ocean alone experiment under present day climate forcing, fresh water has to be this large to shut down AMOC. We have run several different experiments, with fresh water forcing increasing from 0 to 1 Sv (Table below). If fresh water is 0.1 Sv, which is the order of realistic fresh water flux during Heinrich Event Stadial 1 (HS1), the AMOC is reduced only a little compared with control experiment. However, using the same model, under realistic forcing, this model is able to simulate the transient AMOC responses from 22ka to 13ka (Zhang et al., 2017). Therefore, AMOC response may depend on the initial climatology or depend on the location of fresh water forcing, which is out of the scope of this study. In our study, our experiments are highly idealized and we want to test how $^{231}$Pa/$^{230}$Th responds to AMOC change. To shut down AMOC, we have to use this unrealistically large fresh water flux.

| FW (50-70N) | 0 | 0.1 | 0.3 | 0.5 | 0.7 | 1.0 |
|---|---|---|---|---|---|---|
| AMOC | 15.6 | 13.4 | 8.7 | 4.9 | 3.4 | 2.0 |

*They also miss an opportunity by not-implementing bottom-scavenging. There are new*
*GEOTRACES data out, which suggest non-negligible effects from nepheloid layers on*
*231Pa/230Th. I would expect that (at least for p-fixed) this would be very laborious.*
We agree nepheloid layers are important. Rempfer et al., 2017 came out after we have
prepared our study. Their results suggest that the relationship between $^{231}Pa/^{230}Th$ and
AMOC is not affected by boundary scavenging or bottom scavenging (pointed out in line
200-204). Therefore, since we are focusing on sediment $^{231}Pa/^{230}Th$ instead of water
column activity, our results will not be influenced too much by including nepheloid layer.
However, nepheloid layer must be included in future works. We have included this in line
487-489.

*Lien 242: I don't understand this sentence at all. There is a reference to statistical values*
*in Fig. 4a which are not there.*
Sorry there is no longer values in Fig.4 after modification in our last version. We have
deleted this reference in the text (line 240).

*Figure 5: I cannot follow the statistics provided here. The yellow points in 5a hardly lead*
*to a slope close to 1.*
The purple line in Fig.5 is the least squared liner regression. For shallow layers, model
results are much smaller than observation (red, blue dots). For deep layers, model results
are much larger than observation (yellow dots). The least squared method of regression
gives the result of the purple line and the slope is 1.02.

*Line 259: I think I cannot accept that the dissolved fractions are simulated so utterly bad,*
*simply "[…] because boundary scavenging and sediment resuspensions are not included*
*in our model […]". I suggest first that the authors re-examine the observational data.*
*Which of the outliers (e.g. Fig. 5c) are reliable values with reasonable errors? Because*
*what can we learn from a parameter set and model which is not able to reconstruct the*
*magnitude of the particulate fraction. If this was already a problem in the studies by*
*(Dutay, Lacan, Roy-Barman, & Bopp, 2009) and (Siddall et al., 2005), why not*
*recalibrate the model? How did (Rempfer et al., 2017) cope with this problem?*
In Fig 2 and 3, we show water column dissolved $^{231}Pa$ and $^{230}Th$ activity and the
particulate $[^{231}Pa/^{230}Th]_p$ along two GEOTRACES transects. What each figure is about is
listed at the top left of each figure. In both transects, the dissolved $^{231}Pa$ and $^{230}Th$ activity
is too large in the abyssal compared with observations. That why we state that "Our
model is unable to simulate the realistic dissolved 231Pa and 230Th activities in abyssal"
in line 257-258. In Rempfer et al., 2017, they also show dissolved $^{231}Pa$ and $^{230}Th$ activity
and the particulate $[^{231}Pa/^{230}Th]_p$ along the same two GEOTRACES transects. They show
results in Re3d (without boundary scavenging and nepheloid layer), Rd3d_Bd (with
boundary scavenging but with nepheloid layer), and Rd3d_BtBd (with boundary
scavenging and nepheloid layer). There results shows that water column dissolved $^{231}Pa$
and $^{230}Th$ activity is very large in abyssal if there is no boundary scavenging and
nepheloid layer (Re3d), which is similar to our results. But in Rd3d_BtBd, the water
column dissolved $^{231}Pa$ and $^{230}Th$ activity is in the right magnitude compared with
observation. This suggests that boundary scavenging and nepheloid layer are important for simulating dissolved $^{231}$Pa and $^{230}$Th activity in abyssal. That's why we state that
"With boundary scavenging and sediment resuspensions added, dissolved $^{231}$Pa and $^{230}$Th
activities in the abyssal should be reduced" in line 259-261.
*Line 281: "The sediment 231Pa/230Th in CTRL is overall consistent with observations*
*[...]". Wouldn't it be interesting to go into more detail here? Where are they consistent?*
*Which basin, which water depth? Is margin distance an issue? By carving out which*
*region is worse represented than others a lot could be learned about and from the model.*
*E.g. Southern Ocean: because opal fluxes are so high 231Pa/230Th can vary a lot (much*
*more than in the Atlantic). Simulating correct absolute values is almost impossible*
*because opal flux varies on very small spatial scales, which cannot be captured by any*
*model. Thus, the quality of the model run assessed by observations from this area will*
*inevitably lead to bad agreement.*
We appreciate this suggestion.  In this part, we are focusing on large scale sediment
$^{231}$Pa/$^{230}$Th distribution, which our model is able to capture as discussed in line 281-296.
We did not go into details about sediment $^{231}$Pa/$^{230}$Th distribution because it is not the
focus of this study. But details of model sediment $^{231}$Pa/$^{230}$Th performance can be useful,
for example, to improve model biogeochemical module, and therefore worth further
study.
*Line296: Where is the statement given here shown/demonstrated? Figure?*
In Fig 1c, there is an opal maximum at about 40°N in the Atlantic. In this region,
sediment $^{231}$Pa/$^{230}$Th is also larger than surroundings (Fig. 4). We add this in line 296.
*Line 303, Fig.6: I cannot follow the argumentation here. It would be necessary to*
*increase the scale on Fig. 6and b in order to better resolve the high values. At the*
*moment any variations are hidden within the red colour. The finding, that K influences*
*dissolved fractions but not particulate fractions needs much more explanation. The*
*simplification with reference to Eq. 3 and 7 does not help much.*
Thanks for the suggestion. We have changed the color scale for Fig. 6 a and b. The
overall structures of dissolved $^{231}$Pa and $^{230}$Th activity are similar in two sensitivity
experiments, but the magnitude is much larger in Exp_1 (smaller K) than Exp_2 (larger
K).
We have re-written this part (line 303-332). We first derive the particulate and dissolved
isotope activity under the assumption that there is no isotope decay and no ocean
transport (Eq. 7 and 8). This can help us understand the difference between Exp_1 and
Exp_2. For dissolved isotope activity, Eq. 8 suggests that increased K will lead to
decreased dissolved isotope activity. For particulate isotope activity, Eq. 7 suggests that
particulate isotope activity is independent of K. Therefore, particulate isotope activity in
Exp_1 and Exp_2 does not change too much, especially compared with the changes in
dissolved isotope activity.
*Table 2: More realistic values for EXP1 and 2 would be appreciated in order to derive*
*helpful insights from the model runs.*

In (Siddall et al., 2005), they show model sensitivity with K one order of magnitude
larger or smaller than the CTRL. Our experiments are similar to theirs. We want to show
that how water column activity and sediment $^{231}Pa/^{230}Th$ change with K and also the K
used in the control experiment is of the right order of magnitude. We agree that more
experiments with K changes slightly around control will be helpful. This will act as a
parameter tuning process and worth the effort in future studies. We have added this part
in line 489-492.
*Line 329: This statement should be proved statistically (like Fig. 5).*
Thanks for this suggestion. We have included the RMSE for different experiments in line
339-340. The parameters in CTRL produce the minimum RMSE comparing with Exp_1
and Exp_2.
*Line 360: In the following paragraph the effects of opal on 231Pa/230Th is discussed.*
*However, the model generates opal fluxes not in agreement with reality. In the response*
*to the reviewer the authors claim that the large scale global opal production is reflected*
*well in the model (e.g. high in SO). I agree. They also claim that the question, why the*
*models produces a "fake-bloom" of opal production in the Western North Atlantic, is*
*beyond the scope of this study. I may accept this (but then one may questioning the*
*validity of the model approach), however in this case the paragraph following line 360*
*needs to be written more carefully and with a clear statement, that opal is not well*
*represented on smaller spatial scales. Same with line 409.*
Thanks for this suggestion. It is hard to reproduce the productivity everywhere, especially
on small scales. The productivity pattern produced by the biogeochemical module is
consistent with observations over most regions. Model is never perfect. As long as it can
help us understand something, it is useful. The opal bloom in the northwest Atlantic
produced by the biogeochemical module in the CESM is not in the observation. But at
least the pattern of Pa/Th response to the fresh water hosing is self-consistent with the
productivity pattern in our model and can give some insights of interpreting sediment
Pa/Th. We have added this part in line 377-380.
*Line 419: Of course studies on AMOC reconstructions need to cross check opal fluxes,*
*but this sentence spreads a way too negative message when based on unrealistic opal*
*fluxes and hence I do not agree. Please rephrase.*
Thanks for this suggestion. We have changed in line 438-439.
*Fig 9: the difference between coupled and fixed are partly so big, that I wonder how both*
*methods did agree so well before. Differences in the range of Δ231Pa/230Th>0.1 (e.g.*
*9d) are not increasing my confidence in the model. Observations are much more*
*constrained. Again I plead for applying realistic model parameters only. Further, I could*
*not find information on water depth and longitude of the values shown in Fig. 9*
*diagrams, which are essential for the interpretation.*
The big difference between coupled and fixed in HOSING is the point we want to make:
the particle fields matters. At time 0, when there is no freshwater forcing (CTRL

experiment), the fixed and coupled are the same. This is what shown in Fig. 4 and
discussed in line 235-244: in CTRL, p-fixed and p-coupled results are identical, which is
because the particle fields are essentially the same (control experiment, no extra forcing).
However, when freshwater is added, both AMOC and particle field produced by
biogeochemical module (Fig.8) change. E.g. Fig.9d, for p-fixed curve (green), the particle
field is held the same. The increase of Pa/Th is caused by the reduce of AMOC. For p-
coupled curve (red), AMOC will lead to an increase in Pa/Th (similar to the green curve),
but particle change effect at this site will lead to a decrease in Pa/Th. Therefore, p-
coupled Pa/Th at equilibrium (red) is much smaller than p-fixed Pa/Th (green). The
difference between p-fixed and p-coupled is caused by the change of particle fields (Fig.
8). In reconstructions, it is hard to know how much Pa/Th change is caused by AMOC
and how much is by particle. But in our model, by comparing p-fixed with p-coupled
results, we can detangle the AMOC effect and particle effect. This is the point we have
emphasized in several places (e.g. line 89-93, line 411-416 and line 417-439)
Fig.9 c-f are four sites picked in the North Atlantic to representing different mechanisms.
The depth of each site is decided by the model topography (bottom cell at that location).
Locations for Fig.9 c-f are picked for different reasons: (c) is a location in high latitude
North Atlantic where opal production increases after applying fresh water forcing over
50°N-70°N (Fig. 8f); (d) is a location where opal production is the maximum in our
model; (e) and (f) are locations near Bermuda Rise (McManus et al., 2004). These four
locations behave differently in HOSING as discussed in section 4.3 (line 411-439)
*Line 424: Why is there a decrease of 231Pa/230Th above 2 km only? To my*
*understanding and as stated in line 442 the decrease affects all of the NADW seized*
*water depths.*
The pattern in Fig. 12 is also produced in Rempfer et al., 2017 (their Fig. 8). Vertical
decrease of Pa_p/Th_p is suggested to be caused by the lateral transport by AMOC (line
277-280). Northward transport in the upper limb of AMOC will lead to $^{231}$Pa import
while southward transport in the lower limb of AMOC will lead to $^{231}$Pa export.
Therefore, we see a vertical decrease of $^{231}$Pa/$^{230}$Th (Fig. 12c). If there is no more ocean
transport by AMOC, then there is no more $^{231}$Pa import for upper layer and $^{231}$Pa/$^{230}$Th
shows a decrease, and vice versa for deep layer. This is explained in detail in line 451-
463.
*Line 460: Yes, the parameters are somewhere in the range of the right magnitude, but not*
*more. It would be great if this study would help to represent 231Pa/230Th in a realistic*
*model, not only somewhere in the range of a factor of 25.*
We have change this to "right order of magnitude" (line 341).
*Fig10b: site locations are not visible.*
We have enlarged the site location in this figure.

*Fig12c: Please explain the change of direction of 231Pa/230Th with depth at about*
*4000m for ON*
The increase of Pa/Th at about 4,000m for AMOC_on case is probably caused by AABW
transport. AABW from the Southern Ocean transport [231]Pa enriched water northward,
which results in the increase of Pa/Th. This is similar to the argument by (Thomas et al.,
2006) (line 66-72).

---

## Author Response (AR3)

Dear editor,

Thanks for your comments.

In the following, we have addressed all the comments, with the original review text underlined in italics and red. Lines referred in this reply to comments are lines in the final version instead of the version with tracked changes.

Regarding your response to the reviewer, saying 'There is no particular reason for the iterating and non-iterating grey layers. The left column is references for water column activity and the right column is for Holocene core top Pa/Th. Some references have both column activity and Pa/Th, therefore appear twice'. Please add this explanation to the caption, or remove duplicate references.

We have added "left column" and "right column" in the caption (line 918-919) and there are no duplicate references in each column.

Regarding response to the reviewer ending in 'Our study is the first step trying to implementing 231Pa and 230Th into CESM. The parameters can be improved in the future with more observations available (line 489-492)'. Please can you add this point to the text. At present, you say 'In addition, partition coefficient for different particles can be further tuned in the future, which can improve our understanding of the affinity of 231Pa and 230Th to different particles'. This leaves the reader wondering why you have not 'further tuned' the model. What I think you mean to say is that the parameters will hopefully be better constrained when more observational evidence becomes available?

Thanks for this suggestion. The partition coefficient tuning requires a lot of computational resources, which is beyond our resources at this stage. The tuning process includes two parts: the relative affinity of Pa and Th to different particles and also the absolute value of the partition coefficient. This is 3 degrees of freedom, which should be varied explicitly, therefore a lot of experiments are required, which is beyond the purpose of this study and needs future work. This is also discussed in the last comments. The last paragraph has been improved.

Regarding your response to the reviewer comment which start with 'Line 230 and Fig. 9: a freshwater...' This is fine. This is a model description paper, rather than a paper trying to present new scientific results. However, please can you refer to the previous literature when describing this.

In our ocean alone model, if we add 0.1Sv of fresh water to the North Atlantic, the AMOC cannot be shut down. This is also suggested in Stouffer et al., 2006. In their study, AMOC is reduced by about 25% from different model ensemble if only 0.1Sv is applied.

Following the reviewers comments about figure 5, is the purple the least squares regression through all of the data (i.e. all depth ranges)? Please explain in the caption. I also suggest that you change the plotting style so that points behind other points are still visible. It would also be valuable for the reader to see the slopes of the relationships in

the different water depths. For example, the near 1:1 slope in fig. 5a is the result of cancelling slopes at different depths. the purple line is therefore not particularly useful.

Thanks for your suggestion. We have modified Figure 5. We change the size of the dots so that it looks less clustered. If we further make the dots smaller, it will be hard to see. Several data points were not successfully included in the previous version. Now the problem is fixed and this causes the small change in the slope for Pa\_d and Pa\_p. But the overall features are the same. We add the regression lines for each depth using the same color. For example, the dissolved Pa and Th in depth deeper than 3,000 m in the model is systematically larger than observation (yellow). Description of each lines are added in the caption.

Figures 2 and 3 captions must be improved. Explain what the filled contours are and what the circles are. It is obvious to someone who is familiar with the observational dataset, but not to others.

Thanks for the suggestion. We have improved the captions in Fig.2 and Fig.3.

In your response to the reviewer ending in "With boundary scavenging and sediment resuspensions added, dissolved 231Pa and 230Th activities in the abyssal should be reduced" in line 259-261.' This statement is true, but what the reviewer took issue with is the fact that you state that 'Our model is unable to simulate the realistic dissolved 231Pa and 230Th activities in abyssal because boundary scavenging and sediment resuspensions are not included in our model.' You provide a plausible hypothesis for this, but yours is a statement of fact. This requires further investigation. We have improved to make this point clearer in line 257-265.

Regarding the reviewers comment beginning with 'Line 281: "The sediment 231Pa...', I do not feel that you have addressed this comment adequately in your response. I would like to see this explored further, as the reviewer asks.

*The original reviewer's comment:*

Line 281: "The sediment 231Pa/230Th in CTRL is overall consistent with observations [...]". Wouldn't it be interesting to go into more detail here? Where are they consistent? Which basin, which water depth? Is margin distance an issue? By carving out which region is worse represented than others a lot could be learned about and from the model. E.g. Southern Ocean: because opal fluxes are so high 231Pa/230Th can vary a lot (much more than in the Atlantic). Simulating correct absolute values is almost impossible because opal flux varies on very small spatial scales, which cannot be captured by any model. Thus, the quality of the model run assessed by observations from this area will inevitably lead to bad agreement.

We compare sediment  ${}^{231}Pa/{}^{230}Th$  performance in different basins and Atlantic is better than other basins. The results are shown in Table S1 in the supplementary information and discussed in line 300-310.

Regarding the reviewers comment 'Line 329: This statement should be proved statistically (like Fig. 5)', as far as I can see, adding the RMSE has not demonstrated this

point. I agree with the reviewer that analysis like that used in figure 5 is the sort of thing that is required.

Thanks for this suggestion. We have added the linear regression coefficient for different experiments in Table S1 in the supplementary information and discussed this in line 354-363.

In addition, in Rempfer et al. 2017, they use the RMSE as the only criteria for model performance.

Regarding the reviewer's comment beginning with 'Line 360: In the following paragraph', I agree with the reviewer that this section still needs to be more carefully written. Reading this paragraph at present it is still not clear what the model can and can not reproduce. It is positive to acknowledge the limitations. Please also expand on what you mean by 'fresh water hosing is self-consistent with the productivity pattern'.

With reduced AMOC, sediment Pa/Th in the North Atlantic should decrease. However, the magnitude of the decrease depends on the distribution of particle flux, especially opal flux, because of the particle flux effect explained in line 393-409. This is the main point of this paragraph. We choose the 40°N western Atlantic as an example. In this region, opal flux is the regional maximum in the North Atlantic. The sediment Pa/Th increase in this region is also the regional maximum. This is what we mean the response of Pa/Th is self-consistent with the particle flux in our model.

Perhaps the most important reviewer's point that I do not feel has been addressed here is made clear by the reviewers comment 'Line 492: Yes, the parameters are somewhere in the range of the right magnitude, but not more. It would be great if this study would help to represent 231Pa/230Th in a realistic model, not only somewhere in the range of a factor of 25.'. The point being made is that there is observational data which can help constrain the parameters. The review has clearly asked in a number of places for an experiment to be done with parameters chosen to reflect this understanding. Please can you either undertake this experiment, or justify robustly why you do not feel that this is useful?

This can be referred to the second comments in this reply. First of all, tuning parameters requires a lot of computational resources and efforts. Secondly, our sediment Pa/Th and water column activity can capture the major features of the observations and the response of Pa/Th in the idealized hosing experiment can be understood and have some implications. Ideally, we would love to carry out many more sensitivity experiments to further improve the parameters. In practice, at present, this is beyond our resources at this stage. Therefore, the major purpose of this paper is to show the performance of this base version. In the future, it is our strong desire to further improve this model with more experiments, parameter turning and the implementation of additional processes, such as the nepheloid layer included. This is reflected in the last paragraph of the paper.

**Reference:**

Stouffer, R. J., Yin, J., Gregory, J. M. J. M., Dixon, K. W., Spelman, M. J., Hurlin, W., ... Weber, N. (2006). Investigating the causes of the response of the thermohaline circulation to past and future climate changes. *Journal of Climate*, 19(8), 1365–1387. https://doi.org/10.1175/JCLI3689.1

| 231 Pa and 230 Th in the ocean model of the Community Earth System Model+ | Formatted: Justified, Tabs:Not at 0.35"           |
|-------------------------------------------------------------------------------------------------|---------------------------------------------------|
| (CESM1.3)                                                                                       | Formatted: Font:Times, Not Bold, Font color: Text |
| Sifan Gu 1 , Zhengyu Liu 2                                                | 1                                                 |
|                                                                                                 | Deleted: 1                                        |

1Department of Atmospheric and Oceanic Sciences and Center for Climate Research,

University of Wisconsin-Madison, Madison, WI, USA

2.Atmospheric Science Program, Department of Geography, Ohio State University, Columbus, OH, USA

Correspondence to: Sifan Gu (sgu28@wisc.edu)

**Abstract**

Sediment 231Pa/230Th activity ratio is emerging as an important proxy for deep ocean circulation in the past. In order to allow for a direct model-data comparison and to improve our understanding of sediment 231Pa/230Th activity ratio, we implement 231Pa and 230Th in the ocean component of the Community Earth System Model (CESM). In addition to the fully coupled implementation of the scavenging behavior of 231Pa and 230Th with the active marine ecosystem module (pcoupled), another form of 231Pa and 230Th 
[revised manuscript text omitted]
 231Pa and 230Th in CTRL, but only p-fixed 231Pa and 230Th in sensitivity experiments. Equilibrium partition coefficients for 231Pa and 230Th vary among different particle types and the magnitude of the partition coefficients for different particle types remains uncertain (Chase et al., 2002; Chase and Robert F, 2004; Luo and Ku, 1999). Since the control experiment in Siddall et al., (2005) is able to simulate major features of 231Pa and 230Th distributions, we use the partition coefficients from the control experiment in Siddall et al., (2005) in our CTRL (Table 2). Two sensitivity experiments are performed with decreased (EXP\_1) and increased (EXP\_2) partition coefficients by a factor of 5 (Table 2).

All the experiments are ocean-alone experiments with the normal year forcing by CORE-II data (Large and Yeager, 2008). The 231Pa and 230Th activities are initiated from 0 in CTRL and are integrated for 2,000 model years until equilibrium is reached. EXP\_1 and EXP\_2 are initiated from 1,400 model year in CTRL and are integrated for another 800 model years to reach equilibrium.

Since sediment 231Pa/230Th in North Atlantic has been used to reflect the strength of AMOC, to test how sediment 231Pa/230Th in our model responds to the change of AMOC and the change of particle fluxes, we carried out a fresh water perturbation experiment (HOSING) with both p-fixed and p-coupled 231Pa and 230Th. Starting from 2,000 model year of CTRL, a freshwater flux of 1 Sv is imposed over the North Atlantic region of 50°N~70°N and the experiment is integrated for 1400 model years until both p-fixed and p-coupled sediment 231Pa/230Th ratio have reached quasi-equilibrium. The partition coefficients used in HOSING are the same as in CTRL.

**4. Results**

**4.1 Control Experiment**

P-fixed and p-coupled version of 231Pa and 230Th in CTRL show identical results (Fig. 2-4). P-fixed and p-coupled dissolved and particulate 231Pa and 230Th in CTRL are highly correlated with each other with correlations greater than 0.995 and

regression coefficients are all near 1.0 ( $R^2$ >0.995). The correlation coefficient between p-fixed and p-coupled sediment  ${}^{231}Pa/{}^{230}Th$  activity ratios in CTRL is 0.99 and the regression coefficient is 0.9 ( $R^2$ =0.98). This is expected because the particle fields used in p-fixed version are prescribed as the climatology of the particle fields used in the p-coupled version. Therefore, under the same climate forcing, p-fixed and p-coupled version of  ${}^{231}Pa$  and  ${}^{230}Th$  should be very similar. For the discussion of results in CTRL below, we only discuss the p-fixed  ${}^{231}Pa$  and  ${}^{230}Th$ .

The residence time of both 231Pa and 230Th in CTRL are comparable with observations. The residence time is calculated as the ratio of global average total isotope activity and the radioactive ingrowth of the isotope. Residence time in CTRL is 118 yr for 231Pa and 33 yr for 230Th (Table 2), which are of the same magnitude as 111 yr for 231Pa and 26 yr for 230Th in observation (Yu et al., 1996).

CTRL can simulate the general features of dissolved water column 231Pa and 230Th activities. Dissolved 231Pa and 230Th activities increase with depth in CTRL, as shown in two GEOTRACES transects (Deng et al., 2014; Hayes et al., 2015) in the Atlantic (Fig. 2 and 3). The dissolved 231Pa and 230Th activities in CTRL are also at the same order of magnitude as in observations in the most of the ocean, except that simulated values are larger than observations in the abyssal, which is also the case in Siddall et al., (2005) and Rempfer et al., (2017) (their Fig. 2 and 3, experiment Re3d). Our model is unable to simulate the realistic dissolved 231Pa and 230Th activities in the abyssal probably because boundary scavenging and sediment resuspensions are not included in our model. In Rempfer et al., 2017, without boundary scavenging and sediment resuspension, dissolved 231Pa and 230Th activities are quite large in the deep ocean. However, if boundary scavenging and sediment resuspension are included, the water column dissolved 231Pa and 230Th activity is in the right magnitude compared with observation. Therefore, we hypothesize that with boundary scavenging and sediment resuspensions added, dissolved 231Pa and 230Th activities in the abyssal should be greatly reduced.

A more quantitative model-data comparison is shown in Fig. 5. The linear regression coefficient between model results and observations (references of observations are listed in Table 3), an indication of model ability to simulate 231Pa

and 230Th activity (Dutay et al., 2009), is near 1.0 for dissolved 231Pa and 230Th (1.02 for  $[^{231}\mbox{Pa}]_d$  and 1.14 for  $[^{230}\mbox{Th}]_d)$  , suggesting that CTRL can simulate the dissolved 231Pa and 230Th in good agreement with observations. However, the simulation of the particulate activity is not as good as the dissolved activity. Particulate activity is overall larger than observation in the surface ocean and smaller than observation in the deep ocean for both particulate 231Pa and 230Th. The regression coefficient for particulate 231Pa and 230Th is 0.02 for [231Pa]p and 0.05 for [230Th]p. The poor performance in simulating water column particulate 231Pa and 230Th activities is also in previous modeling studies (Dutay et al., 2009; Siddall et al., 2005), because of similar modelling scheme applied. However, the simulated  ${}^{231}Pa_p/{}^{230}Th_p$  is in reasonable agreement with observations. The 231Pap/230Thp along two GEOTRACES transects (Fig. 2 and 3) show the similar pattern and magnitude as in Rempfer et al., (2017), consistent with observations. Decrease of 231Pap/230Thp with depth is well simulated, which is suggested to be caused by the lateral transport of 231Pa from North Atlantic to Southern Ocean by AMOC (Gherardi et al., 2009; Lippold et al., 2011, 2012a; Luo et al., 2010; Rempfer et al., 2017).

The sediment 231Pa/230Th in CTRL is overall consistent with observations (references of observations are listed in Table 3). The North Atlantic shows low sediment 231Pa/230Th activity ratio as in observations because 231Pa is more subject to the southward transport by active ocean circulation than 230Th because of its longer residence time. The Southern Ocean maximum in the sediment 231Pa/230Th activity ratio is also simulated in CTRL. High opal fluxes in the Southern Ocean, which preferentially removes 231Pa into sediment ( $K_{opal}^{231Pa} > K_{opal}^{230}$ Th) (Chase et al., 2002), leading to increased sediment 231Pa/230Th activity ratio. In addition, upwelling in the Southern Ocean brings up deep water enriched with 231Pa, which is transported from the North Atlantic, to shallower depth and further contribute to the scavenging. CTRL can also produce higher sediment 231Pa/230Th activity ratio in regions with high particle production (e.g. the Eastern equatorial Pacific, the North Pacific and the Indian Ocean) due to the "particle flux effect". Specifically, in North Atlantic, the distribution of sediment 231Pa/230Th matches the distribution of

particle, especially opal, production: sediment 231Pa/230Th is higher where opal production is high, and vice versa (Fig. 4 and Fig. 1c). Quantitatively, the regression coefficient between sediment 231Pa/230Th in CTRL and observation in the Atlantic is 0.86, which is larger than in other basins. This suggests that sediment 231Pa/230Th is better simulated in the Atlantic than in other basins. One possible explanation is that sediment 231Pa/230Th in the Atlantic is controlled by both ocean circulation and particle flux, while in other basins sediment 231Pa/230Th is controlled almost only by particle flux. With active AMOC, the north south gradient of sediment 231Pa/230Th can be simulated. However, for example, in the Southern Ocean, sediment 231Pa/230Th is dominantly controlled by opal flux, which varies on small scales and is difficult for, simulation, Therefore, model performance in simulating sediment 231Pa/230Th in the Southern Ocean is not as good as in the Atlantic.

**4.2 Sensitivity on partition coefficient K**

In this section, we show model sensitivity on partition coefficient by increasing and decreasing the partition coefficient, K, by a factor of 5, but keeping the relative ratio for different particles the same (Table 2). Our model shows similar model sensitivity as in Siddall et al., (2005) as discussed below.

As stated in Siddall et al., (2005), the isotope decay term in Eq. (3) is three orders of magnitude less than the production term. If we neglect the transport term and the decay term in Eq. (3) and assume particulate phase activity at the surface as 0, when reach equilibrium, the activity of particulate phase will be as in Eq. (7). Eq. (7) combined with Eq.(2) and  $R_i = \frac{F}{w_S * \rho}$ , we can obtain Eq.(8). Under the assumption that there is isotope decay and ocean transport, Eq. (7) suggests that the particulate isotope activity depends on the production rate and settling velocity and will increase linearly with depth. Eq. (8) suggests that the dissolved isotope activity depends on the production rate and settling with depth is linear relationship with depth is suggested by observations (Bacon and Anderson,

1982; Roy-Barman et al., 1996). Results of Eq. (7) and Eq. (8) can help to understand the differences in Exp\_1 and Exp\_2.

Increasing K will decrease water column dissolved 231Pa and 230Th activities but won't change particulate 231Pa and 230Th too much (Fig. 6). Magnitude of dissolved 231Pa and 230Th in Exp\_1 (smaller K) is at least one order larger than that in Exp\_2 (larger K), while magnitude of particulate 231Pa and 230Th in Exp\_1 and Exp\_2 is in the same order. As suggested by Eq. (8), if there is no isotope decay and no ocean transport, larger K will lead to smaller dissolved isotope activity but unchanged particulate activity. Intuitively, larger K will lead to more 231Pa and 230Th attached to particles and further buried into sediment, which increases the sink for the 231Pa and 230Th budget. With the sources for 231Pa and 230Th staying the same, dissolved 231Pa and 230Th 
[revised manuscript text omitted]

[... [1]]

---

## Author Response (AR4)

Reply to comments
Dear editor,
Thanks for your comments. In the following, we have addressed the comment, with the
original review text underlined in italics and red.
*Perhaps the most important reviewer's point that I do not feel has been addressed here is*
*made clear by the reviewers comment 'Line 492: Yes, the parameters are somewhere in*
*the range of the right magnitude, but not more. It would be great if this study would help*
*to represent 231Pa/230Th in a realistic model, not only somewhere in the range of a*
*factor of 25.'. The point being made is that there is observational data which can help*
*constrain the parameters. The review has clearly asked in a number of places for an*
*experiment to be done with parameters chosen to reflect this understanding. Please can*
*you either undertake this experiment, or justify robustly why you do not feel that this is*
*useful?*
*Your argument against this is (1) it is not feasible to undertake a large ensemble of*
*sensitivity experiments, and (2) the model does an adequate job of capturing the major*
*features as it is. However:*
*(1) Here the reviewer is not asking for a large sensitivity analysis, simply a single run to*
*be done with more realistic parameters, i.e. parameters derived from observations. I*
*would imagine that this is not too resource intensive?*
*(2) If the model is capturing the observed behaviour with unrealistic parameters, either*
*the parameters are not important, or there is something wrong.*
*Please can you either perform this simulation and present the results, or explain why this*
*is not a sensible approach?*
First of all, to answer your question 2 "*If the model is capturing the observed behavior*
*with unrealistic parameters....*", the parameters used in our control experiment is not
unrealistic. It is suggested by observations (Chase et al., 2002). In the two sensitivity
experiments, we increase and decrease the parameters by a factor of 5 (line 217), not a
factor of 25 (in the reviewer's comment). Compared with two sensitivity experiments, the
control experiment is better simulating Pa and Th (discussed in section 4.2). Therefore,
we state that "the partition coefficient in CTRL is of the right order of magnitude". Using
the parameters suggested by observation (Chase et al., 2002), our model is able to capture
the major features of Pa and Th.
We do have an experiment with parameters increased by a factor of 1.5 (Exp3). The
results of Exp3 is similar to CTRL. The water column dissolved Pa and Th (Fig. R1) is
slightly smaller than CTRL (Fig. S3) because of the increased partition coefficient (line
330). The interbasin gradient of sediment $^{231}Pa/^{230}Th$ in Exp3 (Fig. R3) is also slightly
smaller than CTRL (Fig. 4) (line 348). The performance of CTRL is better than Exp3 in that the regression coefficient of sediment $^{231}$Pa/$^{230}$Th is 0.2 in CTRL and 0.11 in Exp3
globally; 0.86 in CTRL and 0.77 in Exp3 in the Atlantic; 0.16 in CTRL and 0.02 in Exp3
in the Pacific; 0.18 in CTRL and 0.11 in Exp3 in the Southern Ocean. Overall, the
difference between Exp3 and CTRL is similar to the difference between Exp2 (increase
parameters by a factor of 5) and CTRL, but with much smaller magnitude. Therefore, we
don't feel it is necessary to show the results of Exp3 in the text.
In Exp3, we only change the magnitude of the partition coefficients and keep the relative
fractionation factor by different particles the same. However, fractionation factor
suggested by different studies also varies (Table A1 from Rempfer et al., 2017).
Therefore, when tuning parameters in future studies, both magnitude of partition
coefficients and the fractionation factor by different particles should vary systematically
to test which combination yields the best results (this is also discussed in our previous
reply to comments). But at current stage, this kind of experiments is beyond our
resources.

[Figure]

Figure R1. Atlantic zonal mean dissolved and particulate $^{231}$Pa and $^{230}$Th in Exp3.

[Figure]

Figure R2. Scatter plot of global dissolved and particulate [231]Pa and [230]Th between
observation and model results in Exp3 (unit: dpm/m3). (a) dissolved [231]Pa; (b) particulate
[231]Pa; (c) dissolved [230]Th; (d) particulate [230]Th. Lines and colors are the same in Figure 5.

[revised manuscript text omitted]